# Foundations of Top-$k$ Decoding for Language Models

**Georgy Noarov**[1]* **Soham Mallick**[1]* **Tao Wang**[1]*

**Sunay Joshi**[1]    **Yan Sun**[2]    **Yangxinyu Xie**[1]    **Mengxin Yu**[3]    **Edgar Dobriban**[1]

[1] University of Pennsylvania
[2] New Jersey Institute of Technology  [3] Washington University in St. Louis

## Abstract

Top-$k$ decoding is a widely used method for sampling from LLMs: at each token, only the largest $k$ next-token-probabilities are kept, and the next token is sampled after renormalizing them to sum to unity. Top-$k$ and other sampling methods are motivated by the intuition that true next-token distributions are sparse, and the noisy LLM probabilities need to be truncated. However, to our knowledge, a precise theoretical motivation for the use of top-$k$ decoding is missing. In this work, we develop a theoretical framework that both explains and generalizes top-$k$ decoding. We view decoding at a fixed token as the recovery of a sparse probability distribution. We introduce *Bregman decoders* obtained by minimizing a separable Bregman divergence (for both the *primal* and *dual* cases) with a sparsity-inducing $\ell_0$-regularization; in particular, these decoders are *adaptive* in the sense that the sparsity parameter $k$ is chosen depending on the underlying token distribution. Despite the combinatorial nature of the sparse Bregman objective, we show how to optimize it efficiently for a large class of divergences. We prove that (i) the optimal decoding strategies are greedy, and further that (ii) the objective is discretely convex in $k$, such that the optimal $k$ can be identified in logarithmic time. We note that standard top-$k$ decoding arises as a special case for the KL divergence, and construct new decoding strategies with substantially different behaviors (e.g., non-linearly up-weighting larger probabilities after renormalization).

## 1   Introduction

Large language models (LLMs) are powerful generative AI tools for producing text. When pre-trained on large text corpora and aligned according to human preferences, they can be used for a wide range of tasks. On a technical level, they are probability distributions over text: given any user text prompt $x$, an LLM samples an answer $Y \sim \pi(\cdot|x)$ from a probability distribution $\pi(\cdot|x)$ over text. However, even after obtaining a pre-trained, fine-tuned, and human preference-aligned model $\pi$, it is uncommon to directly sample from the model. Instead, several sampling/decoding methods are commonly used, including top-$k$ [21] or top-$p$ sampling [32]. Due to their improved empirical performance compared to direct sampling, they are used by default or as an option in many popular LLMs, including the GPT series, Gemini, and Claude. From a broader perspective, per-token samplers/decoders belong to an expanding collection of post-hoc methods for improving LLM performance, which range from pre-sampling transforms (e.g. temperature decoding), to sequence-level decoding strategies (e.g. beam search), to post-hoc selection (e.g. best-of-$N$ or self-consistency), to a variety of test-time scaling approaches; see e.g., [12, 21, 32].

In this paper, we focus on decoding methods that modify each next-token-probability distribution to induce *sparsity*, i.e., to keep only a small number of tokens with a nonzero probability. This includes the widely used top-$k$ [21] and top-$p$ [32] sampling methods, among others. These methods

---

*Co-first authors.  Correspondence to: `gnoarov@seas.upenn.edu`, `kcillam@wharton.upenn.edu`, `tawan@wharton.upenn.edu`, `dobriban@wharton.upenn.edu`.

are motivated by the intuition that the noisy LLM probabilities need to be truncated to denoise the "unreliable tail" [32]. In particular, we focus on the popular top-$k$ decoding method, which keeps only the largest $k$ next-token-probabilities at each decoding step. These are renormalized—via dividing by their sum—to a probability distribution from which the next token is sampled.

Despite the wide use and rich intuition behind top-$k$ decoding, to our knowledge, a precise theoretical understanding of top-$k$ decoding is not available (see Section 6 for a discussion of related work). Therefore, in this work, we develop a theoretical framework that flexibly generalizes and sheds light on the key properties of top-$k$ decoding. For a fixed token, we view decoding as recovering a sparse probability distribution from a given (generally non-sparse) LLM token distribution. We consider denoisers obtained by minimizing a Bregman divergence (such as KL divergence or Brier score) from the "raw" LLM token distribution, with a sparsity-inducing $\ell_0$ regularization. This approach is motivated by a rich literature of both Bregman divergences and sparsity, see Section 6 for details.

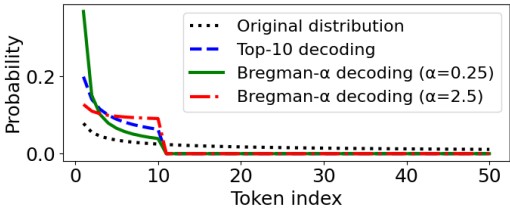

**Bregman renormalizations.** Our approach both generalizes top-$k$ decoding and opens up a rich field of efficient adaptive "Bregman decoding" methods with a wide and tunable range of behaviors. As an example, we consider Bregman divergences generated by the $\alpha$-entropies $x \mapsto x^\alpha/[\alpha(\alpha-1)]$ [29, 51], and display in the adjacent figure how Bregman decoders modulate a token distribution for several values of $\alpha$. For $\alpha \to 1$, we obtain standard top-$k$ decoding (for $k = 10$ here). By contrast, for $\alpha = 0.25$, the decoder shifts most of the mass onto the top few tokens; while for $\alpha = 2.5$, the mass is spread much more uniformly across the top-$k$ tokens. This exemplifies how our framework enables the design of novel decoders eliciting a wide range of behaviors.

**Provable adaptivity.** An important feature of our framework is that it studies, and provides, provably **adaptive** decoding strategies. Namely, given any raw LLM token probability vector $p$, our Bregman decoders effectively perform (a generalization of) top-$k^*$ decoding of $p$ for an *optimal* $k^* = k^*(p)$: the utilized $k^*$ varies depending on the LLM token distribution $p$, and is chosen to minimize the decoder's $\ell_0$-regularized Bregman divergence from $p$. This rigorous sparse-objective-centric foundation of adaptivity in LLM decoding is, to our knowledge, new in the literature. Moreover, perhaps surprisingly, we are able to show in substantial generality that an optimal $k^*$ can be found provably and efficiently without relying on grid search or other heuristics.

## 1.1 A roadmap of our contributions

In Section 2, we introduce our theoretical framework. We view top-$k$ decoding strategies as two-step: (i) select a number of tokens $k$, and (ii) renormalize the selected $k$ tokens' entries to a probability distribution (Section 2.1). We introduce two rich classes of decoding strategies (Section 2.2): **primal Bregman decoding** and **dual Bregman decoding**. These correspond to $\ell_0$-regularized minimization of a Bregman divergence to the "raw" LLM distribution over tokens, in its first vs. second argument.[2]

In general, $\ell_0$-regularization leads to combinatorial optimization problems, for which there are no known polynomial-time algorithms [11, 42]. Our main contribution is to show that, despite this, the sparse Bregman decoding objective can be efficiently optimized under mild assumptions, by virtue of having two key structural properties: (1) **Greedy selection**: Choosing the $k$ largest probabilities is optimal (Theorems 3.2 and 3.3 in Section 3.2); (2) $k$-**convexity**: Searching for the optimal $k^*$ is a (discretely) convex problem in $k$ (Theorem 3.4 in Section 3.3). While simple to state and desirable, these properties are non-trivial to establish, and require a range of novel structural insights into the sparse Bregman objective that could be of independent interest.

In Section 4, we illustrate our theory by introducing $\alpha$-Bregman decoding strategies, generated by Tsallis $\alpha$-entropies $x \mapsto x^\alpha/[\alpha(\alpha-1)]$. We study how their behavior depends on $\alpha$, and highlight several closed-form cases of interest. One example of the optimization-theoretic elegance of $\alpha$-decoders is their convergence to *water-filling* as $\alpha \to \infty$. Finally, in Section 5, we study the empirical performance of some of the novel decoding schemes on open-ended text generation and mathematical problem solving tasks with LLMs, and find that they perform competitively with top-$k$ decoding.

---

[2]Bregman divergences being asymmetric in general, their distinct behavior in both arguments has been widely studied in optimization and statistical learning [see e.g., 1, 10, 24, 56, etc].

## 2    Regularized sparse Bregman decoding

### 2.1    Top-$k$ decoding preliminaries

**Top-$k$ decoding.** Given a probability distribution $p = (p_1, \ldots, p_V)$ (where $V$ stands for "vocabulary size"), and some $1 \leqslant k \leqslant V$, **top-$k$ decoding** first selects the indices $S_k = (i_1, \ldots, i_k)$ of the largest $k$ probabilities, breaking ties arbitrarily. Setting all other coordinates to zero in $p$, one obtains the vector $p[1:k]$ of the $k$ largest entries. Then, it renormalizes this vector by dividing it by its sum. Letting $(p_{(1)}, p_{(2)}, \ldots, p_{(k)}) = (p_{i_1}, \ldots, p_{i_k})$ be the largest $k$ entries of $p$,

$$\text{top-}k(p) = p[1:k] / \left( \sum_{j=1}^{k} p_{(j)} \right). \tag{1}$$

One then draws a sample from the distribution top-$k(p)$.

**Decoding strategies.** Next, we aim to generalize top-$k$ decoding. We will refer to any operator Dec on probability distributions as a *decoding strategy*; formally $\text{Dec} : \Delta_V \to \Delta_V$, where $\Delta_V = \{x \in [0,1]^V : \sum_{i=1}^{V} x_i = 1\}$ is the simplex of $V$-dimensional probability distributions. Observe that top-$k$ decoding consists of two steps: selecting the largest coordinates and renormalizing them. The second step can be viewed as "re-distributing" the probability mass that has been thresholded away by selection among the remaining indices. This step can be performed in a lot of other meaningful ways besides division by the sum. For instance, we may put a larger weight on the larger remaining probabilities, if we consider them more reliable.

**Renormalization.** Motivated by this, we define the notion of a *renormalization* mapping, which takes as input a thresholded probability vector with $k$ nonzero entries remaining. We consider renormalization maps that are *permutation-equivariant*, i.e., when their input is permuted, their output is permuted accordingly; which clearly holds for the sum-division used in top-$k$. Therefore, since the sum of probabilities after selection can be less than unity, we can define them as maps from the *sub-probability simplex* $\Delta_{\mathrm{sub},k} = \{x \in [0,1]^k : \sum_{i=1}^{k} x_i \leqslant 1\}$ to the simplex $\Delta_k$.

**Definition 2.1** (Renormalization). *For a positive integer $k$, we call a permutation-equivariant map $T : \Delta_{\mathrm{sub},k} \to \Delta_k$ a* renormalization map.

A renormalization map can be extended to the full simplex $\Delta_V$, by applying it only on the nonzero coordinates.[3] We can now define generalized top-$k$ decoding as renormalizing the top-$k$ entries via a general renormalization map.

**Definition 2.2** (Generalized top-$k$ decoding). *For a fixed $k$, a generalized top-$k$ decoding strategy $\text{Dec}_{k,T} : \Delta_V \to \Delta_V$, parameterized by the choice of $k$ and renormalization map $T$, takes as input any $V$-class probability vector $p$, thresholds it to the sub-vector $p[1:k]$ consisting of its top-$k$ elements, and renormalizes it to $T(p[1:k]) \in \Delta_V$.*

**Adaptivity.** A natural extension is to choose $k$ adaptively based on $p$. For this, we consider a $k$-selector map $\hat{k} : \Delta_V \to [V] := \{1, \ldots, V\}$, and a collection of renormalization maps $T_k : \Delta_{\mathrm{sub},k} \to \Delta_k$, $k = 1, \ldots, V$. We define an *adaptive generalized top-$k$ decoding strategy* $\text{Dec}_T : \Delta_V \to \Delta_V$ via $p \mapsto T_{\hat{k}(p)}(p[1:\hat{k}(p)])$. Below, we will design specific renormalizers $T$ and ways to choose $k$.

### 2.2    Regularized sparse Bregman decoding

**Decoding via sparse divergence minimization.** Consider a divergence $\text{Div}(\cdot, \cdot) : \Delta_V \times \Delta_V \to \mathbb{R}$ between two distributions. Classical examples include the squared error $\text{Div}(p, q) = \|p - q\|_2^2$ and the KL divergence $\text{Div}(p, q) = \sum_{j=1}^{V} p_j \ln(p_j / q_j)$. We define the decoding strategy $\text{Dec}_{\text{Div}}$, via sparsity-regularized divergence minimization[4] under divergence Div, for any probability vector $p$ as:

$$\text{Dec}_{\text{Div}}(p) \in \underset{\hat{p} \in \Delta_V}{\arg\min} \left\{ \text{Div}(\hat{p}, p) + \lambda \|\hat{p}\|_0 \right\} \quad \textbf{(sparsity-regularized decoding)}. \tag{2}$$

---

[3]Formally, for any $p \in \mathbb{R}^V$ and $S \subset [V]$, let $p_S$ be the restriction of $p$ to the coordinates in $S$. Given a vector $p \in \Delta_V$ with $S \subseteq [V]$ indexing its nonzero entries, a renormalization map $T(p)$ can be extended to $\Delta_V$ by embedding it into the original coordinates: $[T(p)]_j = [T(p_S)]_j$ for $j \in S$, and $[T(p)]_j = 0$ otherwise.

[4]In our examples of interest, we will show that this optimization problem is well-defined. When there are multiple minimizers, we assume that one is selected in an arbitrary measurable way.

Here, the $\ell_0$-pseudonorm $\|\hat{p}\|_0$ is the number of nonzero entries of $\hat{p}$, and $\lambda \geqslant 0$ is a *sparsity cost* hyperparameter. As $\lambda$ increases, the optimal solution $\hat{p} = p^*$ gets increasingly more sparse.

**Separable Bregman divergences.** In this work, we shall instantiate Div in Problem 2 with separable Bregman divergences [1, 10]. We will see that this class is expressive enough to induce top-$k$ decoding and many fruitful generalizations of it. For a convex domain $\mathrm{Dom} \subseteq \mathbb{R}$ and a convex differentiable function $\phi : \mathrm{Dom} \to \mathbb{R}$, the one-dimensional Bregman $\phi$-divergence $\mathrm{d}_\phi$ is defined as: $\mathrm{d}_\phi(x, y) = \phi(x) - \phi(y) - \phi'(y)(x - y)$, for $x, y \in \mathrm{Dom}$. The separable $V$-dimensional Bregman $\phi$-divergence $\mathrm{D}_\phi : \mathrm{Dom}^V \to \mathbb{R}$ is then defined as:

$$\mathrm{D}_\phi(x, y) = \sum_{i \in [V]} \mathrm{d}_\phi(x_i, y_i), \quad \text{for } x = (x_1, \ldots, x_V), y = (y_1, \ldots, y_V) \in \mathrm{Dom}^V.$$

A well-known property of Bregman divergences is that $\mathrm{D}_\phi(x, y) \geqslant 0$ for all $x, y$, with equality if $x = y$; when $\phi$ is strictly convex, $x = y$ in fact becomes the unique minimum.

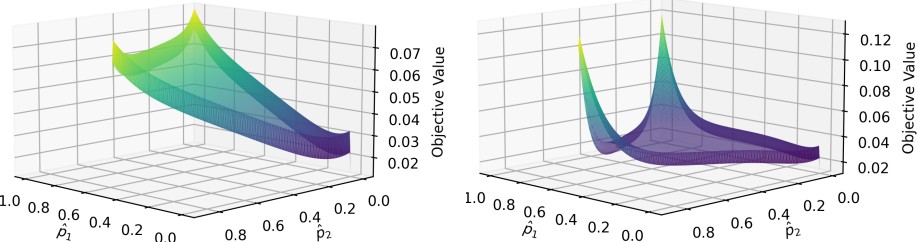

Figure 1: Illustration of the landscape of the sparse Bregman objective for the primal (left) and dual (right) cases. We choose a $V = 3$ dimensional example where the target vector is $p = (0.1, 0.01, 0.001)/0.111$. We show an $\alpha$-Bregman divergence (see Section 4) with $\alpha = 10$ and $\lambda = 0.01$.

**Primal and dual Bregman decoding.** Since Bregman divergences are generally non-symmetric in their arguments, we may instantiate the sparse Bregman decoding Problem 2 in two substantially distinct ways: by placing the estimand $\hat{p}$ in the first (*primal*) or second (*dual*) argument:

$$\mathrm{Div}(\hat{p}, p) := \mathrm{D}_\phi(\hat{p}, p) \quad \textbf{(primal decoding)}, \qquad \mathrm{Div}(\hat{p}, p) := \mathrm{D}_\phi(p, \hat{p}) \quad \textbf{(dual decoding)}. \quad (3)$$

Both formulations possess a sound theoretical motivation. *Bregman projections* are commonly defined as minimization in the first argument, while Bregman-based *proper scoring rules* for mean elicitation correspond to minimization in the second argument [see e.g., 24, 39, etc].

The landscapes of primal and dual decoding are illustrated in Figure 1. The dual objective can be non-convex even in the interior of the simplex. However, crucially, the objectives are discontinuous at the edges of the simplex due to the $\ell_0$ penalty. While in general these decoding objectives could be combinatorial problems that may be hard to solve, we will show in Section 3 that for separable Bregman divergences, both the primal and dual problems can be solved efficiently.

In both the primal and the dual Bregman case, when $\lambda = 0$, the corresponding sparse decoding Problem 2 is solved at $\hat{p} = p$ (and uniquely so if $\phi$ is strictly convex), with the intuition that absent sparsity requirements the best guess is to preserve the original distribution $p$. Henceforth we focus on the sparse regime $\lambda > 0$, which forces some entries of $\hat{p}$ to be zero at optimality. Our main results in Section 3 show for both primal and dual decoding that, under mild conditions on $\mathrm{D}_\phi$, the optimal solution keeps exactly the top-$k^\star$ coordinates of $p$, for an objective-chosen $k^\star = k^\star(p)$. This yields a principled and broad generalization of top-$k$ decoding.

## 3 Efficient computation of primal and dual Bregman decoding

We now investigate the optimization of the sparse objectives that give rise to primal and dual Bregman decoding. Absent further structure in these objectives, for any fixed $k$ one would have to search over all (combinatorially many) size-$k$ sparsity patterns to decide which $k$ probabilities to keep; and one would have to try all $k \in [V]$ to determine the optimal $k$. Fortunately, we will now show that Bregman decoding objectives admit computationally efficient optimization, which rests on two pillar properties: (1) The **greedy property**: Given any $k$, it is optimal to select the top $k$ tokens. (2) $k$**-convexity**: The sparse Bregman objective is (discretely) convex as a function of $k$.

First, in Section 3.1, we deal with the innermost optimization layer: the renormalization of the selected token probabilities (which is performed after the optimal $k$ and the optimal sparsity pattern have been identified). Under certain conditions on the Bregman generator, we show that it reduces to scalar root-finding both in the primal and in the dual case (the dual case in fact necessitates *nested* root finding). We then proceed to show the greedy property in Section 3.2. Finally, for the outermost layer of our optimization problem, we demonstrate the $k$-convexity property in Section 3.3.

## 3.1 Renormalization for a fixed sparsity pattern

We first investigate the renormalization component of a Bregman decoding strategy. Once the optimal sparsity pattern $S \subseteq [V]$ (of some size $|S| = k$) has been identified, the vector $x$ — which denotes the sub-vector of $p$ restricted to indices in $S$ — needs to be projected onto the simplex $\Delta_k$. Since the $\ell_0$ regularization term becomes fixed to $\lambda k$, Problem 2 becomes equivalent to: $\arg\min_{\hat{p} \in \Delta_k} \mathrm{Div}(\hat{p}, x)$. This is a $k$-dimensional Bregman projection problem to the simplex (without sparsity regularization). We will now, for both primal and dual decoding, (i) derive conditions under which this problem is well defined, and (ii) show that it can be efficiently solved by reduction to *scalar root finding*.

**Primal renormalization.** We impose the following mild condition on the Bregman generator $\phi$; compared to a minimal set of assumptions for a Bregman divergence to be well-defined, it additionally requires first-order smoothness and strict convexity to hold on the entirety of the relevant interval.

**Assumption 3.1** (Primal validity). *The map $\phi$ is convex and continuously differentiable on $[0, 1]$ as well as strictly convex on $(0, 1)$.*

Existing results [33, 34] then imply that for a primal valid potential $\phi$, denoting $f = \phi'$ (and extending its inverse $f^{-1}$ so that $f^{-1}(x) = 0$ for $x < f(0)$ and $f^{-1}(x) = 1$ for $x > f(1)$, making it continuous and non-decreasing on all of $\mathbb{R}$), the **primal renormalization** map $T_\phi$ is given for $x \in \Delta_{\mathrm{sub},k}$ by:

$$[T_\phi(x)]_i = f^{-1}(f(x_i) + \nu) \quad \text{for all } i \in [k], \text{ where } \nu \in \mathbb{R} \text{ is chosen so that } \sum_{i=1}^{k} [T_\phi(x)]_i = 1. \quad (4)$$

Since $\nu \mapsto f^{-1}(f(x_i) + \nu)$ is non-decreasing[5] in $\nu$, the solution can be found efficiently using off-the-shelf root-finding algorithms such as Brent's method.

**Dual renormalization.** In contrast to the primal case, dual Bregman projections have (to our knowledge) not been directly studied in prior literature. They also offer new challenges: even their uniqueness cannot be taken for granted due to the general nonconvexity of Bregman divergences in the second argument [3]. To pave the road towards dual Bregman projections, we will therefore rely on additional structure in $\phi$ and $\mathrm{d}_\phi$, expressed as the following dual validity condition.

**Assumption 3.2** (Dual validity). *The map $\phi$ is thrice differentiable on $(0, 1]$ with $\lim_{x \to 0^+} x\phi''(x) = 0$. For $x \in (0, 1], y \mapsto \mathrm{d}_\phi(x, y)$ is strictly convex for $y \in [x, 1]$, and $y \mapsto \mathrm{d}_\phi(0, y)$ is strictly convex for $y \in (0, 1]$.*

We establish in Theorem A.1 (see Appendix A) that subject to dual validity, the **dual renormalization** map $T_\phi^*$ is uniquely defined for any $x \in \Delta_{\mathrm{sub},k}$ with $x \neq 0_k$ by the following implicit equations:

$$[T_\phi^*(x)]_i = x_i + \nu^*/f'([T_\phi^*(x)]_i) \text{ for } i \in [k], \text{ with } \nu^* \in \mathbb{R} \text{ chosen so that } \sum_{i=1}^{k} [T_\phi^*(x)]_i = 1. \quad (5)$$

This transformation is interpretable despite its implicit nature: For every index $i \in [k]$, Equation 5 has the effect of *increasing* the corresponding probability $x_i$ by a positive additive amount regulated by an auxiliary variable $\nu^*$; the latter is chosen to make the increased top-$k$ probabilities sum to 1.

Assumption 3.2, short of requiring global convexity of $\mathrm{d}_\phi(x, \cdot)$ on $[0, 1]$, only enforces it for $y \in [x, 1]$. To enable this relaxation, the proof of Theorem A.1 carefully excludes optimal solutions belonging to the region $y \leq x$ or to the simplex boundary. Rather than a mere curiosity, this refinement substantially expands the scope of dual decoding. In particular, in our later specialization, it is essential for ensuring that dual $\alpha$-decoding is uniquely defined for all $\alpha > 1$, not just $\alpha \in (1, 2]$: as plots in Appendix G.4 demonstrate, $\alpha$-Bregman divergences are nonconvex for $y \leq x$ for $\alpha > 2$.

---

[5]It is strictly increasing for $\nu \in [-f(x_i), 1 - f(x_i)]$, but the required $\nu$ may lie outside this range.

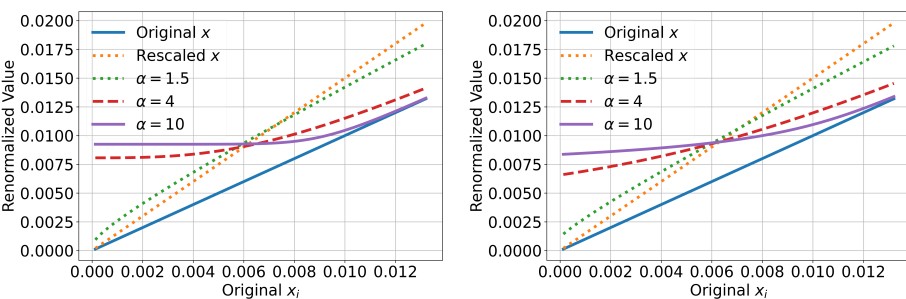

Figure 2: Comparison of primal (left) and dual (right) Bregman $\alpha$-renormalization maps (see Section 4) on input vector $x = \frac{0.67}{\sum_{i=1}^{k} \frac{i}{k}} \left[1, \frac{k-1}{k}, \ldots, \frac{1}{k}\right] \in \Delta_{\text{sub},k}$ with $k = 100$. We plot the renormalized values against the original coordinate values of $x$.

See Section F for algorithmic details on computing the dual map, as well as pseudocode for our algorithms. Figure 2 illustrates the primal and dual renormalization maps for $\alpha$-Bregman divergences (introduced in Section 4). In this concrete example, $T_\phi$ and $T_\phi^*$ appear similar; however, for different, e.g. more "peaked", inputs $x \in \Delta_{\text{sub},k}$, they are more distinct, as we illustrate in Appendix G.3.

## 3.2 Greedy property: Justifying top-$k$ selection

The viewpoint that lower-probability tokens can be considered as noisy [32] suggests that it would be natural and indeed desirable for a decoding strategy to be "greedy"—dictating that it is optimal to renormalize over the top-$k$-probability tokens, for some $k \in [V]$. We formalize this as follows.

**Definition 3.1** (Greedy decoding). *A decoding strategy* $\text{Dec} : \Delta_V \to \Delta_V$ *is called* greedy *if for every* $p \in \Delta_V$, *the set of nonzero entries of* $\text{Dec}(p)$ *is a set of top-$\hat{k}$ entries of $p$, for some $\hat{k} = \hat{k}(p)$.*

While many popular decoding methods are greedy [12, 21, 32, 38], some are not [22, 36]; justifications for non-greediness, i.e., the ability to occasionally throw out some of the top-$k$ tokens, include that this can e.g. help generate more "typical" text. As such, our assertion that Bregman decoding strategies are greedy is nontrivial and requires proof. First, we state our result for primal Bregman decoding.

**Theorem 3.2** (Primal Bregman decoding is greedy). *The primal Bregman decoding strategy from* (2) *is greedy for any primal valid potential $\phi$.*

The proof is provided in Appendix B. It proceeds by decomposing the Bregman objective into several terms, see Lemma B.2, and bounding them with the help of the primal renormalization equations (4).

The dual case, owing i.a. to the implicit form of the dual renormalization formulas (5), is correspondingly more complex to handle. Unlike in Theorem 3.2, our next result requires further conditions, which we state as a menu of two options. The relationship between the extra assumptions is intricate; Assumption (A2) is implied by, but is strictly weaker than, log-convexity of $\phi'$.

**Theorem 3.3** (Dual Bregman decoding is greedy). *The dual Bregman decoding strategy from* (2) *is greedy for any dual-valid $\phi$ with $\phi'(0) = 0$ that further satisfies either of the following conditions:*
*(A1) $\phi'$ is convex;*
*(A2) The maps[6] $u$ defined as $u(x) := x\phi''(x)/\phi'(x)$ for $x \in (0,1]$ and $\phi$ are nondecreasing.*

The proof is provided in Appendix C. In it, we use two different proof techniques for both conditions: For Condition (A1), our proof in Appendix C.1 leverages the decomposition from the primal case along with the change of variables $\mathrm{d}_\phi(x, y) = \mathrm{d}_{\phi^*}(\phi'(y), \phi'(x))$, where $\phi^*$ is the convex conjugate of $\phi$. For Condition (A2), we develop a saddle-point proof approach in Appendix C.2. For that, we perform a sensitivity analysis of both the renormalized values $[T_\phi^*(p)]_i$ and of the per-coordinate Bregman loss terms, relative to hypothetical changes in the dual Lagrange multiplier $\nu^*$ and in the entries $p_i$ of $p$; we carry this out via implicit differentiation of the defining equations (5).

---

[6]In the economics literature, $u(x) = x\phi''(x)/\phi'(x)$ is referred to as the *elasticity* of the function $\phi'$.

### 3.3 $k$-convexity: Speeding up the search for optimal adaptive $k$

We have seen that for fixed $k$, greedily selecting the top $k$ tokens is optimal. However, without further structure, we would still have to search over all $k \in [V]$ to determine the optimal $k^*$, which would be cost-prohibitive for large token vocabularies. Fortunately, as we will see, only *logarithmically* many values of $k$ will need to be tried, as under greedy selection, the primal and dual Bregman decoding objectives both enjoy *discrete convexity* with respect to $k$.

To formally state our result, fix a divergence Div, probability vector $p \in \Delta_V$, and hyperparameter $\lambda$. We denote the regularized cost of selecting the top-$k$ entries of $p$, as a function of $k \in [V]$, by:

$$\mathrm{cost}(k) := \min_{\hat{p} \in \Delta_k} \left\{ \mathrm{Div}\left((\hat{p}, 0_{V-k}), p\right) + \lambda k \right\}. \tag{6}$$

Recall that a function $h : [V] \to \mathbb{R}$ is *discretely convex* if for all $k \in \{2, \dots, V-1\}$, its discrete second derivative $\Delta^2 h(k) := \Delta h(k+1) - \Delta h(k) := \{h(k+1) - h(k)\} - \{h(k) - h(k-1)\} \geqslant 0$.

**Theorem 3.4** (Discrete primal and dual cost convexity). $\mathrm{cost}(\cdot)$ *is discretely convex in* $k \in [V]$ *for:*

    1. $\mathrm{Div}(\hat{p}, p) = \mathrm{D}_\phi(\hat{p}, p)$, *if $\phi$ is primal valid;*      2. $\mathrm{Div}(\hat{p}, p) = \mathrm{D}_\phi(p, \hat{p})$, *if $\phi$ is dual valid.*

Figure 6 in Appendix G.5) illustrates the result of Theorem 3.4 by displaying the $\mathrm{cost}(\cdot)$ functions for primal and dual Bregman $\alpha$-decoding (defined in Section 4 below) for assorted $\alpha$.

**Implications for efficient computation.** As a corollary of Theorem 3.4, an optimal $k^*$ is provably identifiable by searching for $k$ for which $\Delta\mathrm{cost}(k) \leqslant 0$ and $\Delta\mathrm{cost}(k+1) \geqslant 0$, for which repeated bisection (binary search) over $1 \leq k \leq V$ suffices — and thus, only $O(\log V)$ tries of $k$ are necessary. However, even less computation can suffice if one leverages that the optimal $k$ is typically small. First, if one heuristically sets a hard limit $k_\mathrm{u}$ on $k$ (e.g. $k_\mathrm{u} = 50$), then identifying an optimal $k \in [k_\mathrm{u}]$ requires $O(k_\mathrm{u})$ tries. Secondly, one may use exponential search[7] instead of binary search over $k$: this requires only $O(\log k^*)$ tries — very small for typical values of $k^*$ — and has the added benefit that only renormalizations over at most $O(k^*)$ tokens are performed at each step.

**Proving Theorem 3.4.** Our proof uses two distinct approaches for the primal and the dual cases:

*Primal $k$-convexity.* The proof is developed in Appendix D. As its cornerstone, we use the Legendre dual mapping $\phi^*$ of the generator $\phi$ to establish and leverage the following cost structure: for any $k$, $\mathrm{cost}(k)$ can up to additional terms be represented as $\max_{\nu \geq 0} \left[ \nu - \sum_{i=1}^k \phi^*(\phi'(p_i) + \nu) \right]$. This expression is concave in $\nu$, and its unique optimizer is $\nu_k$, the optimal Lagrange multiplier for renormalizing the top $k$ probabilities of $p$ from (4). Using this, we then establish $\Delta^2 \mathrm{cost}(k) \geq 0$.

*Dual $k$-convexity.* The proof is in Appendix E. The above dualization strategy does not directly apply. Instead, we lower bound $\Delta^2 \mathrm{cost}^*(k)$ by regrouping the loss contributions of the indices $i \in [k+1]$, and —via intricate term rearrangement and bounding—reduce to proving the local concavity of a special transformation (Equation 20) that turns out to hold by our dual-validity assumption.

## 4 Example: Bregman $\alpha$-decoding

We now consider, as an illustration, a single-parameter family of Bregman decoding strategies, which arises via the generators of the Havrda–Charvát–Tsallis $\alpha$-entropies [8, 29, 45, 51, 52]:

$$\phi_\alpha(x) = x^\alpha / [\alpha(\alpha-1)], x \in [0,1], \quad \text{for } \alpha \in J := (-\infty, 0) \cup (0,1) \cup (1, \infty).$$

When $\alpha < 0$ and $x = 0$, we set $x^\alpha := +\infty$ so that $\phi_\alpha(0) = \infty$. For $\alpha = 1$, one defines $\phi_1(x) = x \log(x)$, which corresponds to the Shannon entropy, arising in the limit[8] as $\alpha \to 1$. Observe that $\phi_\alpha$ is *primal valid* for all $\alpha \neq 0$, as $\phi_\alpha''(x) = x^{\alpha-2}$. This yields the following primal family of renormalizations, which we will index by $\alpha$ rather than $\phi$:

**Definition 4.1** (Primal Bregman $\alpha$-decoding). *Fix $\alpha \in J, k \in [V]$. The renormalization map $T_\alpha$ is given for $p \in \Delta_{\mathrm{sub},k}$ as:* $[T_\alpha(p)]_i = (p_i^{\alpha-1} + \nu)^{\frac{1}{\alpha-1}}$ *for $i \in [k]$, with $\nu \in \mathbb{R}$ chosen so that* $\sum_{i \in [k]} [T_\alpha(p)]_i = 1$.

---

[7]First, identify a *true* upper bound $k_\mathrm{u}^*$ on $k^*$ by sequentially trying $k_\mathrm{u} = 1, 2, 4, \dots$, and then perform binary search in $O(\log k_\mathrm{u}^*)$ rounds.

[8]One conventionally defines the entropies via $(x^\alpha - x)/[\alpha(\alpha-1)]$, in which case the Shannon entropy is obtained in the limit as $\alpha \to 1$. In our case, we use the definition $\phi_\alpha(x) = x^\alpha/[\alpha(\alpha-1)]$ so that some technical conditions (such as $\phi_\alpha'(0) = 0$) hold in the proofs. Both definitions lead to the same decoding strategies in (4).

Note that for $\alpha = 1$, we have $\phi_1'(x) = \log x + 1$. Hence, (4) implies $e^\nu \sum_{i=1}^k p_i = 1$, and we obtain the "standard" renormalization: $[T_1(p)]_i = p_i/(\sum_{j=1}^k p_j)$, for $i \in [k]$. Therefore, *primal Bregman 1-decoding is top-$k$ decoding*, showing how one recovers top-$k$ in our framework. It turns out that some further values of $\alpha$ also lead to renormalization maps of special interest. For any fixed $p$, we let $T_{-\infty}(p) = \liminf_{\alpha \to -\infty} T_\alpha(p)$ and $T_\infty(p) = \liminf_{\alpha \to \infty} T_\alpha(p)$, where the limits are entrywise.

**Proposition 4.2** (Special primal $\alpha$-renormalization maps)**.** *We have the following special instances[9] of the primal Bregman $\alpha$-renormalization map, defined for all $i \in [k]$ as follows:*

$[T_{-\infty}(p)]_i = p_i + \mathbb{1}[i = i^*] \cdot \left(1 - \sum_{j=1}^k p_j\right)$, *assuming that* $\arg\max_i p_i = \{i^*\}$.

$[T_{1.5}(p)]_i = \left(\sqrt{p_i} + \left[\sqrt{r^2 + k(1-s)} - r\right]/k\right)^2$, *where* $r = \sum_{j=1}^k \sqrt{p_j}$ *and* $s = \sum_{j=1}^k p_j$.

$[T_2(p)]_i = p_i + (1 - \sum_{j=1}^k p_j)/k$.

$[T_\infty(p)]_i = \max\{p_i, \nu\}$, *where* $\nu \in \mathbb{R}$ *is the "water level" for which* $\sum_{i=1}^k [T_\infty(p)]_i = 1$.

Along with the primal family, the dual $\alpha$-decoding family can also be defined based on $\phi_\alpha$. Unlike $\alpha$-decoding, the dual Bregman sparse decoding Problem 2 can be non-convex, as displayed in Figure 1 above. Figure 5 in Appendix G.4 further demonstrates the nonconvexity of $\mathrm{D}_{\phi_\alpha}$ on the unit square for some $\alpha$. Yet, we can still show that any dual $\alpha$-decoding with $\alpha > 1$ is valid, greedy and $k$-convex:

**Lemma 4.3.** *All generator functions $\phi_\alpha$, $\alpha > 1$, are dual-valid and satisfy Assumption (A2).*

We give an illustration contrasting primal and dual $\alpha$-decoding for various $\alpha > 1$ in Appendix G.3.

## 5 Experiments

We now illustrate some of the decoding schemes described in our paper in the context of LLMs. Since our goal is to develop the theoretical foundations of top-$k$ decoding, our aim in this section is simply to illustrate that the performance of our novel decoding schemes can be competitive with standard top-$k$ decoding. In particular, we do not aim to compare or compete with other popular and established decoding methods, which is beyond the scope of our theory-focused paper.

### 5.1 Experimental Setup

**Method.** In addition to standard top-$k$ decoding, which coincides with the $\alpha = 1$ case of our primal $\alpha$-decoding family described in Section 4, we illustrate primal $\alpha$-decoding strategies for $\alpha = 1.5$ and $\alpha = 2$. These have closed-form renormalization maps that are as fast as standard renormalization.

**Full and partial evaluation.** Further, we perform two types of experiments: (1) For the evaluation of our *full* decoding strategy, we decode by adaptively selecting the optimal sparsity parameter $k^*$ by optimizing our sparse Bregman objective. In this approach, we aim to observe the behavior when adaptively choosing $k^*$. Since practical choices of $k^*$ are always upper bounded, we set a maximum $k^* \leqslant k_{\max} := 50$. (2) In the *partial* evaluation approach, we instead directly evaluate—for each fixed choice of $k$ in the grid $k \in \{5, 10, \ldots, 50\}$—our proposed renormalization strategies along with standard top-$k$ renormalization.

**Models and benchmarks.** We conduct experiments using the GPT-2 Large [43] and Llama 3.1 8B [25] models. We evaluate on two benchmarks: (1) open-ended text generation using the WebText test set from the GPT-2 output dataset [40], and (2) grade school math reasoning using the GSM8K Chain-of-Thought benchmark [13]. Additional experiments with larger models, Qwen2.5-14B-Instruct and Phi-3-Medium-4K-Instruct, as well as evaluations on the TriviaQA benchmark, are presented in Appendix H.

**Evaluation metrics.** For open-ended text generation, following Chen et al. [12], we use the first 35 tokens of each WebText test sample as a prompt and generate up to 256 tokens. We evaluate the following standard metrics [see e.g., 12, 32, 38, etc]:

(1) *Perplexity difference*, which measures the perplexity (according to base model $p_{\text{base}}$) of human text compared to that obtained from a decoding strategy $p_{\text{decoding}}$ derived from the base model, where lower is better. This equals $\mathbb{E}_{X \sim \mathcal{D}}[\mathbb{E}_{Y \sim \mathcal{D}(\cdot|\mathcal{X})}(p_{\text{base}}(Y \mid X)^{-1/|Y|}) - \mathbb{E}_{Y \sim p_{\text{decoding}}(\cdot|X)}(p_{\text{base}}(Y \mid X)^{-1/|Y|})]$, where $X \sim \mathcal{D}$ is a

---

[9]In particular, $T_{-\infty}(p), T_{1.5}(p), T_2(p)$ do not require solving for $\nu$ in Definition 4.1, enabling a fast implementation just like in the case of the canonical top-$k$ renormalization.

Table 1: Accuracy on GSM8K for LLaMA 3.1 8B using Bregman primal decoding ($\lambda \in \{0.01, 0.0001\}$, $\alpha \in \{1.5, 2.0\}$) and top-$k$ decoding, for various temperatures. For top-$k$, $k$ equals the averaged $k^*$ from primal decoding with matching temperature, $\lambda$, and $\alpha$. Standard deviations are over 1000 bootstrap resamples.

| Temp | $\lambda = 0.01$ | | Top-$k$ ($\lambda = 0.01$) | $\lambda = 0.0001$ | | Top-$k$ ($\lambda = 0.0001$) |
|---|---|---|---|---|---|---|
| | $\alpha = 1.5$ | $\alpha = 2.0$ | | $\alpha = 1.5$ | $\alpha = 2.0$ | |
| 0.3 | $85.14_{\pm0.80}$ | $84.38_{\pm1.00}$ | $83.62_{\pm1.02}$ $84.69_{\pm0.99}$ | $84.69_{\pm0.99}$ | $84.46_{\pm1.00}$ | $85.14_{\pm0.98}$ $83.62_{\pm1.02}$ |
| 0.7 | $83.24_{\pm1.02}$ | $81.73_{\pm1.06}$ | $83.78_{\pm1.02}$ $84.69_{\pm0.99}$ | $82.03_{\pm1.06}$ | $82.03_{\pm1.06}$ | $82.11_{\pm1.06}$ $83.78_{\pm1.02}$ |
| 1.0 | $81.20_{\pm1.08}$ | $80.97_{\pm1.08}$ | $81.20_{\pm1.08}$ $81.20_{\pm1.08}$ | $77.41_{\pm1.15}$ | $77.26_{\pm1.15}$ | $79.23_{\pm1.12}$ $78.54_{\pm1.13}$ |
| 1.5 | $79.00_{\pm1.12}$ | $80.06_{\pm1.10}$ | $75.97_{\pm1.18}$ $75.97_{\pm1.18}$ | $57.24_{\pm1.36}$ | $64.97_{\pm1.31}$ | $43.21_{\pm1.36}$ $58.53_{\pm1.36}$ |

prompt drawn from the dataset, $Y \sim \mathcal{D}(\cdot|\mathcal{X})$ denotes a human-written continuation drawn from the dataset, and $Y \sim p_{\text{decoding}}(\cdot \mid X)$ denotes a model-generated continuation using a specific decoding strategy. Here, $|Y|$ is the length of the continuation.

(2) *Repetition difference*: $\mathbb{E}_{X \sim \mathcal{D}} \left[ \mathbb{P}_{Y \sim p_{\text{decoding}}(\cdot|X)} (\text{rep}(Y)) - \mathbb{P}_{Y \sim \mathcal{D}(\cdot|X)}(\text{rep}(Y)) \right]$, where $\text{rep}(Y)$ is the event that $Y$ contains two contiguous and identical token spans of length $\geqslant 2$; lower is better.

### 5.2 Results

**Open-ended text generation.** Using the *partial* evaluation setup with temperature fixed at 1.0, Figure 3 reports the differences in perplexity and repetition frequency between model-generated and human-written text across a range of $k$ values. Primal decoding strategies are competitive with top-$k$ in terms of both metrics. In particular, $\alpha = 2.0$ has the smallest gaps in perplexity and repetition frequency. The marked decrease in repetitiveness for $\alpha \in \{1.5, 2\}$ relative to standard top-$k$ ($\alpha = 1$) is consistent with the theoretical behavior of $\alpha$-Bregman renormalization that we have derived. For fixed $k$, increasing $\alpha$ shifts the renormalization from upweighting the largest probabilities to comparatively upweighting the smaller probabilities within the selected top-$k$ set, thereby increasing sampling diversity.

**GSM8K dataset.** Using the *full* decoding strategy, we evaluate the LLaMA 3.1 8B model using 8-shot CoT prompting. We test various temperatures, regularization strengths $\lambda \in \{0.01, 0.0001\}$ and primal decoding parameters $\alpha \in \{1.5, 2.0\}$. Results for other settings are in Appendix H. To ensure a matched comparison, we run top-$k$ with $k = k^*$ for the Bregman decoding run with the same temperature, $\lambda$, and $\alpha$, rounded to the nearest integer, see Table 11 in Appendix H. As seen in Table 1, across all temperature settings, primal decoding with adaptive $k^*$ achieves accuracy comparable to top-$k$. At higher temperatures (such as 1.5), the performance of top-$k$ decoding degrades more rapidly than that of primal decoding.

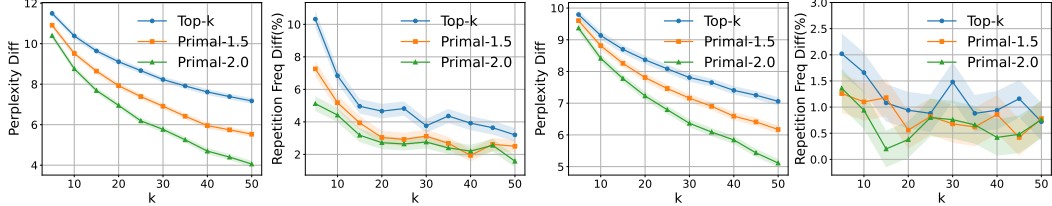

Figure 3: Perplexity and repetition frequency differences between generated and human-written text for GPT2-large (left two panels) and LLaMA 3.1 8B (right two panels), for various $k$ values. We show top-$k$ decoding and primal decoding with $\alpha \in \{1.5, 2.0\}$. Standard deviations are estimated using 1000 bootstrap resamples.

## 6 Related work

**Bregman projection.** Michelot [37] considered the Brier score projection problem and derived an efficient algorithm. Later, Shalev-Shwartz et al. [48] revisited the properties of optimal Brier projection, and Duchi et al. [17] gave and analyzed the explicit algorithm that we discuss in what follows. Wang and Carreira-Perpinán [53] simplified and distilled the proof. [35] further studied the projection as a method for generating sparse probability predictions in multiclass prediction problems. [33, 34] developed methods for efficient Bregman projections to the simplex; for a fixed support, these results characterize our primal decoding. [44, 46] developed differentiable variants of top-$k$ decoding. In contrast to these works, we: (1) consider Bregman projections under $\ell_0$ regularization, and (2) offer, to the best of our knowledge, novel analyses of *dual* Bregman projections.

$\ell_0$ **regularization.** Regularization via the $\ell_0$-pseudonorm has been studied widely, with various approximate algorithms (based on surrogates, integer programming, branch-and-bound methods, etc.) developed for problems ranging from linear regression to more general learning tasks [see e.g., 2, 6, 9, 15, 18–20, 30, 41, 49, 50, 58, 61, etc]. In contrast, the algorithms we propose are exact within numerical precision for the specific class of problems we consider.

**Bregman divergences.** The properties of Bregman divergences [10] have been widely studied; see, e.g., [1, 3, 5, 8, 27, 39, 47, 55, 57], etc. In particular, there are a number of relations between Bregman divergences and their versions with reversed arguments, motivated by the fact that convexity in the first parameter allows for minimization, making it useful to switch the order of the variables, see e.g., [1, 26] etc. We both leverage some of these results in our work, and contribute some, to the best of our knowledge, novel proof techniques and insights into the (primal and dual) Bregman geometry.

**LLM decoding.** There is a vast range of work on LLM sampling (or decoding), see e.g., [54] and references therein. Classical methods include greedy sampling and beam search. Sparse sampling methods such as top-$k$ sampling [21] are motivated by intuition that the "unreliable tail" of low-probability tokens is mis-estimated [32]. In particular, [32] propose top-$p$ sampling, and [38] propose min-$p$ sampling. Other sampling methods were proposed in [4, 22, 31, 36]. [12] propose the decoding game, a two-player game between an LLM and an adversary that distorts the true distribution. They show that certain sparse truncated sampling methods are approximately minimax optimal. For other approaches to make LLM output probabilities sparse, see e.g., [14, 59, 60]. In contrast, our goal is to develop a deep theoretical understanding of top-$k$ decoding, placing it into a broader framework.

**General motivation.** The motivation for our general approach is two-fold: (1) Without sparsity considerations, Bregman divergences closely correspond to proper scoring rules, and are minimized at the true probability distribution, see e.g., [10, 24]. This property is highly desirable in probabilistic forecasting and prediction, incentivizing a forecaster to predict the true distribution in order to minimize their loss. (2) The $\ell_0$-pseudonorm has been widely argued to both be a reasonable measure of sparsity, and to have good properties as a regularizer in certain sparse estimation problems such as sparse regression [see e.g., 7, 16, 23, 28, etc]. Combining these two lines of thought provides the motivation for studying $\ell_0$-regularized Bregman divergence minimization.

## 7  Discussion

This paper develops a theoretical foundation for top-$k$ decoding. We hope that our framework, which rests on the structural pillars of (i) greedy selection and (ii) $k$-convexity, will spur the development of novel theoretically motivated adaptive sparse decoding methods for LLMs and beyond.

*LLM decoding beyond top-$k$.* Our analysis treats per-step decoding as jointly selecting a support size $k$ and applying a renormalization via Bregman projection. A natural extension that future work may address is to develop analogous optimization-theoretic foundations for other popular truncation rules, such as top-$p$ and min-$p$, which also follow the template "truncate a tail, then renormalize".

*Quantifying and controlling adaptivity.* A central feature of our framework is that the parameter $k^\star(p)$ is chosen adaptively with $p$. A useful next step is to characterize how $k^\star(p)$ varies with the hyperparameter $\lambda$ and with properties of the token distribution $p$ (e.g., concentration or tail decay). For $\alpha$-decoding, our preliminary calculations in stylized settings suggest that there may be natural scaling laws relating $k^\star$ to $(\alpha, \lambda)$; an interesting open problem is to rigorously derive such laws under distributional models that reflect LLM next-token probabilities (e.g., peaked or heavy-tailed regimes), and to translate them into practical guidance for targeting a desired average sparsity level.

*Primal vs. dual decoding.* While we established greedy selection and $k$-convexity for both primal and dual formulations, their relationship is not yet well understood. As we illustrate via several examples, primal and dual $\alpha$-decoders are similar yet subtly different for corresponding $\alpha$. Thus, more generally quantifying the primal-dual Bregman decoding gap would be of interest.

*Broader evaluation.* Our initial experiments hint that some performance gains over vanilla top-$k$ might be found even amongst $\alpha$-decoders. It is promising and essential to further understand how Bregman decoders could be optimized towards downstream quality/factuality measures in practice.

*Beyond LLM decoding.* Finally, at a technical level, our main results identify a broad class of nonconvex sparse optimization problems— $\ell_0$-regularized separable Bregman projection onto the simplex—that nevertheless admit exact, efficient solutions via greedy support selection and discrete convexity in the support size. We conjecture that the resulting general-purpose efficient optimization primitive for sparsifying probability vectors could be applicable well beyond LLM decoding.

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

# Appendix Contents

## A  Existence and uniqueness of dual Bregman decoding

**Theorem A.1** (Uniqueness and formula for dual Bregman renormalization)**.** *Fix a dual valid potential $\phi$. Then, for any $x \in \Delta_{\mathrm{sub},k}$ with $\sum_i x_i > 0$, the renormalization map $T_\phi^*$ is uniquely defined by:*

$$[T_\phi^*(x)]_i = x_i + \nu^*/f'([T_\phi^*(x)]_i) \quad \text{for all } i \in [k], \text{ where } \nu^* \in \mathbb{R} \text{ is chosen so that } \sum_{i=1}^{k}[T_\phi^*(x)]_i = 1.$$

*Proof.* First, assume without loss of generality that $0 < \sum_{i \in [k]} x_i < 1$. Otherwise, if $\sum_{i \in [k]} x_i = 1$ then $x \in \Delta_k$, so the unique unconstrained optimum, which is at $x$ by the standard property of Bregman divergences, is also the unique optimum of our constrained projection problem.

Note that Slater's condition is satisfied for this projection problem as we are optimizing over the simplex (whose relative interior is nonempty). Therefore, in this differentiable problem, its optimal solutions can be characterized via its KKT conditions.

Introduce a Lagrange multiplier $\nu \in \mathbb{R}$ for the simplex constraint, and Lagrange multipliers $(\lambda_i)_{i \in [k]}$ for the nonnegativity constraints. Then, the Lagrangian is as follows:

$$\mathcal{L}(\hat{p}, \nu) = \sum_{i=1}^{k} \big[\phi(x_i) - \phi(\hat{p}_i) - \phi'(\hat{p}_i)\,(x_i - \hat{p}_i)\big] - \nu\Big(\sum_{i=1}^{k}\hat{p}_i - 1\Big) - \sum_{i=1}^{k}\lambda_i\hat{p}_i.$$

Here, $\lambda_i \geqslant 0$ for all $i$, and by complementary slackness, at optimality $\lambda_i = 0$ whenever $\hat{p}_i > 0$.

For each $i \in [k]$, the stationarity condition reads (except possibly when $\hat{p}_i = 0$, where the second derivative could be infinite):

$$0 = \frac{\partial \mathcal{L}}{\partial \hat{p}_i} = -\phi''(\hat{p}_i)\,(x_i - \hat{p}_i) - \nu - \lambda_i \iff \phi''(\hat{p}_i)(\hat{p}_i - x_i) = \nu + \lambda_i.$$

In particular, for each coordinate $i$ for which the optimal $\hat{p}_i \in (0, 1)$, the stationarity condition is:

$$\phi''(\hat{p}_i)(\hat{p}_i - x_i) = \nu \implies \hat{p}_i = x_i + \frac{\nu}{\phi''(\hat{p}_i)} = x_i + \frac{\nu}{f'(\hat{p}_i)}. \tag{7}$$

Now, we show that $\nu > 0$. Indeed, observe that there must be at least one index $i$ for which $\hat{p}_i > x_i$. If that was not the case, we would get $\sum_{i \in [k]} \hat{p}_i \leqslant \sum_{i \in [k]} x_i < 1$ by our assumption, contradicting that $\hat{p} \in \Delta_k$. In particular, then, $\hat{p}_i > x_i \geqslant 0$, and therefore we have $\phi''(\hat{p}_i)(\hat{p}_i - x_i) = \nu$. Since $\phi''(\hat{p}_i) > 0$ and $\hat{p}_i - x_i > 0$, we thus conclude that $\nu > 0$.

Having shown that $\nu > 0$, we now proceed to show that all $\hat{p}_i > 0$ at optimality. Note that $\frac{\partial}{\partial y}\mathrm{d}_\phi(x, y) = \phi''(y)\,(y - x)$ for $y > 0$. We will now consider two cases:

1. $\phi''(0)$ is finite;
2. $\lim_{y \to 0} \phi''(y) = +\infty$.

If $\phi''(0)$ is finite, $\hat{p}_i > 0$ for all $i$. Indeed, suppose that was not the case, and $\hat{p}_i = 0$ for some $i$. Then we would have: $\phi''(0)(0 - x_i) = \nu + \lambda_i$, or equivalently, $\phi''(0) \cdot x_i + \nu + \lambda_i = 0$. Each of the three terms is nonnegative, and $\nu > 0$, so we arrive at a contradiction.

Next, consider the case in which $\lim_{y \to 0} \phi''(y) = +\infty$. Then, $\lim_{y \to 0} \frac{\partial}{\partial y} d_\phi(x, y) = -\infty$ for all $x \in (0, 1]$. Then, since $\lim_{y \to 0} \frac{\partial}{\partial y} d_\phi(x, y) = -\infty$ for all $x \in (0, 1]$, for any $i$ such that $x_i > 0$, setting $\hat{p}_i = 0$ would lead to $\nu = -\infty$, hence necessarily $\hat{p}_i > 0$. On the other hand, for any $i$ for which $x_i = 0$, since $\lim_{y \to 0} y \phi''(y) = 0$, setting $\hat{p}_i = 0$ would lead to $\nu = 0$, which is a contradiction.

In all cases, the optimal $\hat{p}$ is in the strict interior of the simplex, so it suffices to solve (7) over this range. To show that the solution exists and is unique, we collect together the following information about $\Psi$ from (13) with $\Psi(x, y, \nu) := \phi''(y)(y - x) - \nu$ for all $x, y, \nu$. Then, for a fixed $\nu$, (7) is equivalent to solving $\Psi(x_i, \hat{p}_i, \nu) = 0$. First, consider $x > 0$. Then, we have the following:

1. Since the map $y \mapsto d_\phi(x, y)$ is strictly convex for $y \in [x, 1]$, it follows that $\frac{\partial}{\partial y} d_\phi(x, y) = \Psi(x, y, 0)$ is strictly increasing for $y \in [x, 1]$, and so is $\Psi(x, y, \nu)$.

2. We have $\Psi(x, x, \nu) = -\nu \leqslant 0$. Further, $\Psi(x, 1, \nu) = \phi''(1)(1 - x) - \nu \geqslant 0$, whenever $\nu \leqslant \phi''(1)(1 - x)$.

Hence, the map $y \mapsto \Psi(x, y, \nu)$ has a unique zero on the interval $[x, 1]$, as long as $0 < \nu \leqslant \phi''(1)(1 - x)$.

Next, consider $x = 0$, in which case we need to solve the equation $\phi''(y)y = \nu$. Then, we have the following:

1. Since the map $y \mapsto d_\phi(0, y)$ is strictly convex for $y \in (0, 1]$, it follows that $\frac{\partial}{\partial y} d_\phi(0, y) = \Psi(0, y, 0) = \phi''(y)y$ is strictly increasing for $y \in (0, 1]$, and so is $\Psi(0, y, \nu)$.

2. By assumption, $\lim_{y \to 0^+} y \phi''(y) = 0$, hence we have $\lim_{y \to 0^+} \Psi(x, x, \nu) = -\nu \leqslant 0$. Further, $\Psi(0, 1, \nu) = \phi''(1)(1 - x) - \nu \geqslant 0$, whenever $\nu \leqslant \phi''(1)$.

Hence, the map $y \mapsto \Psi(0, y, \nu)$ has a unique zero on the interval $(0, 1]$, as long as $0 < \nu \leqslant \phi''(1)$.

Now define $M := \min_i \phi''(1)(1 - x_i) = \phi''(1)(1 - \max_i x_i)$. Since by assumption $\sum_i x_i < 1$, it follows that $M > 0$. From the above analysis, it follows that, as long as $\nu \in (0, M]$, for each $i$, the equation $\phi''(y_i)(y_i - x_i) = \nu$. has a unique solution $y_i(\nu) \in (x_i, 1]$.

Furthermore, as we establish in Lemma C.2, the map $\nu \mapsto y_i(\nu)$ is strictly increasing for $\nu > 0$, also owing to the assumed second-argument convexity of $d_\phi$. In particular, define $G(\nu) = \sum_{i=1}^k y_i(\nu)$ for $\nu > 0$; then $G$ is continuous and strictly increasing, and satisfies $\lim_{\nu \to 0} G(\nu) = \sum_i x_i < 1$ and $G(M) \geqslant y_{i^*}(M) = 1$, where $i^*$ is any index achieving the maximum among the coordinates of $x$. Hence there is a unique $\nu^* \in (0, M]$ with $G(\nu^*) = 1$. Setting $\hat{p}_i = y_i(\nu^*)$ yields a vector in $\Delta_k$ that satisfies the KKT stationarity.

Finally, note that the solution $\hat{p}$ that we just identified is unique. Indeed, we have earlier excluded boundary solutions from consideration, and then further excluded any solutions in which $\hat{p}_i < x_i$ for any $i \in [k]$; thus, it suffices to recall that the Bregman objective is assumed to be strictly convex in the interior of the region of the simplex given by $\{\hat{p} \in \Delta_k : \hat{p}_i \geqslant x_i \text{ for all } i \in [k]\}$, thus concluding the proof. $\qquad \square$

## B   Proof of the primal greedy property in Theorem 3.2

We will first fix some notations. Henceforth, we will assume that the vector $p$ has been sorted, i.e., $p_1 \geqslant p_2 \geqslant \ldots \geqslant p_V$. For any subset $Q = \{i_1, \ldots, i_k\} \subseteq [V]$ of size $k$, let $Q^c = [V] \backslash Q$. Let $p_Q$

denote the sub-probability vector with the entries of $p$ whose indices are in $Q$. We define the loss $L(Q)$ as

$$L(Q) = \min_{\hat{p} \in \Delta_k} \mathrm{D}_\phi((\hat{p}, 0_{V-k}), (p_Q, p_{Q^c})) = \min_{\hat{p} \in \Delta_k} \sum_{j=1}^{k} \mathrm{d}_\phi(\hat{p}_j, p_{i_j}) + S_{Q^c}. \tag{8}$$

Here, $S_{Q^c} = \sum_{j \notin Q} \mathrm{d}_\phi(0, p_j)$. To prove Theorem 3.2, we will show that $L(S') \geqslant L(S)$ for any $S' \subseteq [V]$ of size $k$, where $S = [k]$ consists of the top-$k$ indices. We will further show that strict inequality always holds if $p_{S'} \neq p_S$. To do this, we proceed in three steps: (1) We first simplify the form of the loss function $L(Q)$ in Lemma B.1, (2) For any two subsets $S, S'$, we decompose the loss difference $L(S') - L(S)$ into three terms in Lemma B.2, (3) We individually analyze each of the terms in this decomposition and prove they are non-negative.

## B.1 Decomposing the Bregman cost function on subsets

**Lemma B.1.** *For any $Q = \{i_1, i_2, \ldots i_k\} \subseteq [V]$ of size $k$, the loss function as defined in (8) simplifies to:*

$$L(Q) = \sum_{j=1}^{k} [\phi([T_Q(p)]_j) - \phi'(p_{i_j})[T_Q(p)]_j] + S_{[V]} - |Q|\phi(0). \tag{9}$$

*Proof.* Observe that:

$$L(Q) = \mathrm{D}_\phi((\hat{p}_Q, 0_{V-k}), (p_Q, p_{Q^c})) = \sum_{j=1}^{k} \mathrm{d}([T_Q(p)]_j, p_{i_j}) + S_{Q^c}$$

$$= \sum_{j=1}^{k} [\phi([T_Q(p)]_j) - \phi(p_{i_j}) - \phi'(p_{i_j})([T_Q(p)]_j - p_{i_j})] + S_{Q^c}$$

$$= \sum_{j=1}^{k} [\phi([T_Q(p)]_j) - \phi'(p_{i_j})[T_Q(p)]_j] + \sum_{j=1}^{k} [-\phi(p_{i_j}) + f(p_{i_j})p_{i_j}] + S_{Q^c}.$$

This further equals

$$\sum_{j=1}^{k} [\phi([T_Q(p)]_j) - \phi'(p_{i_j})[T_Q(p)]_j] + \sum_{j \in Q} \mathrm{d}_\phi(0, p_j) + S_{Q^c} - |Q|\phi(0)$$

$$= \sum_{j=1}^{k} [\phi([T_Q(p)]_j) - \phi'(p_{i_j})[T_Q(p)]_j] + S_Q + S_{Q^c} - |Q|\phi(0)$$

$$= \sum_{j=1}^{k} [\phi([T_Q(p)]_j) - \phi'(p_{i_j})[T_Q(p)]_j] + S_{[V]} - |Q|\phi(0).$$

This finishes the proof. $\square$

Let $T_Q(p)$ denote a minimizer of the above loss $L(Q)$, i.e.,

$$T_Q(p) \in \arg\min_{\hat{p} \in \Delta_k} \mathrm{D}_\phi((\hat{p}, 0_{V-k}), (p_Q, p_{Q^c})) \overset{(a)}{=} \arg\min_{\hat{p} \in \Delta_k} \sum_{j=1}^{k} \mathrm{d}_\phi(\hat{p}_j, p_{i_j}).$$

Note that $(a)$ holds above as the term $S_{Q^c}$ does not play any role in the location of the minimizer. However, it does contribute to the final loss $L(Q)$. Also, as the divergence is separable, once we have selected a subset $Q$, the ordering of its elements does not matter for the calculation of the above loss and minimizer. Thus, without loss of generality, we may assume $i_1 < i_2 < \ldots < i_k$ for $k \in [V]$. By forming the Lagrangian and differentiating it, we obtain the primal thresholding from (4):

$$\phi'([T_Q(p)]_j) = \phi'(p_{i_j}) + \nu_Q \; \forall \, j \in [k]. \tag{10}$$

Here, $\nu_Q$ is chosen such that $\sum_{j=1}^{k} [T_Q(p)]_j = 1$.

**Lemma B.2.** *Let* $S = \{i_1, \ldots, i_k\}, S' = \{i'_1, \ldots, i'_k\} \subseteq [V]$ *and* $T_S(p)$ *and* $T_{S'}(p)$ *be the corresponding minimizers. Then, the following decomposition holds:*

$$L(S') - L(S) = D_\phi(T_{S'}(p), T_S(p)) + \sum_{j=1}^{k} ([T_{S'}(p)]_j - [T_S(p)]_j) \left( \phi'([T_S(p)]_j) - \phi'(p_{i_j}) \right)$$

$$+ \sum_{j=1}^{k} [T_{S'}(p)]_j \left( \phi'(p_{i_j}) - \phi'(p_{i'_j}) \right). \tag{11}$$

*Proof.* We have from Lemma B.1 that

$$L(S') - L(S) = \sum_{j=1}^{k} [\phi([T_{S'}(p)]_j) - \phi'(p_{i'_j})[T_{S'}(p)]_j] - \sum_{j=1}^{k} [\phi([T_S(p)]_j) - \phi'(p_{i_j})[T_S(p)]_j]$$

$$= \sum_{j=1}^{k} [\phi([T_{S'}(p)]_j) - \phi([T_S(p)]_j)] + \phi'(p_{i_j})[T_S(p)]_j - \phi'(p_{i'_j})[T_{S'}(p)]_j.$$

This further equals

$$\sum_{j=1}^{k} [\phi([T_{S'}(p)]_j) - \phi([T_S(p)]_j) - \phi'([T_S(p)]_j)([T_{S'}(p)]_j - [T_S(p)]_j)]$$

$$+ \sum_{j=1}^{k} \left( [T_{S'}(p)]_j \left[ \phi'([T_S(p)]_j) - \phi'(p_{i'_j}) \right] - [T_S(p)]_j \left[ \phi'([T_S(p)]_j) - \phi'(p_{i_j}) \right] \right)$$

$$= D_\phi(T_{S'}(p), T_S(p)) + \sum_{j=1}^{k} ([T_{S'}(p)]_j - [T_S(p)]_j) \left( \phi'([T_S(p)]_j) - \phi'(p_{i_j}) \right)$$

$$+ \sum_{j=1}^{k} [T_{S'}(p)]_j \left( \phi'(p_{i_j}) - \phi'(p_{i'_j}) \right).$$

$\square$

Now, returning to our proof, suppose $S = [k]$ and $S' = \{i'_1, \ldots i'_k\}$. We know from Lemma B.2 that

$$L(S') - L(S) = \underbrace{D_\phi(T_{S'}(p), T_S(p))}_{\mathbf{I}} + \underbrace{\sum_{j=1}^{k} ([T_{S'}(p)]_j - [T_S(p)]_j) \left( \phi'([T_S(p)]_j) - \phi'(p_{i_j}) \right)}_{\mathbf{II}}$$

$$+ \underbrace{\sum_{j=1}^{k} [T_{S'}(p)]_j \left( \phi'(p_{i_j}) - \phi'(p_{i'_j}) \right)}_{\mathbf{III}}.$$

Now, consider the term **II**. Using (10), we can simplify this further as follows:

$$\mathbf{II} = \sum_{j=1}^{k} ([T_{S'}(p)]_j - [T_S(p)]_j) \nu_S = \nu_S \left( \sum_{j=1}^{k} [T_{S'}(p)]_j - \sum_{j=1}^{k} [T_S(p)]_j \right) \overset{(a)}{=} 0,$$

where $(a)$ follows as $\sum_{j=1}^{k} [T_{S'}(p)]_j = \sum_{j=1}^{k} [T_S(p)]_j = 1$. Also, $\mathbf{I} \geqslant 0$ as $D_\phi$ is a divergence measure.

Finally, to conclude our proof, we show that $\mathbf{III} \geqslant 0$. Since the entries of $p$ are sorted in a non-decreasing order and as the indices in $S = [k]$ and $S'$ are sorted in ascending order, we have

$$\forall j \in [k], \ j = i_j \leqslant i'_j \Rightarrow \forall j \in [k], \ p(i_j) \geqslant p(i'_j)$$

$$\Rightarrow \sum_{j=1}^{k} [T_{S'}(p)]_j \left( \phi'(p_{i_j}) - \phi'(p_{i'_j}) \right) = \mathbf{III} \geqslant 0.$$

Strict inequality holds as long as some $p_{i'_j}$ is not among the top-$k$ indices of $p$.

# C   Proof of the dual greedy property in Theorem 3.3

To prove the greedy property for the two alternate conditions in Theorem 3.3, we will provide two distinct proof techniques for the two cases (A1) and (A2). The first one uses duality and the second one uses a saddle point argument. We will now recall the definition of the Legendre dual of a convex function—in this case, of the generator function $\phi$—and its defining property that will help us. Below, $f([0,1])$ denotes the image of $[0,1]$ under $f$.

**Lemma C.1** (Classical). *For a valid $\phi$, let $\phi^*(x) = \sup_{p \geq 0}\{px - \phi(p)\}$ be the Legendre dual of $\phi$, defined for all $x \in f([0,1])$. Then, we have for every $x \in f([0,1])$ the identity: $\phi(f^{-1}(x)) = xf^{-1}(x) - \phi^*(x)$. Moreover $(\phi^*)' = f^{-1}$, and $\phi^*$ is strictly increasing.*

*Proof.* Since the map $p \mapsto R(p) := px - \phi(p)$ is continuous, it achieves a maximum on $[0,1]$. From the first order condition of the defining equation for $\phi^*$, if the maximum is achieved in $(0,1)$, we have:

$$\frac{\partial R}{\partial p} = x - \phi'(p) = x - f(p) = 0,$$

so for the maximizer $p_{\max}$ we have $f(p_{\max}) = x \Rightarrow p_{\max} = f^{-1}(x)$. Now, since $f$ is increasing and $x \in f([0,1])$, we have $R'(0) = x - f(0) \geq 0$, with equality if $x = f(0)$. Similarly, $R'(1) = x - f(1) \leq 0$, with equality if $x = f(1)$. Hence, it follows that the above characterization for the maximizer $p_{\max}$ also applies on the boundaries of $[0,1]$. To conclude the proof of the identity, it suffices to observe that $\phi^*(x) = p_{\max}x - \phi(p_{\max}) = xf^{-1}(x) - \phi(f^{-1}(x))$. The expression for $(\phi^*)'$ follows by direct calculation. $\square$

## C.1   Proof under Assumption (A1)

With the dual convex conjugate $\phi^*$ as per Lemma C.1, the divergence measure satisfies:

$$d_\phi(p,q) = d_{\phi^*}(\phi'(q), \phi'(p)). \tag{12}$$

Let the loss for the dual problem be denoted as $L^*$, (the divergence measure with the arguments swapped), and let $T_Q^*$ be the dual renormalization map from Lemma A.1 applied to $p_Q$, i.e.,

$$L^*(Q) = \min_{\hat{p} \in \Delta_k} D_\phi((p_Q, p_{Q^c}), (\hat{p}, 0_{V-k})) = \min_{\hat{p} \in \Delta_k} \sum_{j=1}^{k} d_\phi(p_{i_j}, \hat{p}_j) + S_{Q^c}^*, \text{ where } S_{Q^c}^* = \sum_{j \notin Q} d_\phi(p_j, 0)$$

$$= \sum_{j=1}^{k} d_\phi(p_{i_j}, [T_Q^*(p)]_j) + S_{Q^c}^*.$$

### C.1.1   Decomposition of the loss difference

Using the form of the loss difference in Lemma (B.2) and (12), we can compute the loss difference for the dual problem as follows:

$$L^*(S') - L^*(S) = \sum_{j=1}^{V} d_\phi(p_{i'_j}, [T_{S'}^*(p)]_j) - \sum_{j=1}^{V} d_\phi(p_{i_j}, [T_S^*(p)]_j)$$

$$\stackrel{\text{(due to (12))}}{=} \sum_{i=1}^{V} d_{\phi^*}(\phi'([T_{S'}^*(p)]_j), \phi'(p_{i'_j})) - \sum_{i=1}^{V} d_{\phi^*}(\phi'([T_S^*(p)]_j), \phi'(p_{i_j}))$$

Indeed, changing the potential $\phi$ to $\phi^*$, and changing all the arguments $p_{i_j}, p_{i'_j}, T_S^*, T_{S'}^*$ to $\phi'(p_{i_j}), \phi'(p_{i'_j}), \phi'(T_S^*), \phi'(T_{S'}^*)$ respectively in Lemma (B.2) suffices. Thus, under the same setup

of the two subsets $S = [k]$ and $S'$ and denoting $\phi' = f$, we obtain:

$$L^*(S') - L^*(S) = \mathrm{D}_{\phi^*}\left(f(T^*_{S'}(p)), f(T^*_S(p))\right)$$

$$+ \sum_{j=1}^k \left(f([T^*_{S'}(p)]_j) - f([T^*_S(p)]_j)\right)\left((\phi^*)'\left(f([T^*_S(p)]_j)\right) - (\phi^*)'\left(f(p_{i_j})\right)\right)$$

$$+ \sum_{j=1}^k f\left([T^*_{S'}(p)]_j\right)\left((\phi^*)'(f(p_{i_j})) - (\phi^*)'(f(p_{i'_j}))\right).$$

Since $(\phi^*)' = f^{-1}$, this further equals

$$\underbrace{\mathrm{Div}_{\phi^*}\left(f(T^*_{S'}(p)), f(T^*_S(p))\right)}_{\mathbf{I}'}$$

$$+ \underbrace{\sum_{j=1}^k \left(f([T^*_{S'}(p)]_j) - f([T^*_S(p)]_j)\right)\left([T^*_S(p)]_j - p_{i_j}\right) + \sum_{j=1}^k f\left([T^*_{S'}(p)]_j\right)\left(p_{i_j} - p_{i'_j}\right)}_{\mathbf{II}' \qquad\qquad\qquad\qquad\qquad\qquad\qquad\qquad\qquad\qquad \mathbf{III}'}.$$

### C.1.2 Analysis of terms based on the dual solution

Similar to the proof for the primal case, the term $\mathbf{I}' \geqslant 0$, as $\mathrm{D}_{\phi^*}$ is a divergence, and $\mathbf{III}' \geqslant 0$ as $\phi' = f \geqslant 0$, as $f(0) = 0$ and $f$ is increasing. Moreover, as $f$ is strictly increasing, if any of the $p_{i'_j}$ are not among the top-$k$ entries, then strict inequality holds.

To analyze $\mathbf{II}$, we have

$$\mathbf{II} = \sum_{j=1}^k \left(f([T^*_{S'}(p)]_j) - f([T^*_S(p)]_j)\right)\left([T^*_S(p)]_j - p_{i_j}\right)$$

$$\overset{\text{from Lemma A.1}}{=} \sum_{j=1}^k \left(f([T^*_{S'}(p)]_j) - f([T^*_S(p)]_j)\right)\frac{\nu^*_S}{f'\left([T^*_S(p)]_j\right)}.$$

Since $f$ is convex,

$$\left(f([T^*_{S'}(p)]_j) - f([T^*_S(p)]_j)\right) \geqslant f'\left([T^*_S(p)]_j\right)\left([T^*_{S'}(p)]_j - [T^*_S(p)]_j\right)$$

$$\overset{(a)}{\Rightarrow} \frac{1}{f'\left([T^*_S(p)]_j\right)} \cdot \left(f([T^*_{S'}(p)]_j) - f([T^*_S(p)]_j)\right) \geqslant [T^*_{S'}(p)]_j - [T^*_S(p)]_j$$

$$\overset{(b)}{\Rightarrow} \sum_{j=1}^k \frac{1}{f'\left([T^*_S(p)]_j\right)} \cdot \left(f([T^*_{S'}(p)]_j) - f([T^*_S(p)]_j)\right) \geqslant \sum_{j=1}^k \left([T^*_{S'}(p)]_j - [T^*_S(p)]_j\right) = 0.$$

In the above steps, $(a)$ follows as $f' > 0$ as $f$ is strictly increasing and $(b)$ follows as $\sum_{j=1}^k [T^*_{S'}(p)]_j = \sum_{j=1}^k [T^*_S(p)]_j = 1$. This implies $\mathbf{II}' \geqslant 0$, finishing the proof.

## C.2 Proof under Assumption (A2)

### C.2.1 Extra notation

Since $\frac{\partial}{\partial y}\mathrm{d}_\phi(x,y) = \phi''(y)(y-x)$ for $y > 0$, we define for $(x,y,\nu) \in D := [0,1] \times (0,1] \times (0,\infty)$,

$$\Psi(x,y,\nu) := \phi''(y)(y-x) - \nu. \tag{13}$$

Define the mapping derived from solving $\Psi(x,y,\nu) = 0$ over $y$ by:

$$\xi(x,\nu) : [0,1] \times (0,\infty) \to (0,1], \text{ such that } [T(p)]_i = \xi(p_i, \nu) \text{ for all } i, \text{ and for optimal } \nu.$$

It follows from the proof of Lemma A.1 that the solution $\xi$ is well-defined. Define two auxiliary functions $\psi, h$ that will be used in the computation of the Bregman costs below, such that for all $(x,y,\nu) \in D$:

$$\psi(x,y) := \phi(y) - \phi'(y)(y-x), \quad \text{and } h(x,\nu) := \psi(x, \xi(x,\nu)).$$

### C.2.2 Properties of the auxiliary functions

**Lemma C.2** (Derivatives $\frac{\partial \xi}{\partial x}$, $\frac{\partial \xi}{\partial \nu}$). *Define* $v : [0,1] \times (0,1] \to [0,\infty)$ *as* $v(x,y) = \phi''(y) + \phi'''(y)(y-x)$. *We have for all* $(x,\nu) \in [0,1] \times (0,\infty)$:

$$\frac{\partial \xi}{\partial \nu}(x,\nu) = \frac{1}{v(x,\xi(x,\nu))}, \quad \text{and} \quad \frac{\partial \xi}{\partial x}(x,\nu) = \frac{\phi''(\xi(x,\nu))}{v(x,\xi(x,\nu))}. \tag{14}$$

*Proof.* The proof of either identity follows by applying implicit differentiation to the function $\Psi$. Fix $x \in [0,1]$ and consider

$$F(y,\nu) = \Psi(x,y,\nu) = \phi''(y)(y-x) - \nu \quad \text{for } (y,\nu) \in (0,1] \times (0,\infty).$$

Because $\phi$ is $\mathcal{C}^3$ on $(0,1]$, $F$ is continuously differentiable, and

$$\frac{\partial F}{\partial y}(y,\nu) = \phi'''(y)(y-x) + \phi''(y) = v(x,y) > 0$$

by Assumption 3.2. Hence, by the implicit function theorem, the map $\nu \mapsto \xi(x,\nu)$ is $\mathcal{C}^1$ with

$$\frac{\partial \xi}{\partial \nu}(x,\nu) = -\frac{\partial F/\partial \nu}{\partial F/\partial y} = \frac{1}{v(x,\xi(x,\nu))}.$$

For the latter identity, fix $\nu > 0$ and define

$$G(x,y) := \Psi(x,y,\nu) = \phi''(y)(y-x) - \nu, \qquad (x,y) \in [0,1] \times (0,1].$$

For each $x_0 \in (0,1]$ let $y_0 := \xi(x_0,\nu) \in (0,1]$ satisfy $G(x_0,y_0) = 0$. We have $\frac{\partial G}{\partial y}(x,y) = v(x,y)$. Assumption 3.2 gives $v(x,y) > 0$ for all $0 < y \leqslant 1$ and $0 \leqslant x \leqslant y$. Hence $\partial G/\partial y(x_0,y_0) \neq 0$.

Since $G$ is continuously differentiable and $\partial G/\partial y \neq 0$ at $(x_0,y_0)$, the implicit-function theorem guarantees a $\mathcal{C}^1$ map $x \mapsto \xi(x,\nu)$ in a neighborhood of $x_0$ with $G(x,\xi(x,\nu)) = 0$.

Differentiating $G(x,\xi(x,\nu)) \equiv 0$ with respect to $x$ and using $\partial G/\partial x = -\phi''(y)$ gives

$$0 = \frac{\partial G}{\partial x} + \frac{\partial G}{\partial y}\frac{\partial \xi}{\partial x} = -\phi''(\xi(x,\nu)) + v(x,\xi(x,\nu))\frac{\partial \xi}{\partial x},$$

so

$$\frac{\partial \xi}{\partial x}(x,\nu) = \frac{\phi''(\xi(x,\nu))}{v(x,\xi(x,\nu))}.$$

When $x = 0$, the same argument applies, because $\frac{\partial G}{\partial y}(0,y) = v(0,y) > 0$ and $\partial G/\partial x|_{(0,y)} = -\phi''(y)$ is finite (the solution $y = \xi(0,\nu)$ is strictly positive, so $\phi''(y)$ is finite even if $\phi''(y) \to \infty$ as $y \downarrow 0$). Thus $\partial \xi/\partial x|_{(0,\nu)}$ exists and the same formula holds. This completes the proof. $\square$

**Lemma C.3** (Derivative $\frac{\partial h}{\partial \nu}$). *Under the condition that* $x \mapsto u(x) := x\phi''(x)/\phi'(x)$ *is non-decreasing from Assumption (A2), we have* $\frac{\partial h}{\partial \nu}(x,\nu) \leqslant 0$ *for all* $x \in [0,1]$ *and* $\nu > 0$.

*Proof.* For the derivative with respect to $\nu$, observe first that

$$\frac{\partial \psi}{\partial y}(x,y) = \phi'(y) - [\phi''(y)\,y + \phi'(y)] + x\,\phi''(y) = \phi''(y)\,(x-y).$$

Hence, by the chain rule,

$$\frac{\partial}{\partial \nu}\psi(x,\xi(x,\nu)) = \frac{\partial \psi}{\partial y}(x,\xi(x,\nu))\frac{\partial \xi}{\partial \nu}(x,\nu) = \phi''(\xi(x,\nu))\,[x - \xi(x,\nu)]\frac{\partial \xi}{\partial \nu}(x,\nu).$$

Due to the defining equation $\phi''(\xi)\,(\xi - x) = \nu$, this simplifies to

$$\frac{\partial h}{\partial \nu}(x,\nu) = \frac{\partial}{\partial \nu}\psi(x,\xi(x,\nu)) = -\nu\frac{\partial \xi}{\partial \nu}(x,\nu) = -\frac{\nu}{v(x,\xi(x,\nu))} \leqslant 0,$$

where the last equality uses $\frac{\partial \xi}{\partial \nu}(x,\nu) = \frac{1}{v(x,\xi(x,\nu))}$ and $\nu > 0$. $\square$

**Lemma C.4** (Derivative $\frac{\partial h}{\partial x}$). *Assumption (A2) implies* $\frac{\partial h}{\partial x}(x, \nu) \geqslant 0$ *for all* $x \in [0, 1]$ *and* $\nu > 0$.

*Proof.* First recall that

$$\psi(x, y) = \phi(y) - \phi'(y)\,(y - x) \quad \Longrightarrow \quad \frac{\partial \psi}{\partial x}(x, y) = \phi'(y), \qquad \frac{\partial \psi}{\partial y}(x, y) = \phi''(y)\,(x - y).$$

Hence, with $y = \xi(x, \nu)$,

$$\frac{\partial h}{\partial x}(x, \nu) = \frac{\partial \psi}{\partial x}(x, \xi) + \frac{\partial \psi}{\partial y}(x, \xi)\,\frac{\partial \xi}{\partial x}(x, \nu) = \phi'(\xi) + \phi''(\xi)\,[x - \xi]\,\frac{\partial \xi}{\partial x}(x, \nu).$$

Because $\xi = \xi(x, \nu)$ satisfies $\phi''(\xi)\,(\xi - x) = \nu$, we have

$$\frac{\partial h}{\partial x}(x, \nu) = \phi'(\xi) - \nu\,\frac{\partial \xi}{\partial x}(x, \nu) = \phi'(\xi) - \nu\,\frac{\phi''(\xi)}{v(x, \xi)}.$$

Write

$$N(x, \nu) = \phi'(\xi)\,\phi''(\xi) + (\xi - x)\left[\phi'(\xi)\,\phi'''(\xi) - \phi''(\xi)^2\right] = \phi'(\xi)\,\phi''(\xi) + (\xi - x)\,A(\xi),$$

where $A(t) := \phi'(t)\phi'''(t) - \phi''(t)^2$.

Case 1: $A(\xi) \geqslant 0$. Because $\xi \geqslant x$ from Lemma A.1, the second term is non-negative; with $\phi', \phi'' \geqslant 0$ the first term is also non-negative, so $N \geqslant 0$.

Case 2: $A(\xi) < 0$. Since $\xi \geqslant x$, we have

$$N(x, \nu) \geqslant \phi'(\xi)\phi''(\xi) + \xi\,A(\xi) = \phi'(\xi)^2\,u'(\xi),$$

where $u(t) := t\,\phi''(t)/\phi'(t)$. Indeed,

$$u'(t)\,\phi'(t)^2 = \phi'(t)\left[\phi''(t) + t\,\phi'''(t)\right] - t\,\phi''(t)^2 = \phi'(t)\phi''(t) + t\left[\phi'(t)\phi'''(t) - \phi''(t)^2\right].$$

By Assumption (A2), $u$ is non-decreasing, so $u'(\xi) \geqslant 0$; hence $N(x, \nu) \geqslant 0$ in this case as well.

Because $v(x, \xi) > 0$ and $N(x, \nu) \geqslant 0$ in both cases, we conclude $\partial h(x, \nu)/\partial x \geqslant 0$ for all $x \in [0, 1]$ and $\nu > 0$, thereby proving the lemma. $\qquad\square$

### C.2.3   Proving the dual greedy property

Denote an arbitrary subset of the indices by: $S \subseteq [J]$. Let $\nu_S$ be the corresponding Lagrange multiplier. Below, for a vector $x \in \mathbb{R}^V$ and a set $S \subset [V]$, we denote by $x[S]$ the sub-vector of $x$ restricted to the coordinates in $S$. Since $\phi'(0) = 0$ by the assumptions of Theorem 3.3, denoting $\Gamma = \sum_{m=1}^{J} d_\phi(p_m, 0) + \phi(0)|S|$ we can write for every $S$:

$$D_\phi(p, \hat{p}[S]) = \sum_{m \in S} \phi(p_m) - \phi([T(p)]_m) - \phi'([T(p)]_m) \cdot (p_m - [T(p)]_m) + \sum_{m \in [J] \setminus S} d_\phi(p_m, 0)$$

$$= \sum_{m \in S} -\left(\phi([T(p)]_m) - \phi'([T(p)]_m) \cdot ([T(p)]_m - p_m)\right) + \Gamma$$

$$= \sum_{m \in S} -\psi(p_m, [T(p)]_m) + \Gamma = \sum_{m \in S} -h(p_m, \nu_S) + \Gamma.$$

Now, let us prove that the greedy property holds. Suppose $S$ is optimal among all subsets of indices of size $k$ but does not consist of some of the top $k$ probability tokens. Then there exist some $i \neq j$ such that $i \in S$, $j \notin S$, and $p_j > p_i$. Denote $S' = S \setminus \{i\} \cup \{j\}$.

Let $\nu_S, \nu_{S'}$ denote the choice of $\nu$ that makes the projected probabilities sum to unity. Now since $S'$ only differs from $S$ in that it includes the larger $p_j > p_i$, we can conclude that $\nu_S > \nu_{S'}$.

Then, using the above formula for the value of the objective function on an arbitrary subset, we have:

$$D_\phi(p, \hat{p}[S]) - D_\phi(p, \hat{p}[S']) = h(p_j, \nu_{S'}) - h(p_i, \nu_S) + \sum_{m \in S \setminus \{i\}} \left(h(p_m, \nu_{S'}) - h(p_m, \nu_S)\right).$$

Now, since $h$ decreases in $\nu$ by Lemma C.3, we have that the sum is nonnegative since $\nu_{S'} < \nu_S$. As for the remaining term, we have:

$$h(p_j, \nu_{S'}) \geqslant h(p_j, \nu_S) \geqslant h(p_i, \nu_S),$$

where the first inequality is by the fact that $\nu_{S'} < \nu_S$ and Lemma C.3, and the second inequality is by the fact that $p_j > p_i$ and Lemma C.4. This concludes the proof of the dual greedy property under Assumption (A2).

## D  Proof of discrete convexity for primal Bregman projection

We follow the notations that were introduced in the beginning of the proof in Section B. To show that the cost function is discretely convex in $k$ for the primal, it suffices to show that

$$L([k]) := \min_{\hat{p} \in \Delta_k} D_\phi((\hat{p}, 0_{V-k}), p) = D_\phi((T_{[k]}(p), 0_{V-k}), p)$$

is discretely convex in $k$. Indeed, the difference $\text{cost}(k) - L([k]) = \lambda k$ is linear in $k$.

To simplify notation, let us denote $L([k])$ by $L(k)$ and $T_{[k]}$ by $T_k$. From Lemma (B.1) we know that with $\tilde{S}_V := S_{[V]} - k\phi(0)$

$$L(k) = \sum_{j=1}^{k} \{\phi([T_k(p)]_j) - \phi'(p_j)[T_k(p)]_j\} + \tilde{S}_V.$$

Using (10), we know that $f([T_k(p)]_j) = f(p_j) + \nu_{[k]} \; \forall \, j \in [k]$. Again, we simply denote $\nu_{[k]}$ as $\nu_k$. For $j \in [k]$, letting $x = f(p_j) + \nu_k$ in Lemma C.1, we have:

$$
\begin{aligned}
\phi([T_k(p)]_j) - \phi'(p_j)[T_k(p)]_j &= \phi(f^{-1}(f(p_j) + \nu_k)) - f(p_j)f^{-1}(f(p_j) + \nu_k) \\
&= \phi(f^{-1}(x)) - f(p_j)f^{-1}(x) = xf^{-1}(x) - \phi^*(x) - f(p_j)f^{-1}(x) \\
&= (x - f(p_j))f^{-1}(x) - \phi^*(x) = \nu_k[T_k(p)]_j - \phi^*(f(p_j) + \nu_k).
\end{aligned}
$$

But now, using that the nonzero entries of $T_k(p)$ must sum to unity, we find the following simplification:

$$
\begin{aligned}
L(k) &= \sum_{j=1}^{k} \{\nu_k[T_k(p)]_j - \phi^*(f(p_j) + \nu_k)\} + \tilde{S}_V \\
&= \nu_k \sum_{j=1}^{k}[T_k(p)]_j - \sum_{j=1}^{k} \phi^*(f(p_j) + \nu_k) + \tilde{S}_V = \nu_k - \sum_{j=1}^{k} \phi^*(f(p_j) + \nu_k) + \tilde{S}_V. \quad (15)
\end{aligned}
$$

Now, define the auxiliary function $W$ for all $j, \nu$ for which the expression below is well defined:

$$W(k, \nu) := \nu - \sum_{j=1}^{k} \phi^*(f(p_j) + \nu), \quad (16)$$

where $p$ is implicitly kept fixed. From the above calculation, we thus obtain after canceling out terms:

$$L(k+1) - 2L(k) + L(k-1) = W(k+1, \nu_{k+1}) - 2W(k, \nu_k) + W(k-1, \nu_{k-1}).$$

To prove that this is nonnegative, we leverage that $W(k, \cdot)$ is strictly concave in $\nu$ for each $k$, which follows as the Legendre dual mapping $\phi^*$ is strictly convex since so is $\phi$. Then, observe that for every $j$,

$$\frac{\partial}{\partial \nu} W(k, \nu) = 1 - \sum_{j=1}^{k} (\phi^*)'(f(p_i) + \nu) = 1 - \sum_{j=1}^{k} f^{-1}(f(p_j) + \nu). \quad (17)$$

Thus,

$$\frac{\partial}{\partial \nu} W(k, \nu) \mid_{\nu = \nu_k} = 1 - \sum_{j=1}^{k} f^{-1}(f(p_j) + \nu_k) = 1 - \sum_{j=1}^{k} [T_k(p)]_j = 0.$$

As $W(k, \cdot)$ is strictly concave in $\nu$, $W(k, \cdot)$ is maximized at $\nu_k$. Thus, we have: (1) $W(k+1, \nu_{k+1}) \geqslant W(k+1, \nu_k)$, and (2) $W(k-1, \nu_{k-1}) \geqslant W(k-1, \nu_k)$. With these in hand, we have:

$$L(k+1) - 2L(k) + L(k-1) = W(k+1, \nu_{k+1}) - 2W(k, \nu_k) + W(k-1, \nu_{k-1}) \qquad (18)$$
$$\geqslant [W(k+1, \nu_k) - W(k, \nu_k)] - [W(k, \nu_k) - W(k-1, \nu_k)].$$

Now, due to the definition of $W$, the last display equals

$$-\phi^*(f(p_{k+1}) + \nu_k) + \phi^*(f(p_k) + \nu_k) \geqslant 0, \qquad (19)$$

the inequality holding as $p_k \geqslant p_{k+1}$, and as the mapping $p \mapsto \phi^*(f(p) + \nu_k)$ is increasing in $p$ since so are $\phi^*$ and $f$. This concludes the proof.

## E   Proof of discrete convexity for dual Bregman projection

We denote $\theta_x(y) = \phi''(y)(y - x)$. As observed before, we have for all admissible $x, y$ that

$$\frac{\partial}{\partial y} \mathrm{d}_\phi(x, y) = \theta_x(y),$$

and the convexity condition for the second argument of $\mathrm{d}_\phi$ of Assumption 3.2 is given by:

$$\frac{\partial}{\partial y} \theta_x(y) \geqslant 0 \Leftrightarrow \phi''(y) + \phi'''(y)(y - x) \geqslant 0 \quad \text{for all } y \geqslant x \geqslant 0.$$

The dual projection for any $1 \leqslant i \leqslant j \leqslant V$ is given (for optimal Lagrange multiplier $\nu_j$) by:

$$\theta_{p_i}([T_j^*(p)]_i) = \nu_j \Leftrightarrow \phi''([T_j^*(p)]_i)([T_j^*(p)]_i - p_i) = \nu_j.$$

Denote the dual Bregman objective, as a function of the selected sparsity $k$, as:

$$\mathrm{cost}^*(k) = \mathrm{D}_\phi\left(p, (T_k^*(p), 0_{V-k})\right) + \lambda k.$$

We now demonstrate that $\mathrm{cost}^*(k)$ is discretely convex in $k$. For this, we will directly show that the second-order differences of this function are nonnegative at every $k \in \{2, \ldots, V-1\}$. Specifically, we can write:

$$\Delta^{*,2}(k) := \mathrm{cost}^*(k+1) - 2\mathrm{cost}^*(k) + \mathrm{cost}^*(k-1)$$
$$= \mathrm{D}_\phi\left(p, \left(T_{k+1}^*(p), 0_{V-k-1}\right)\right) - 2\mathrm{D}_\phi\left(p, (T_k^*(p), 0_{V-k})\right) + \mathrm{D}_\phi\left(p, \left(T_{k-1}^*(p), 0_{V-k+1}\right)\right)$$

We now decompose this quantity into three terms corresponding to three ranges of index $i \in [V]$, namely $i \in [k-1]$, $i \in \{k, k+1\}$, and $i \in \{k+2, \ldots, V\}$. We obtain:

$$\Delta^{*,2}(k) = \sum_{i=1}^{k-1} \left\{ \left\{ \mathrm{d}_\phi(p_i, [T_{k+1}^*(p)]_i) - \mathrm{d}_\phi(p_i, [T_k^*(p)]_i) \right\} + \left\{ \mathrm{d}_\phi(p_i, [T_{k-1}^*(p)]_i) - \mathrm{d}_\phi(p_i, [T_k^*(p)]_i) \right\} \right\}$$

$$+ \left\{ \left(\phi(p_k) - \phi(0) - \phi'(0) \cdot p_k\right) - 2\left(\phi(p_k) - \phi([T_k^*(p)]_k) - \phi'([T_k^*(p)]_k) \cdot (p_k - [T_k^*(p)]_k)\right) \right.$$
$$+ \left(\phi(p_k) - \phi([T_{k+1}^*(p)]_k) - \phi'([T_{k+1}^*(p)]_k) \cdot (p_k - [T_{k+1}^*(p)]_k)\right)$$
$$+ \left(\phi(p_{k+1}) - \phi(0) - \phi'(0) \cdot p_{k+1}\right) - 2\left(\phi(p_{k+1}) - \phi(0) - \phi'(0) \cdot p_{k+1}\right)$$
$$+ \left. \left(\phi(p_{k+1}) - \phi([T_{k+1}^*(p)]_{k+1}) - \phi'([T_{k+1}^*(p)]_{k+1}) \cdot (p_{k+1} - [T_{k+1}^*(p)]_{k+1})\right) \right\}$$

$$- \sum_{i=k+2}^{V} \left\{ \mathrm{d}_\phi(p_i, 0) - 2\mathrm{d}_\phi(p_i, 0) + \mathrm{d}_\phi(p_i, 0) \right\}.$$

The last sum is identically zero, so we engage with the other two ranges of indices.

**Range 1:** $i \in [k-1]$. For Range 1, recall that for any convex function $\psi$, it holds for any two points $x, y$ in its domain that $\psi(x) - \psi(y) \geqslant \psi'(y)(x-y)$. Now, notice that for each $i$ in Range 1, each of the two terms in figure brackets can be bounded via the convexity of $\mathrm{d}_\phi(x, \cdot)$ in its second argument as:

$$\mathrm{d}_\phi(p_i, [T^*_{k+1}(p)]_i) - \mathrm{d}_\phi(p_i, [T^*_k(p)]_i) \geqslant \left( \frac{\partial}{\partial y} \mathrm{d}_\phi(p_i, y) \right) \Big|_{y=[T^*_k(p)]_i} \cdot \left( [T^*_{k+1}(p)]_i - [T^*_k(p)]_i \right)$$

$$= \theta_{p_i} \left( [T^*_k(p)]_i \right) \cdot \left( [T^*_{k+1}(p)]_i - [T^*_k(p)]_i \right) = \nu_k \cdot \left( [T^*_{k+1}(p)]_i - [T^*_k(p)]_i \right)$$

and:

$$\mathrm{d}_\phi(p_i, [T^*_{k-1}(p)]_i) - \mathrm{d}_\phi(p_i, [T^*_k(p)]_i) \geqslant \left( \frac{\partial}{\partial y} \mathrm{d}_\phi(p_i, y) \right) \Big|_{y=[T^*_k(p)]_i} \cdot \left( [T^*_{k-1}(p)]_i - [T^*_k(p)]_i \right)$$

$$= \theta_{p_i} \left( [T^*_k(p)]_i \right) \cdot \left( [T^*_{k-1}(p)]_i - [T^*_k(p)]_i \right) = \nu_k \cdot \left( [T^*_{k-1}(p)]_i - [T^*_k(p)]_i \right).$$

As a result, we may simplify the Range 1 sum as follows, using that by definition, the first $j$ terms in the projection $T^*_j$ for each $j \in \{k-1, k, k+1\}$ sum to unity:

$$\text{Range 1 Sum} \geqslant \sum_{i=1}^{k-1} \nu_k \cdot \left( \left\{ [T^*_{k+1}(p)]_i - [T^*_k(p)]_i \right\} + \left\{ [T^*_{k-1}(p)]_i - [T^*_k(p)]_i \right\} \right)$$

$$= \nu_k \left( \sum_{i=1}^{k-1} [T^*_{k+1}(p)]_i - 2 \sum_{i=1}^{k-1} [T^*_k(p)]_i + \sum_{i=1}^{k-1} [T^*_{k-1}(p)]_i \right)$$

$$= \nu_k \left( \left( 1 - [T^*_{k+1}(p)]_k - [T^*_{k+1}(p)]_{k+1} \right) - 2(1 - [T^*_k(p)]_k) + 1 \right)$$

$$= \nu_k \left( 2[T^*_k(p)]_k - [T^*_{k+1}(p)]_k - [T^*_{k+1}(p)]_{k+1} \right).$$

**Range 2:** $i \in \{k, k+1\}$. For Range 2, we first note that the following three types of terms cancel out: $\phi(0), \phi(p_k), \phi(p_{k+1})$. Furthermore, terms involving $\phi'(0)$ vanish by assumption.

The remaining terms in the Range 2 sum can then be written as:

$$\text{Range 2 Sum} \geqslant \Big\{ -2 \left( -\phi([T^*_k(p)]_k) - \phi'([T^*_k(p)]_k) \cdot (p_k - [T^*_k(p)]_k) \right)$$

$$+ \left( -\phi([T^*_{k+1}(p)]_k) - \phi'([T^*_{k+1}(p)]_k) \cdot (p_k - [T^*_{k+1}(p)]_k) \right) \Big\}$$

$$+ \Big\{ -\phi([T^*_{k+1}(p)]_{k+1}) - \phi'([T^*_{k+1}(p)]_{k+1}) \cdot (p_{k+1} - [T^*_{k+1}(p)]_{k+1}) \Big\}.$$

Now, we can bound

$$-\phi'([T^*_{k+1}(p)]_{k+1}) \cdot p_{k+1} \geqslant -\phi'([T^*_{k+1}(p)]_{k+1}) \cdot p_k,$$

using that $p_k \geqslant p_{k+1}$ and the strict convexity of $\phi$. We find the lower bound

$$\text{Range 2 Sum} \geqslant -2 \Big\{ -\phi([T^*_k(p)]_k) - \phi'([T^*_k(p)]_k) \cdot (p_k - [T^*_k(p)]_k) \Big\}$$

$$+ \Big\{ -\phi([T^*_{k+1}(p)]_k) - \phi'([T^*_{k+1}(p)]_k) \cdot (p_k - [T^*_{k+1}(p)]_k) \Big\}$$

$$+ \Big\{ -\phi([T^*_{k+1}(p)]_{k+1}) - \phi'([T^*_{k+1}(p)]_{k+1}) \cdot (p_k - [T^*_{k+1}(p)]_{k+1}) \Big\}.$$

By adding and subtracting the term $\phi(p_k)$ twice, we have the following equivalent bound:

$$\text{Range 2 Sum} \geqslant -2 \Big\{ \phi(p_k) - \phi([T^*_k(p)]_k) - \phi'([T^*_k(p)]_k) \cdot (p_k - [T^*_k(p)]_k) \Big\}$$

$$+ \Big\{ \phi(p_k) - \phi([T^*_{k+1}(p)]_k) - \phi'([T^*_{k+1}(p)]_k) \cdot (p_k - [T^*_{k+1}(p)]_k) \Big\}$$

$$+ \Big\{ \phi(p_k) - \phi([T^*_{k+1}(p)]_{k+1}) - \phi'([T^*_{k+1}(p)]_{k+1}) \cdot (p_k - [T^*_{k+1}(p)]_{k+1}) \Big\}$$

$$= -2\mathrm{d}_\phi\left(p_k, [T^*_k(p)]_k\right) + \mathrm{d}_\phi\left(p_k, [T^*_{k+1}(p)]_k\right) + \mathrm{d}_\phi\left(p_k, [T^*_{k+1}(p)]_{k+1}\right).$$

**Returning to the main bound** We can now merge the cases, resulting in the following tight lower bound of the second differential of the cost function:

$$
\begin{aligned}
\Delta^{*,2}(k) \geqslant \quad & \nu_k \left( 2[T_k^*(p)]_k - [T_{k+1}^*(p)]_k - [T_{k+1}^*(p)]_{k+1} \right) \\
- \quad & 2\mathrm{d}_\phi \left( p_k, [T_k^*(p)]_k \right) + \mathrm{d}_\phi \left( p_k, [T_{k+1}^*(p)]_k \right) + \mathrm{d}_\phi \left( p_k, [T_{k+1}^*(p)]_{k+1} \right).
\end{aligned}
$$

Now, define the following key auxiliary function $\psi_k : [0,1] \to \mathbb{R}$, such that for all $x \in [0,1]$:

$$
\psi_k(x) = \nu_k \cdot x - \mathrm{d}_\phi(p_k, x).
$$

This lets us rewrite our lower bound equivalently as:

$$
\Delta^{*,2}(k) \geqslant 2\,\psi\left([T_k^*(p)]_k\right) - \psi\left([T_{k+1}^*(p)]_k\right) - \psi\left([T_{k+1}^*(p)]_{k+1}\right). \tag{20}
$$

We now establish a monotonicity property for $\psi_k$.

**Lemma E.1.** *For every $k \in [V]$ the function $\psi_k(x)$ is increasing on $x \in [0, [T_k^*(p)]_k]$.*

*Proof.* We consider the derivative of the function $\psi_k$:

$$
\frac{\partial}{\partial x}\psi_k(x) = \nu_k - \frac{\partial}{\partial x}\mathrm{d}_\phi(p_k, x) = \nu_k - \theta_{p_k}(x) = \theta_{p_k}\left([T_k^*(p)]_k\right) - \theta_{p_k}(x),
$$

where we have used the connection between $\theta_x(y)$ and $\nu_k$ (see Lemma A.1).

Now, recalling that by assumption, $\frac{\partial}{\partial y}\theta_x(y) \geqslant 0$ for all $y \geqslant x \geqslant 0$, and using that $[T_k^*(p)]_k \geqslant p_k$ by the properties of the dual projection method (see Lemma A.1), we have that:

$$
\frac{\partial}{\partial x}\psi_k(x) = \theta_{p_k}\left([T_k^*(p)]_k\right) - \theta_{p_k}(x) \geqslant 0,
$$

so long as $0 \leqslant x \leqslant [T_k^*(p)]_k$. $\qquad\square$

Continuing, by the properties of the dual projection, we have:

$$
[T_k^*(p)]_k \geqslant [T_{k+1}^*(p)]_k \geqslant [T_{k+1}^*(p)]_{k+1}.
$$

In view of Lemma E.1, (20) implies that

$$
\Delta^{*,2}(k) \geqslant \left[\psi\left([T_k^*(p)]_k\right) - \psi\left([T_{k+1}^*(p)]_k\right)\right] + \left[\psi\left([T_k^*(p)]_k\right) - \psi\left([T_{k+1}^*(p)]_{k+1}\right)\right] \geqslant 0 + 0 = 0.
$$

This concludes the proof of dual discrete convexity of the Bregman cost function.

# F   Algorithmic details

## F.1   Computing the dual renormalization map

Recall that when $\phi$ is dual valid, the renormalization map $T_\phi^*$ is uniquely defined for $x \in \Delta_{\mathrm{sub},k}$ with $\sum_i x_i > 0$ by the fixed point equation (see Lemma A.1)

$$
[T_\phi^*(x)]_i = x_i + \nu^*/f'([T_\phi^*(x)]_i) \quad \text{for all } i \in [k], \text{ where } \nu^* \in \mathbb{R} \text{ is chosen so that } \sum_{i=1}^{k}[T_\phi^*(x)]_i = 1.
$$

To compute $T_\phi^*$, recall from Section C.2.1 the function $\Psi$ from (13) with $\Psi(x,y,\nu) := \phi''(y)(y - x) - \nu$ for all $x, y, \nu$. Then, for a fixed $\nu$, $[T(x)]_i$ satisfying the equation $[T(x)]_i = x_i + \nu/f'([T(x)]_i)$ is equivalent to solving $\Psi(x_i, y_i, \nu) = 0$ for $y_i = [T(x)]_i$. The monotonicity properties from Lemma A.1 then suggest the following algorithm, consisting of a binary search over $\nu \in (0, M]$, and then over each coordinate of $T$ solving $\phi''([T(x)]_i)([T(x)]_i - x_i) = \nu$.

**Algorithm 1** Dual Renormalization Map $T_\phi^*(x)$ via Nested Binary Search

---

**Require:** Convex generator $\phi$ with derivatives $f = \phi'$, $f'' = \phi''$; input vector $x \in \Delta_{\mathrm{sub},k}$ with $\sum x_i < 1$; tolerance $\varepsilon > 0$
**Ensure:** Renormalized vector $\hat{p} = T_\phi^*(x) \in \Delta_k$
 1: **function** DUALRENORMALIZE($x, \phi, \varepsilon$)
 2:     $k \leftarrow$ length of $x$
 3:     $f'' \leftarrow \phi''$
 4:     $M \leftarrow \phi''(1) \cdot (1 - \max_i x_i)$                                $\triangleright$ Upper bound on feasible $\nu$
 5:     Initialize $\nu_{\mathrm{low}} \leftarrow 0$, $\nu_{\mathrm{high}} \leftarrow M$
 6:     **while** $\nu_{\mathrm{high}} - \nu_{\mathrm{low}} > \varepsilon$ **do**
 7:         $\nu \leftarrow (\nu_{\mathrm{low}} + \nu_{\mathrm{high}})/2$
 8:         **for** $i = 1$ to $k$ **do**
 9:             $x_i \leftarrow x[i]$
10:             $y[i] \leftarrow$ SOLVEROOT($x_i, \nu, f'', \varepsilon$)
11:         **end for**
12:         $G \leftarrow \sum_{i=1}^k y[i]$
13:         **if** $G < 1$ **then**
14:             $\nu_{\mathrm{low}} \leftarrow \nu$
15:         **else**
16:             $\nu_{\mathrm{high}} \leftarrow \nu$
17:         **end if**
18:     **end while**
19:     **return** $y$
20: **end function**
21: **function** SOLVEROOT($x_i, \nu, f'', \varepsilon$)
22:     $a \leftarrow x_i$, $b \leftarrow 1$
23:     **while** $b - a > \varepsilon$ **do**
24:         $m \leftarrow (a + b)/2$
25:         $\Psi \leftarrow f''(m) \cdot (m - x_i) - \nu$
26:         **if** $\Psi < 0$ **then**
27:             $a \leftarrow m$
28:         **else**
29:             $b \leftarrow m$
30:         **end if**
31:     **end while**
32:     **return** $(a + b)/2$
33: **end function**

---

### F.2 Pseudocode for algorithms

See Algorithm 3 and Algorithm 4 for pseudocode for sparse primal (resp. dual) Bregman decoding.

---

**Algorithm 2** Discrete Binary Search for Unimodal Cost Minimization

---

**Require:** Callable function COMPUTECOST, maximum support size $V$
**Ensure:** Optimal support size $k^*$ minimizing COMPUTECOST$(k)$
 1: **function** BINARYSEARCH(COMPUTECOST, $V$)
 2:     $c_1 \leftarrow$ COMPUTECOST(1)
 3:     $c_2 \leftarrow$ COMPUTECOST(2)
 4:     **if** $c_2 - c_1 \geqslant 0$ **then**
 5:         **return** 1
 6:     **end if**
 7:     $c_{V-1} \leftarrow$ COMPUTECOST$(V - 1)$
 8:     $c_V \leftarrow$ COMPUTECOST$(V)$
 9:     **if** $c_V - c_{V-1} \leqslant 0$ **then**
10:         **return** $V$
11:     **end if**
12:     Initialize $L \leftarrow 1, R \leftarrow V$
13:     **while** $R - L > 1$ **do**
14:         $m \leftarrow \lfloor (L + R)/2 \rfloor$
15:         $c_m \leftarrow$ COMPUTECOST$(m)$
16:         $c_{m+1} \leftarrow$ COMPUTECOST$(m + 1)$
17:         **if** $c_{m+1} - c_m \geqslant 0$ **then**
18:             $R \leftarrow m$
19:         **else**
20:             $L \leftarrow m$
21:         **end if**
22:     **end while**
23:     **return** $R$
24: **end function**

---

---

**Algorithm 3** Regularized Sparse Primal Bregman Decoding

---

**Require:** Probability vector $p \in \Delta_V$, valid convex generator $\phi$, sparsity penalty $\lambda \geqslant 0$
**Ensure:** Sparse decoded distribution $\hat{p} \in \Delta_V$
 1: **function** SPARSEPRIMALBREGMANDECODE$(p, \phi, \lambda)$
 2:     Sort $p$ in descending order: $p_{(1)} \geqslant p_{(2)} \geqslant \cdots \geqslant p_{(V)}$
 3:     Define $f = \phi'$
 4:     **function** COMPUTERENORMALIZATION$(x \in \mathbb{R}^k)$
 5:         Solve for $\nu \in \mathbb{R}$ such that $\sum_{i=1}^{k} f^{-1}(f(x_i) + \nu) = 1$
 6:         **return** $\hat{p}^{(k)}$ with $[\hat{p}^{(k)}]_i = f^{-1}(f(x_i) + \nu)$ for $i \in [k]$
 7:     **end function**
 8:     **function** COMPUTECOST$(k)$
 9:         Let $x = p[1{:}k]$
10:         $\hat{p}^{(k)} \leftarrow$ COMPUTERENORMALIZATION$(x)$
11:         Pad with zeros: $\hat{p}^{(k)} \leftarrow (\hat{p}_1^{(k)}, \ldots, \hat{p}_k^{(k)}, 0, \ldots, 0)$
12:         Compute $D_\phi(\hat{p}^{(k)}, p) = \sum_{i=1}^{V} \left[ \phi(\hat{p}_i^{(k)}) - \phi(p_i) - f(p_i)(\hat{p}_i^{(k)} - p_i) \right]$
13:         **return** cost$(k) = D_\phi(\hat{p}^{(k)}, p) + \lambda k$
14:     **end function**
15:     $k^* \leftarrow$ BINARYSEARCH(ComputeCost, $V$)
16:     Recompute $\hat{p}^{(k^*)}$ using COMPUTERENORMALIZATION$(p[1{:}k^*])$
17:     Pad with zeros to full length $V$
18:     **return** $\hat{p}^{(k^*)}$
19: **end function**

---

---

**Algorithm 4** Regularized Sparse Dual Bregman Decoding

---

**Require:** Probability vector $p \in \Delta_V$, valid convex generator $\phi$, sparsity penalty $\lambda \geqslant 0$
**Ensure:** Sparse decoded distribution $\hat{p} \in \Delta_V$
1: **function** SPARSEDUALBREGMANDECODE($p, \phi, \lambda$)
2:     Sort $p$ in descending order: $p_{(1)} \geqslant p_{(2)} \geqslant \cdots \geqslant p_{(V)}$
3:     Define $f = \phi'$, $\quad f' = \phi''$
4:     **function** COMPUTEDUALRENORMALIZATION($x \in \mathbb{R}^k$)
5:         Solve for $\nu \in \mathbb{R}$ such that: $\sum_{i=1}^{k} [T_\phi^*(x)]_i = 1$, where $[T_\phi^*(x)]_i$ satisfies the fixed-point
    equation: $[T_\phi^*(x)]_i = x_i + \nu / f'([T_\phi^*(x)]_i)$.
6:         **return** $\hat{p}^{(k)} = T_\phi^*(x)$
7:     **end function**
8:     **function** COMPUTEDUALCOST($k$)
9:         Let $x = p[1{:}k]$
10:         $\hat{p}^{(k)} \leftarrow$ COMPUTEDUALRENORMALIZATION($x$)
11:         Pad with zeros: $\hat{p}^{(k)} \leftarrow (\hat{p}_1^{(k)}, \ldots, \hat{p}_k^{(k)}, 0, \ldots, 0)$
12:         Compute $\mathrm{D}_\phi(p, \hat{p}^{(k)}) = \sum_{i=1}^{V} \left[ \phi(p_i) - \phi(\hat{p}_i^{(k)}) - f(\hat{p}_i^{(k)})(p_i - \hat{p}_i^{(k)}) \right]$
13:         **return** $\mathrm{cost}(k) = \mathrm{D}_\phi(p, \hat{p}^{(k)}) + \lambda k$
14:     **end function**
15:     $k^* \leftarrow$ BINARYSEARCH(ComputeDualCost, $V$)
16:     Recompute $\hat{p}^{(k^*)}$ using COMPUTEDUALRENORMALIZATION($p[1{:}k^*]$)
17:     Pad with zeros to full length $V$
18:     **return** $\hat{p}^{(k^*)}$
19: **end function**

---

# G   Example: $\alpha$-Bregman decoding

## G.1   Proof of Lemma 4.3

We first restate the lemma.

**Lemma G.1.** *All generator functions $\phi_\alpha$, $\alpha > 1$, are dual-valid and satisfy Assumption (A2).*

*Proof.* For Assumption 3.2, we can explicitly write:

$$\mathrm{d}_\phi(x,y) = \frac{x^\alpha}{\alpha(\alpha-1)} - \frac{y^\alpha}{\alpha(\alpha-1)} - \frac{y^{\alpha-1}}{\alpha-1}(x-y) = \frac{y^\alpha}{\alpha} - \frac{x}{\alpha-1}y^{\alpha-1} + \frac{x^\alpha}{\alpha(\alpha-1)}.$$

Therefore, the second derivative in $y$ of this expression is

$$(\alpha-1)y^{\alpha-2} - (\alpha-2)xy^{\alpha-3} = y^{\alpha-3}(y(\alpha-1) - x(\alpha-2)) = y^{\alpha-3}\left(y(\alpha-1) + x(2-\alpha)\right).$$

Now, if $y \geqslant x$, then using $\alpha - 1 \geqslant 0$ we have that the above expression is

$$\geqslant y^{\alpha-3}(x(\alpha-1) + x(2-\alpha)) = y^{\alpha-3}x \geqslant 0,$$

confirming the convexity in $y$. Now for the condition that $x \mapsto u(x) := x\phi''(x)/\phi'(x)$ is non-decreasing from Assumption (A2), we can observe that

$$\phi'(x)\phi'''(x) - \phi''(x)^2 = \frac{x^{\alpha-1}}{\alpha-1} \cdot (\alpha-2)x^{\alpha-3} - (x^{\alpha-2})^2 = -\frac{x^{2\alpha-4}}{\alpha-1}.$$

Therefore, we identically have:

$$\phi'(x)\phi''(x) + x(\phi'(x)\phi'''(x) - \phi''(x)^2) = \frac{x^{2\alpha-3}}{\alpha-1} - x\frac{x^{2\alpha-4}}{\alpha-1} = 0,$$

thus concluding the proof. $\qquad\square$

## G.2   Proof of Proposition 4.2

Recall the $\alpha$–renormalization map $[T_\alpha(p)]_i = \left(p_i^{\alpha-1} + \nu\right)^{\frac{1}{\alpha-1}}, i \in [k]$, where the shift parameter $\nu = \nu(\alpha, p)$ is chosen so that $\sum_{i=1}^{k}[T_\alpha(p)]_i = 1$. We treat each value (or limit) of $\alpha$ in turn.

**The limit $\alpha \to -\infty$.** Define

$$F_\beta(\nu) := \sum_{i=1}^k \left(p_i^\beta + \nu\right)^{1/\beta}, \qquad \beta := \alpha - 1 < 0.$$

Because $x \mapsto x^{1/\beta}$ is strictly *decreasing* and convex on $(0, \infty)$ for $\beta < 0$, $F_\beta$ is strictly decreasing and continuous on the interval $\left(-\min_i p_i^\beta, \infty\right)$. Moreover, $\lim_{\nu \downarrow - \min_i p_i^\beta} F_\beta(\nu) = \infty$ and $\lim_{\nu \uparrow \infty} F_\beta(\nu) = 0$, so a unique root $\nu_\beta$ with $F_\beta(\nu_\beta) = 1$ exists. Because $F_\beta(0) = S := \sum_{i=1}^k p_i \leqslant 1$ and $F_\beta$ is decreasing, we have $\nu_\beta \leqslant 0$.

Let $q_i^{(\alpha)} = [T_\alpha(p)]_i = \left(p_i^\beta + \nu_\beta\right)^{1/\beta}$, and $i^*$ be the index where $p_i$ is largest. Using the constraint $\sum_i q_i^{(\alpha)} = 1$,

$$q_{i^\star}^{(\alpha)} = 1 - \sum_{i \neq i^\star} q_i^{(\alpha)} = \delta + p_{i^\star} + \sum_{i \neq i^\star} \left(p_i - q_i^{(\alpha)}\right) \geqslant p_{i^\star} + \delta.$$

Raising $q_{i^\star}^{(\alpha)} = \left(p_{i^\star}^\beta + \nu_\beta\right)^{1/\beta}$ to the power $\beta < 0$ yields

$$\nu_\beta = \left(p_{i^\star} + \delta + R_\beta\right)^\beta - p_{i^\star}^\beta, \qquad R_\beta := \sum_{i \neq i^\star} \left(p_i - q_i^{(\alpha)}\right) \in [0, \delta]. \tag{21}$$

For $i \neq i^\star$, we have $\nu_\beta / p_i^\beta \to 0$. Indeed, (21) implies $|\nu_\beta| \leqslant p_{i^\star}^\beta (c^\beta - 1)$ with $c := (p_{i^\star} + \delta)/p_{i^\star} > 1$. Because $\beta \to -\infty$, $c^\beta \to 0$, we have $|\nu_\beta| = O(p_{i^\star}^\beta) = o(p_i^\beta)$. Then,

$$q_i^{(\alpha)} = p_i \left(1 + \frac{\nu_\beta}{p_i^\beta}\right)^{1/\beta} \to p_i, \qquad i \neq i^\star. \tag{22}$$

Summing (22) over $i \neq i^\star$ and using $\sum_i q_i^{(\alpha)} = 1$ gives

$$q_{i^\star}^{(\alpha)} = 1 - \sum_{i \neq i^\star} q_i^{(\alpha)} \to 1 - \sum_{i \neq i^\star} p_i = p_{i^\star} + \delta. \tag{23}$$

Equations (22) and (23) establish $q^{(\alpha)} \to T_{-\infty}(p)$ component-wise, completing the proof.

**The case $\alpha = \frac{3}{2}$.** Now $\alpha - 1 = \frac{1}{2}$, hence $[T_{1.5}(p)]_i = \left(\sqrt{p_i} + \nu\right)^2$, $i \in [k]$. Set $s := \sum_{j=1}^k \sqrt{p_j}$ and $A := \sum_{j=1}^k p_j$. The normalization condition becomes

$$1 = \sum_{i=1}^k (\sqrt{p_i} + \nu)^2 = A + 2s\nu + k\nu^2.$$

Solving $k\nu^2 + 2s\nu + (A - 1) = 0$ for the root that yields non–negative probabilities gives $\nu = \frac{-s + \sqrt{s^2 + k(1-A)}}{k}$. Hence

$$[T_{1.5}(p)]_i = \left(\sqrt{p_i} + \frac{\sqrt{s^2 + k(1-A)} - s}{k}\right)^2, \qquad i \in [k].$$

**The case $\alpha = 2$.** Here $\alpha - 1 = 1$, so Definition 4.1 yields $[T_2(p)]_i = p_i + \nu$, $i \in [k]$. The normalization condition gives $1 = \sum_{i=1}^k (p_i + \nu) = \sum_{i=1}^k p_i + k\nu$, hence $\nu = \frac{1 - \sum_{j=1}^k p_j}{k}$. Substituting yields

$$[T_2(p)]_i = p_i + \frac{1 - \sum_{j=1}^k p_j}{k}, \qquad i \in [k].$$

**The limit $\alpha \to +\infty$.** Write $\beta := \alpha - 1 \to +\infty$. Let $\nu = c^\beta$ with $c \in [0, 1]$. Then

$$[T_\alpha(p)]_i = \left(p_i^\beta + c^\beta\right)^{1/\beta} = \exp\left\{\frac{1}{\beta} \log\left(p_i^\beta + c^\beta\right)\right\}.$$

Using $\frac{1}{\beta} \log(a^\beta + b^\beta) \to \log(\max\{a, b\})$ as $\beta \to \infty$ gives $\lim_{\alpha \to \infty} [T_\alpha(p)]_i = \max\{p_i, c\}$. Choose the *water level* $c$ so that $\sum_{i=1}^k \max\{p_i, c\} = 1$. This furnishes the claimed water–filling rule.

The four cases above prove Proposition 4.2. $\qquad\square$

### G.3 Illustrating primal and dual renormalization

We consider the peaked vector $v = [0.1, 0.001, 0.001, 0.001, 0.001]$, and plot how both of its distinct constituent values get transformed by the primal and dual Bregman $\alpha$-renormalization (by symmetry, all copies of $0.001$ are guaranteed to get mapped to the same value by any of our renormalizations). The resulting plots are in Figure 4. As predicted by our theory, both renormalization families coincide at three values of the parameter, namely at $\alpha \in \{1, 2, \infty\}$. Furthermore, the primal family evolves more gradually than the dual family between the endpoints of the parameter interval $\alpha \in (1, 2]$, while the reverse behavior occurs for $\alpha \in (2, \infty)$ (where both renormalizations gradually converge to the water-filling limit which, in this case, is the uniform distribution).

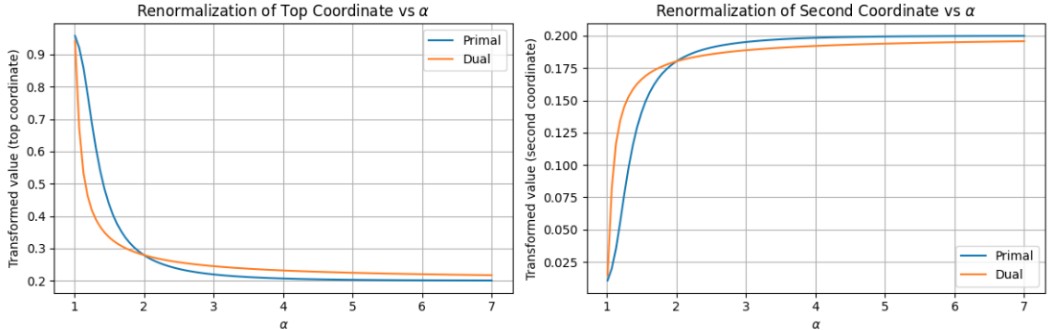

Figure 4: Comparison of primal and dual renormalization maps: The transformation of the larger value ($0.1$, left) and of the smaller value ($0.001$, right).

### G.4 Illustrating general nonconvexity of dual renormalization

Figure 5 illustrates that the dual Bregman objective can in general be non-convex for large $\alpha$.

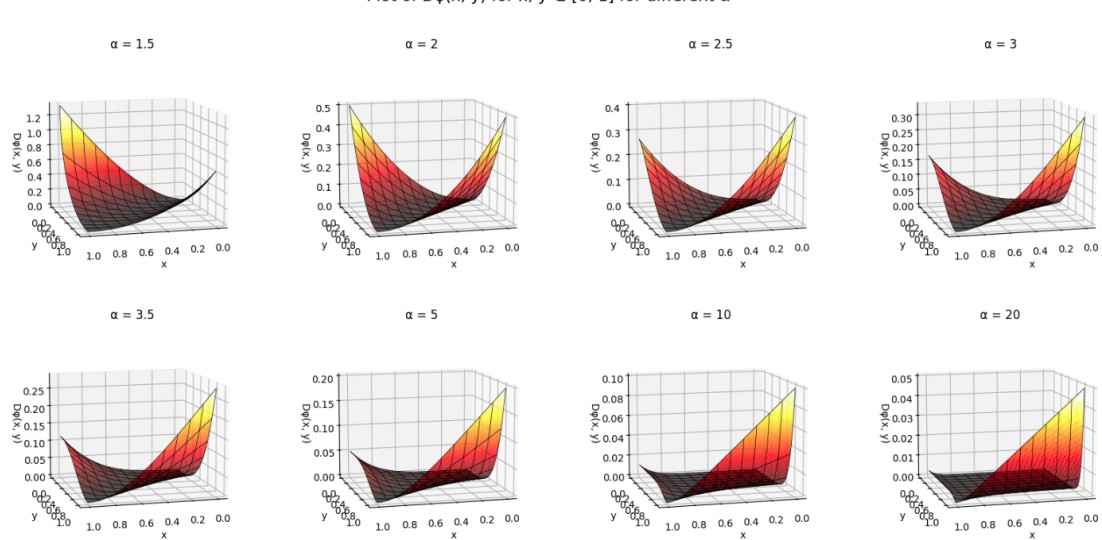

Figure 5: Nonconvexity of the Bregman dual landscape on the square $(x, y) \in [0, 1]^2$.

### G.5 Illustrating discrete convexity

Figure 6 illustrates that the loss function $\mathrm{cost}(\cdot)$ defined in (6) is discretely convex for both the primal and dual decoding strategies. Here, we have chosen $V = 80$ and the regularization parameter $\lambda$ as $1/80$. When $k$ is close to $V$, the renormalization maps are all close to the true vector $p$, regardless of the value of $\alpha$, and hence the loss primarily depends on the regularization term $\lambda k$, which here equals $\lambda k = 1$ for $k = 80$. Thus, all curves (corresponding to different values of $\alpha$) for both the primal and dual plots, asymptote to linearity and converge to this value at $k = 80$.

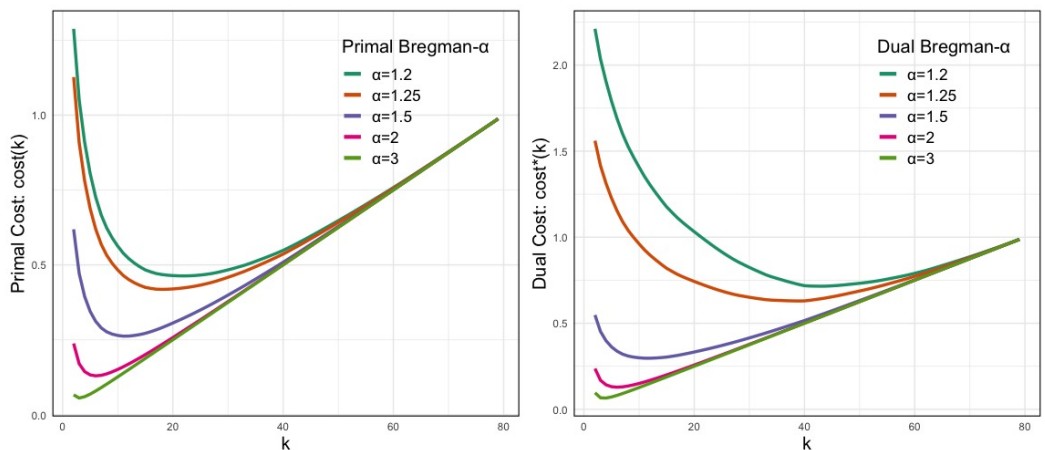

Figure 6: Discrete convexity of the function $k \mapsto \text{cost}(k)$ for primal and dual Bregman $\alpha$-decoding.

### G.6 The simultaneous effects of Bregman decoding and temperature scaling

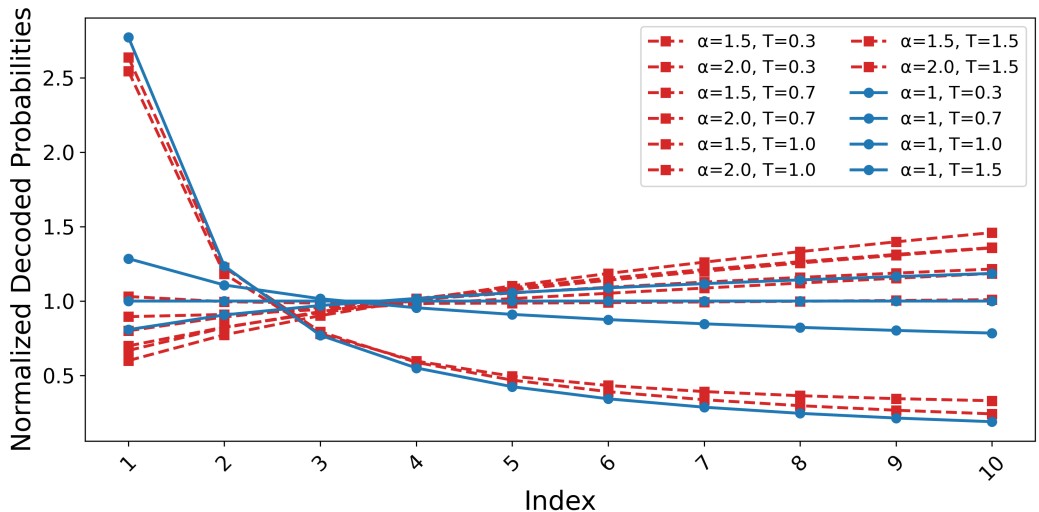

Figure 7: Comparison with changing the temperature.

Here, we provide a plot to help compare the simultaneous effects of Bregman decoding and temperature scaling. We use the same simulation setting and plotting style as in our figure from the introduction (Section 1); except we only plot the nonzero probabilities (i.e., the top $k = 10$ probabilities), and we plot the *relative* sizes of the probabilities compared to the standard top-$k$ decoding. Further, we use the same $\alpha$ and temperature hyperparameters used in our experiments in Table 1. The results are shown in Figure 7. Standard top-$k$ decoding corresponds to $\alpha = 1$ and $T = 1$. From the figure, it appears that the effect of $\alpha > 1$ is to moderate/regularize the amount by which the small probabilities are pushed to zero; which could potentially be one reason why $\alpha$-Bregman decoding with $\alpha > 1$ can perform better at high temperatures.

## H  Supplementary experimental details

### H.1  Compute resources

The experiments were conducted on a system running Rocky Linux 8.10, with 64 CPU cores of Intel(R) Xeon(R) Gold 6448Y processors at 2.10 GHz, 1 TB of RAM, and 8 NVIDIA L40S GPUs with 46 GB of memory each. All experiments can be done with only one GPU and multiple GPUs

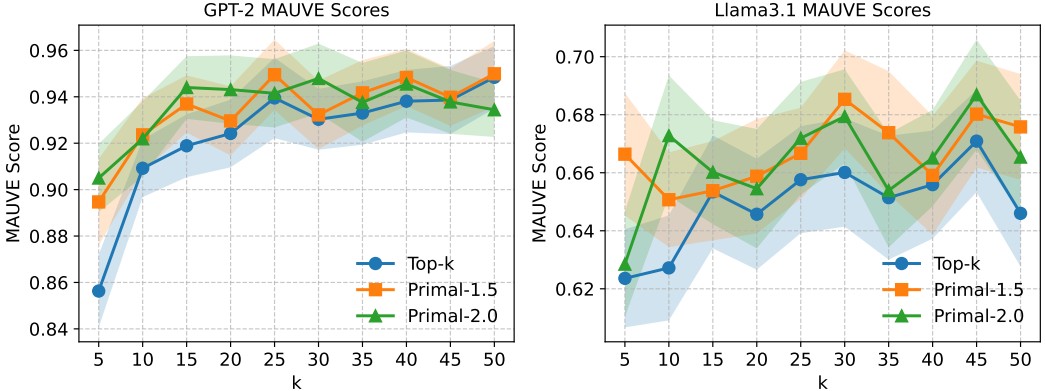

Figure 8: MAUVE scores results between generated and human-written text for GPT2-large (left panel) and LLaMA 3.1 8B (right panel), for various $k$ values. We show top-$k$ decoding and primal decoding with $\alpha \in \{1.5, 2.0\}$. Standard deviations are estimated using 50 bootstrap resamples

were used only to parallelize experiments. The software environment used Python 3.11.11, PyTorch 2.5.1, and CUDA 12.4.

## H.2 Supplementary experimental results

In this section, we provide additional experimental results to supplement those from Section 5.

Table 2 shows results analogous to those in Table 1 for $\lambda \in \{0.1, 0.001\}$.

Table 2: Accuracy on GSM8K for LLaMA 3.1 8B using Bregman primal decoding ($\lambda \in \{0.1, 0.001\}$, $\alpha \in \{1.5, 2.0\}$) and top-$k$ decoding, across different temperature settings. For top-$k$, $k$ equals the averaged optimal $k^*$ from the corresponding primal decoding run (matching temperature, $\lambda$, and $\alpha$). Standard deviations are estimated using 1000 bootstrap resamples.

| Temp | $\lambda = 0.1$ | | Top-$k$ ($\lambda = 0.1$) | $\lambda = 0.001$ | | Top-$k$ ($\lambda = 0.001$) |
| | $\alpha = 1.5$ | $\alpha = 2.0$ | | $\alpha = 1.5$ | $\alpha = 2.0$ | |
|---|---|---|---|---|---|---|
| 0.3 | $83.93_{\pm 1.01}$ | $84.46_{\pm 1.00}$ | $84.69_{\pm 0.99}$ $84.69_{\pm 0.99}$ | $83.93_{\pm 1.01}$ | $85.29_{\pm 0.98}$ | $83.62_{\pm 1.02}$ $83.62_{\pm 1.02}$ |
| 0.7 | $83.47_{\pm 1.02}$ | $85.29_{\pm 0.98}$ | $84.69_{\pm 0.99}$ $84.69_{\pm 0.99}$ | $82.18_{\pm 1.05}$ | $82.41_{\pm 1.05}$ | $83.78_{\pm 1.02}$ $83.78_{\pm 1.02}$ |
| 1.0 | $84.46_{\pm 1.00}$ | $84.38_{\pm 1.00}$ | $84.69_{\pm 0.99}$ $84.69_{\pm 0.99}$ | $78.92_{\pm 1.12}$ | $80.89_{\pm 1.08}$ | $78.54_{\pm 1.13}$ $81.20_{\pm 1.08}$ |
| 1.5 | $83.78_{\pm 1.02}$ | $84.38_{\pm 1.00}$ | $84.69_{\pm 0.99}$ $84.69_{\pm 0.99}$ | $69.22_{\pm 1.23}$ | $73.92_{\pm 1.21}$ | $64.67_{\pm 1.32}$ $75.97_{\pm 1.18}$ |

Figure 8 presents the MAUVE scores comparing generated and human-written text under different decoding strategies. While primal decoding shows a slight advantage over top-$k$ decoding, the differences are not statistically significant. We report standard deviations estimated from 50 bootstrap resamples; a higher number of resamples was not used due to the high computational cost of MAUVE score evaluation.

## H.3 Experiments for Larger models: Qwen and Phi

We repeat our experiments for Qwen2.5-14B-Instruct and Phi-3-medium-4k-instruct.

Figure 9 shows results analogous to those in Figure 3. Table 3 and 4 show the accuracy on GSM8K, analogously to Table 1 and 2. Table 5 and 6 show results for the Phi-3-medium-4k-instruct model.

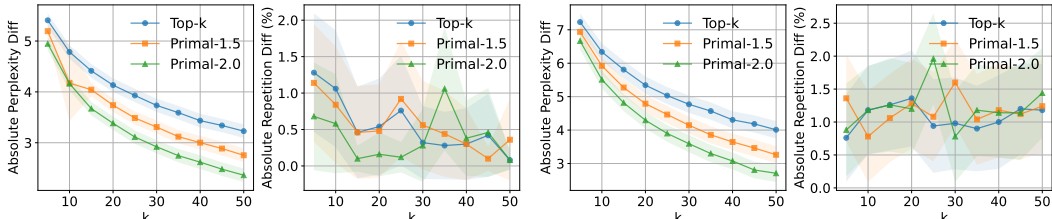

Figure 9: Perplexity and repetition frequency differences between generated and human-written text for Phi-3-medium-4k-instruct (left two panels) and Qwen2.5-14B-Instruct (right two panels), for various $k$ values. We show top-$k$ decoding and primal decoding with $\alpha \in \{1.5, 2.0\}$. Standard deviations are estimated using 1000 bootstrap resamples.

Table 3: Accuracy on GSM8K for Qwen2.5-14B-Instruct using Bregman primal decoding ($\lambda \in \{0.1, 0.01\}$, $\alpha \in \{1.5, 2.0\}$) and top-$k$ decoding, for various temperatures. For top-$k$, $k$ equals the averaged $k^*$ from primal decoding with matching temperature, $\lambda$, and $\alpha$. Standard deviations are over 1000 bootstrap resamples.

| Temp | $\lambda = 0.1$ | | Top-$k$ ($\lambda = 0.1$) | | $\lambda = 0.01$ | | Top-$k$ ($\lambda = 0.01$) | |
| | $\alpha = 1.5$ | $\alpha = 2.0$ | | | $\alpha = 1.5$ | $\alpha = 2.0$ | | |
|---|---|---|---|---|---|---|---|---|
| 0.3 | $82.71_{\pm1.04}$ | $82.26_{\pm1.05}$ | $81.42_{\pm1.07}$ | $81.43_{\pm1.07}$ | $82.64_{\pm1.04}$ | $82.18_{\pm1.05}$ | $81.43_{\pm1.07}$ | $81.43_{\pm1.07}$ |
| 0.7 | $81.73_{\pm1.06}$ | $81.05_{\pm1.08}$ | $81.43_{\pm1.07}$ | $81.43_{\pm1.07}$ | $79.53_{\pm1.11}$ | $80.21_{\pm1.10}$ | $80.21_{\pm1.10}$ | $81.43_{\pm1.07}$ |
| 1.0 | $80.59_{\pm1.09}$ | $81.50_{\pm1.07}$ | $81.43_{\pm1.07}$ | $81.43_{\pm1.07}$ | $78.85_{\pm1.12}$ | $80.29_{\pm1.10}$ | $79.30_{\pm1.12}$ | $81.43_{\pm1.07}$ |
| 1.5 | $80.89_{\pm1.08}$ | $81.73_{\pm1.06}$ | $81.43_{\pm1.07}$ | $81.43_{\pm1.07}$ | $77.18_{\pm1.16}$ | $78.99_{\pm1.12}$ | $77.48_{\pm1.15}$ | $81.43_{\pm1.07}$ |

Table 4: Accuracy on GSM8K for Qwen2.5-14B-Instruct using Bregman primal decoding ($\lambda \in \{0.001, 0.0001\}$, $\alpha \in \{1.5, 2.0\}$) and top-$k$ decoding, for various temperatures. For top-$k$, $k$ equals the averaged $k^*$ from primal decoding with matching temperature, $\lambda$, and $\alpha$. Standard deviations are over 1000 bootstrap resamples.

| Temp | $\lambda = 0.001$ | | Top-$k$ ($\lambda = 0.001$) | | $\lambda = 0.0001$ | | Top-$k$ ($\lambda = 0.0001$) | |
| | $\alpha = 1.5$ | $\alpha = 2.0$ | | | $\alpha = 1.5$ | $\alpha = 2.0$ | | |
|---|---|---|---|---|---|---|---|---|
| 0.3 | $82.11_{\pm1.06}$ | $82.49_{\pm1.05}$ | $82.41_{\pm1.05}$ | $82.56_{\pm1.05}$ | $81.88_{\pm1.06}$ | $82.26_{\pm1.05}$ | $82.03_{\pm1.06}$ | $82.41_{\pm1.05}$ |
| 0.7 | $80.21_{\pm1.10}$ | $79.76_{\pm1.11}$ | $80.06_{\pm1.10}$ | $80.21_{\pm1.10}$ | $79.61_{\pm1.11}$ | $79.76_{\pm1.11}$ | $79.98_{\pm1.10}$ | $80.06_{\pm1.10}$ |
| 1.0 | $78.92_{\pm1.12}$ | $78.32_{\pm1.14}$ | $79.38_{\pm1.11}$ | $79.30_{\pm1.12}$ | $78.47_{\pm1.13}$ | $79.30_{\pm1.12}$ | $78.77_{\pm1.13}$ | $79.38_{\pm1.11}$ |
| 1.5 | $76.72_{\pm1.16}$ | $78.01_{\pm1.14}$ | $75.89_{\pm1.18}$ | $77.48_{\pm1.15}$ | $74.91_{\pm1.19}$ | $74.91_{\pm1.19}$ | $71.19_{\pm1.25}$ | $75.89_{\pm1.18}$ |

Table 5: Accuracy on GSM8K for Phi-3-medium-4k-instruct using Bregman primal decoding ($\lambda \in \{0.1, 0.01\}$, $\alpha \in \{1.5, 2.0\}$) and top-$k$ decoding, for various temperatures. For top-$k$, $k$ equals the averaged $k^*$ from primal decoding with matching temperature, $\mu$, and $\alpha$. Standard deviations are over 1000 bootstrap resamples.

| Temp | $\lambda = 0.1$ | | Top-$k$ ($\lambda = 0.1$) | | $\lambda = 0.01$ | | Top-$k$ ($\lambda = 0.01$) | |
| | $\alpha = 1.5$ | $\alpha = 2.0$ | | | $\alpha = 1.5$ | $\alpha = 2.0$ | | |
|---|---|---|---|---|---|---|---|---|
| 0.3 | $86.81_{\pm0.93}$ | $87.87_{\pm0.90}$ | $85.97_{\pm0.96}$ | $85.97_{\pm0.96}$ | $87.41_{\pm0.91}$ | $87.04_{\pm0.93}$ | $87.26_{\pm0.92}$ | $87.26_{\pm0.92}$ |
| 0.7 | $86.96_{\pm0.93}$ | $88.17_{\pm0.89}$ | $85.97_{\pm0.96}$ | $85.97_{\pm0.96}$ | $85.67_{\pm0.97}$ | $86.88_{\pm0.93}$ | $88.10_{\pm0.89}$ | $88.10_{\pm0.89}$ |
| 1.0 | $86.35_{\pm0.95}$ | $87.11_{\pm0.92}$ | $85.97_{\pm0.96}$ | $85.97_{\pm0.96}$ | $84.99_{\pm0.98}$ | $83.93_{\pm1.01}$ | $85.44_{\pm0.97}$ | $85.44_{\pm0.97}$ |
| 1.5 | $87.19_{\pm0.92}$ | $86.58_{\pm0.94}$ | $85.97_{\pm0.96}$ | $85.97_{\pm0.96}$ | $82.94_{\pm1.04}$ | $83.70_{\pm1.02}$ | $80.14_{\pm1.10}$ | $80.14_{\pm1.10}$ |

Table 6: Accuracy on GSM8K for Phi-3-medium-4k-instruct using Bregman primal decoding ($\lambda \in \{0.001, 0.0001\}$, $\alpha \in \{1.5, 2.0\}$) and top-$k$ decoding, for various temperatures. For top-$k$, $k$ equals the averaged $k^*$ from primal decoding with matching temperature, $\mu$, and $\alpha$. Standard deviations are over 1000 bootstrap resamples.

| Temp | $\lambda = 0.001$ | | Top-$k$ ($\lambda = 0.001$) | | $\lambda = 0.0001$ | | Top-$k$ ($\lambda = 0.0001$) | |
| | $\alpha = 1.5$ | $\alpha = 2.0$ | | | $\alpha = 1.5$ | $\alpha = 2.0$ | | |
|---|---|---|---|---|---|---|---|---|
| 0.3 | $87.11_{\pm0.92}$ | $86.88_{\pm0.93}$ | $86.50_{\pm0.94}$ | $86.81_{\pm0.93}$ | $87.49_{\pm0.91}$ | $87.49_{\pm0.91}$ | $86.20_{\pm0.95}$ | $86.50_{\pm0.94}$ |
| 0.7 | $86.81_{\pm0.93}$ | $86.50_{\pm0.94}$ | $85.29_{\pm0.98}$ | $85.67_{\pm0.97}$ | $84.99_{\pm0.98}$ | $84.91_{\pm0.99}$ | $85.60_{\pm0.97}$ | $85.29_{\pm0.98}$ |
| 1.0 | $83.62_{\pm1.02}$ | $82.34_{\pm1.05}$ | $82.71_{\pm1.04}$ | $82.79_{\pm1.04}$ | $82.71_{\pm1.04}$ | $82.11_{\pm1.06}$ | $81.35_{\pm1.07}$ | $82.71_{\pm1.04}$ |
| 1.5 | $76.95_{\pm1.16}$ | $78.92_{\pm1.12}$ | $69.75_{\pm1.27}$ | $73.84_{\pm1.21}$ | $72.25_{\pm1.23}$ | $76.04_{\pm1.18}$ | $62.62_{\pm1.33}$ | $65.81_{\pm1.31}$ |

## H.4 Experiments for TriviaQA

Table 7 and 8 show accuracy on TriviaQA for LLaMA3.1-8B model. Here we choose 10% ($\approx$ 1800 questions) proportion of TriviQA validation dataset for evaluation.

Table 7: Accuracy on TriviaQA for LLaMA 3.1 8B using Bregman primal decoding ($\lambda \in \{0.1, 0.01\}$, $\alpha \in \{1.5, 2.0\}$) and top-$k$ decoding, for various temperatures. For top-$k$, $k$ equals the averaged $k^*$ from primal decoding with matching temperature, $\lambda$, and $\alpha$. Standard deviations are over 1000 bootstrap resamples.

| Temp | $\lambda = 0.1$ | | Top-$k$ ($\lambda = 0.1$) | | $\lambda = 0.01$ | | Top-$k$ ($\lambda = 0.01$) | |
|---|---|---|---|---|---|---|---|---|
| | $\alpha = 1.5$ | $\alpha = 2.0$ | $\alpha = 1.5$ | $\alpha = 2.0$ | $\alpha = 1.5$ | $\alpha = 2.0$ | $\alpha = 1.5$ | $\alpha = 2.0$ |
| 0.3 | $67.80_{\pm1.10}$ | $67.47_{\pm1.11}$ | $67.58_{\pm1.11}$ | $67.58_{\pm1.11}$ | $66.57_{\pm1.11}$ | $66.69_{\pm1.11}$ | $66.74_{\pm1.11}$ | $66.74_{\pm1.11}$ |
| 0.7 | $65.68_{\pm1.12}$ | $66.35_{\pm1.12}$ | $67.58_{\pm1.11}$ | $67.58_{\pm1.11}$ | $64.23_{\pm1.13}$ | $63.84_{\pm1.13}$ | $65.01_{\pm1.13}$ | $65.01_{\pm1.13}$ |
| 1.0 | $65.63_{\pm1.12}$ | $66.69_{\pm1.11}$ | $67.58_{\pm1.11}$ | $67.58_{\pm1.11}$ | $61.06_{\pm1.15}$ | $61.17_{\pm1.15}$ | $62.67_{\pm1.14}$ | $62.67_{\pm1.14}$ |
| 1.5 | $64.85_{\pm1.13}$ | $66.96_{\pm1.11}$ | $67.58_{\pm1.11}$ | $67.58_{\pm1.11}$ | $59.78_{\pm1.16}$ | $60.84_{\pm1.15}$ | $60.84_{\pm1.15}$ | $60.84_{\pm1.15}$ |

Table 8: Accuracy on TriviaQA for LLaMA 3.1 8B using Bregman primal decoding ($\lambda \in \{0.001, 0.0001\}$, $\alpha \in \{1.5, 2.0\}$) and top-$k$ decoding, for various temperatures. For top-$k$, $k$ equals the averaged $k^*$ from primal decoding with matching temperature, $\lambda$, and $\alpha$. Standard deviations are over 1000 bootstrap resamples.

| Temp | $\lambda = 0.001$ | | Top-$k$ ($\lambda = 0.001$) | | $\lambda = 0.0001$ | | Top-$k$ ($\lambda = 0.0001$) | |
|---|---|---|---|---|---|---|---|---|
| | $\alpha = 1.5$ | $\alpha = 2.0$ | $\alpha = 1.5$ | $\alpha = 2.0$ | $\alpha = 1.5$ | $\alpha = 2.0$ | $\alpha = 1.5$ | $\alpha = 2.0$ |
| 0.3 | $66.85_{\pm1.11}$ | $67.58_{\pm1.11}$ | $67.13_{\pm1.11}$ | $67.13_{\pm1.11}$ | $66.69_{\pm1.11}$ | $67.08_{\pm1.11}$ | $67.19_{\pm1.11}$ | $67.58_{\pm1.11}$ |
| 0.7 | $63.40_{\pm1.14}$ | $63.18_{\pm1.14}$ | $64.68_{\pm1.13}$ | $64.79_{\pm1.13}$ | $62.73_{\pm1.14}$ | $62.73_{\pm1.14}$ | $63.79_{\pm1.13}$ | $63.68_{\pm1.14}$ |
| 1.0 | $59.00_{\pm1.16}$ | $59.00_{\pm1.16}$ | $60.17_{\pm1.16}$ | $62.23_{\pm1.14}$ | $57.99_{\pm1.17}$ | $59.11_{\pm1.16}$ | $58.55_{\pm1.16}$ | $60.11_{\pm1.16}$ |
| 1.5 | $55.04_{\pm1.17}$ | $55.71_{\pm1.17}$ | $52.81_{\pm1.18}$ | $56.38_{\pm1.17}$ | $49.19_{\pm1.18}$ | $52.59_{\pm1.18}$ | $50.19_{\pm1.18}$ | $51.31_{\pm1.18}$ |

Table 9 and 10 show analogous accuracy results for Phi3-medium-4k-instruct on TriviaQA.

Table 9: Accuracy on TriviaQA for Phi-3-medium-4k-instruct using Bregman primal decoding ($\lambda \in \{0.1, 0.01\}$, $\alpha \in \{1.5, 2.0\}$) and top-$k$ decoding, for various temperatures. For top-$k$, $k$ equals the averaged $k^*$ from primal decoding with matching temperature, $\lambda$, and $\alpha$. Standard deviations are over 1000 bootstrap resamples.

| Temp | $\lambda = 0.1$ | | Top-$k$ ($\lambda = 0.1$) | | $\lambda = 0.01$ | | Top-$k$ ($\lambda = 0.01$) | |
|---|---|---|---|---|---|---|---|---|
| | $\alpha = 1.5$ | $\alpha = 2.0$ | $\alpha = 1.5$ | $\alpha = 2.0$ | $\alpha = 1.5$ | $\alpha = 2.0$ | $\alpha = 1.5$ | $\alpha = 2.0$ |
| 0.3 | $58.44_{\pm1.16}$ | $59.67_{\pm1.16}$ | $59.05_{\pm1.16}$ | $60.50_{\pm1.15}$ | $59.33_{\pm1.16}$ | $59.22_{\pm1.16}$ | $59.11_{\pm1.16}$ | $59.39_{\pm1.16}$ |
| 0.7 | $57.44_{\pm1.17}$ | $58.22_{\pm1.16}$ | $56.77_{\pm1.17}$ | $60.50_{\pm1.15}$ | $55.21_{\pm1.17}$ | $55.88_{\pm1.17}$ | $55.54_{\pm1.17}$ | $56.77_{\pm1.17}$ |
| 1.0 | $56.60_{\pm1.17}$ | $56.94_{\pm1.17}$ | $54.54_{\pm1.18}$ | $60.50_{\pm1.15}$ | $52.09_{\pm1.18}$ | $51.75_{\pm1.18}$ | $50.31_{\pm1.18}$ | $52.37_{\pm1.18}$ |
| 1.5 | $57.16_{\pm1.17}$ | $58.22_{\pm1.16}$ | $50.14_{\pm1.18}$ | $60.50_{\pm1.15}$ | $49.47_{\pm1.18}$ | $50.19_{\pm1.18}$ | $43.57_{\pm1.17}$ | $45.29_{\pm1.18}$ |

Table 10: Accuracy on TriviaQA for Phi-3-medium-4k-instruct using Bregman primal decoding ($\lambda \in \{0.001, 0.0001\}$, $\alpha \in \{1.5, 2.0\}$) and top-$k$ decoding, for various temperatures. For top-$k$, $k$ equals the averaged $k^*$ from primal decoding with matching temperature, $\lambda$, and $\alpha$. Standard deviations are over 1000 bootstrap resamples.

| Temp | $\lambda = 0.001$ | | Top-$k$ ($\lambda = 0.001$) | | $\lambda = 0.0001$ | | Top-$k$ ($\lambda = 0.0001$) | |
|---|---|---|---|---|---|---|---|---|
| | $\alpha = 1.5$ | $\alpha = 2.0$ | $\alpha = 1.5$ | $\alpha = 2.0$ | $\alpha = 1.5$ | $\alpha = 2.0$ | $\alpha = 1.5$ | $\alpha = 2.0$ |
| 0.3 | $59.72_{\pm1.16}$ | $58.61_{\pm1.16}$ | $59.44_{\pm1.16}$ | $59.22_{\pm1.16}$ | $59.83_{\pm1.16}$ | $59.39_{\pm1.16}$ | $59.44_{\pm1.16}$ | $59.44_{\pm1.16}$ |
| 0.7 | $54.82_{\pm1.17}$ | $54.04_{\pm1.18}$ | $53.70_{\pm1.18}$ | $54.60_{\pm1.18}$ | $54.54_{\pm1.18}$ | $54.43_{\pm1.18}$ | $56.21_{\pm1.17}$ | $54.71_{\pm1.18}$ |
| 1.0 | $48.13_{\pm1.18}$ | $49.19_{\pm1.18}$ | $49.58_{\pm1.18}$ | $50.64_{\pm1.18}$ | $48.69_{\pm1.18}$ | $48.58_{\pm1.18}$ | $48.64_{\pm1.18}$ | $48.64_{\pm1.18}$ |
| 1.5 | $42.51_{\pm1.17}$ | $44.18_{\pm1.17}$ | $39.55_{\pm1.15}$ | $42.67_{\pm1.17}$ | $38.22_{\pm1.15}$ | $39.94_{\pm1.16}$ | $36.04_{\pm1.13}$ | $37.72_{\pm1.14}$ |

## H.5 Adaptivity

In this section, we consider the adaptivity of primal decoding by presenting the mean, standard deviation and entropy of the $k^*$ chosen by our method during evaluation on GSM8K and TriviaQA datasets.

In Table 11, we show the average $k^*$ values (and their values rounded to the nearest integer) selected by primal Bregman decoding on GSM8K with LLaMA 3.1 8B for various temperatures, $\alpha$, and $\lambda$. Table 12 shows corresponding standard deviation and entropy.

Table 11: Mean (and rounded) average $k^*$ values on GSM8K with LLaMA 3.1 8B for various temperatures, $\alpha$, and $\lambda$.

| Temp | $\lambda = 0.1$ | | $\lambda = 0.01$ | | $\lambda = 0.001$ | | $\lambda = 0.0001$ | |
|---|---|---|---|---|---|---|---|---|
| | $\alpha = 1.5$ | $\alpha = 2.0$ | $\alpha = 1.5$ | $\alpha = 2.0$ | $\alpha = 1.5$ | $\alpha = 2.0$ | $\alpha = 1.5$ | $\alpha = 2.0$ |
| 0.3 | 1.2231(1) | 1.1537 (1) | 1.6201 (2) | 1.4453 (1) | 2.1274 (2) | 1.7964 (2) | 2.8578 (3) | 2.2112 (2) |
| 0.7 | 1.2295 (1) | 1.1554 (1) | 1.6689 (2) | 1.4794 (1) | 2.3193 (2) | 1.9048 (2) | 3.2554 (3) | 2.4974 (2) |
| 1.0 | 1.2287 (1) | 1.1594 (1) | 1.7519 (2) | 1.5048 (2) | 2.7231 (3) | 2.0234 (2) | 4.6926 (5) | 3.0924 (3) |
| 1.5 | 1.2331 (1) | 1.1566 (1) | 1.8106 (2) | 1.5189 (2) | 4.1842 (4) | 2.4067 (2) | 14.2539 (14) | 5.6002 (6) |

Table 12: Standard deviation (and entropy) of average $k^*$ values on GSM8K with LLaMA 3.1 8B for various temperatures, $\alpha$, and $\lambda$.

| Temp | $\lambda = 0.1$ | | $\lambda = 0.01$ | | $\lambda = 0.001$ | | $\lambda = 0.0001$ | |
|---|---|---|---|---|---|---|---|---|
| | $\alpha = 1.5$ | $\alpha = 2.0$ | $\alpha = 1.5$ | $\alpha = 2.0$ | $\alpha = 1.5$ | $\alpha = 2.0$ | $\alpha = 1.5$ | $\alpha = 2.0$ |
| 0.3 | 0.46 (0.82) | 0.36 (0.62) | 1.07 (1.55) | 0.77 (1.28) | 1.89 (2.08) | 1.31 (1.77) | 3.11 (2.58) | 2.00 (2.16) |
| 0.7 | 0.47 (0.84) | 0.36 (0.62) | 1.12 (1.62) | 0.80 (1.34) | 2.21 (2.24) | 1.47 (1.89) | 3.98 (2.78) | 2.53 (2.37) |
| 1.0 | 0.47 (0.84) | 0.37 (0.63) | 1.23 (1.72) | 0.83 (1.38) | 3.03 (2.49) | 1.65 (2.00) | 7.31 (3.21) | 3.69 (2.69) |
| 1.5 | 0.47 (0.85) | 0.36 (0.63) | 1.30 (1.79) | 0.84 (1.40) | 5.37 (3.13) | 2.19 (2.32) | 18.01 (4.04) | 7.77 (3.51) |

Table 13-14 show analougous adaptivity results for Qwen2.5-14B-Instruct.

Table 13: Mean (and rounded) average $k^*$ values on GSM8K with Qwen2.5-14B-Instruct for various temperatures, $\alpha$, and $\lambda$.

| Temp | $\lambda = 0.1$ | | $\lambda = 0.01$ | | $\lambda = 0.001$ | | $\lambda = 0.0001$ | |
|---|---|---|---|---|---|---|---|---|
| | $\alpha = 1.5$ | $\alpha = 2.0$ | $\alpha = 1.5$ | $\alpha = 2.0$ | $\alpha = 1.5$ | $\alpha = 2.0$ | $\alpha = 1.5$ | $\alpha = 2.0$ |
| 0.3 | 1.0973(1) | 1.0660(1) | 1.4899(1) | 1.3425(1) | 2.7614(3) | 1.9317(2) | 5.4537(5) | 3.1726(3) |
| 0.7 | 1.1010(1) | 1.0672(1) | 1.5043(2) | 1.3534(1) | 2.7778(3) | 1.9522(2) | 5.5047(6) | 3.1911(3) |
| 1.0 | 1.1000(1) | 1.0666(1) | 1.5171(2) | 1.3591(1) | 2.7985(3) | 1.9723(2) | 5.5603(6) | 3.2493(3) |
| 1.5 | 1.1008(1) | 1.0662(1) | 1.5211(2) | 1.3628(1) | 2.8761(3) | 2.0028(2) | 5.7831(6) | 3.3285(3) |

Table 14: Standard deviation (and entropy) of average $k^*$ values on GSM8K with Qwen2.5-14B-Instruct under $\lambda = 0.0001$ and varying temperatures.

| Temp | $\alpha = 1.5$ | $\alpha = 2.0$ |
|---|---|---|
| 0.3 | 10.75 (2.81) | 4.88 (2.26) |
| 0.7 | 10.71 (2.86) | 4.85 (2.29) |
| 1.0 | 10.70 (2.90) | 4.88 (2.34) |
| 1.5 | 10.75 (3.03) | 4.90 (2.42) |

Table 15-16 show analougous adaptivity results for Phi-3-medium-4k-instruct.

Table 15: Mean (and rounded) average $k^*$ values on GSM8K with Phi-3-medium-4k-instruct for various temperatures, $\alpha$, and $\mu$.

| Temp | $\lambda = 0.1$ | | $\lambda = 0.01$ | | $\lambda = 0.001$ | | $\lambda = 0.0001$ | |
|---|---|---|---|---|---|---|---|---|
| | $\alpha = 1.5$ | $\alpha = 2.0$ | $\alpha = 1.5$ | $\alpha = 2.0$ | $\alpha = 1.5$ | $\alpha = 2.0$ | $\alpha = 1.5$ | $\alpha = 2.0$ |
| 0.3 | 1.4048(1) | 1.2609(1) | 2.4123(2) | 1.9287(2) | 4.7186(5) | 3.1299(3) | 8.6473(9) | 5.2889(5) |
| 0.7 | 1.4074(1) | 1.2601(1) | 2.4337(2) | 1.9409(2) | 4.6706(5) | 3.1307(3) | 8.6958(9) | 5.3697(5) |
| 1.0 | 1.4073(1) | 1.2603(1) | 2.4541(2) | 1.9364(2) | 4.7772(5) | 3.1792(3) | 8.8501(9) | 5.4394(5) |
| 1.5 | 1.4098(1) | 1.2575(1) | 2.4667(2) | 1.9498(2) | 4.9289(5) | 3.2335(3) | 9.4782(9) | 5.6113(6) |

Table 16: Standard deviation (and entropy) of average $k^*$ values on GSM8K with Phi-3-medium-4k-instruct under $\lambda = 0.0001$ for varying temperatures and $\alpha$.

| Temp | $\alpha = 1.5$ | $\alpha = 2.0$ |
|---|---|---|
| 0.3 | 12.09 (3.83) | 6.77 (3.32) |
| 0.7 | 12.01 (3.89) | 7.23 (3.61) |
| 1.0 | 11.98 (3.98) | 6.74 (3.45) |
| 1.5 | 11.79 (4.24) | 7.29 (3.79) |

In Table 17, we show the average $k^*$ values (and their values rounded to the nearest integer) selected by primal Bregman decoding on TriviaQA with LLaMA 3.1 8B for various temperatures, $\alpha$, and $\lambda$. Table 18 shows corresponding standard deviation and entropy.

Table 17: Mean (and rounded) average $k^*$ values on TriviaQA with LLaMA 3.1 8B for various temperatures, $\alpha$, and $\lambda$.

| Temp | $\lambda = 0.1$ | | $\lambda = 0.01$ | | $\lambda = 0.001$ | | $\lambda = 0.0001$ | |
| | $\alpha = 1.5$ | $\alpha = 2.0$ | $\alpha = 1.5$ | $\alpha = 2.0$ | $\alpha = 1.5$ | $\alpha = 2.0$ | $\alpha = 1.5$ | $\alpha = 2.0$ |
| --- | --- | --- | --- | --- | --- | --- | --- | --- |
| 0.3 | 1.1536(1) | 1.1452(1) | 1.9135(2) | 1.5291(2) | 3.4193(3) | 2.5753(3) | 6.9406(7) | 4.5149(5) |
| 0.7 | 1.2265(1) | 1.1275(1) | 2.0109(2) | 1.6265(2) | 3.8877(4) | 2.7593(3) | 8.8845(9) | 5.1892(5) |
| 1.0 | 1.2138(1) | 1.1324(1) | 2.0273(2) | 1.6818(2) | 3.9715(4) | 2.9759(3) | 8.4552(8) | 5.7381(6) |
| 1.5 | 1.2013(1) | 1.1384(1) | 2.0289(2) | 1.7032(2) | 4.1749(4) | 2.9398(3) | 8.4399(8) | 5.5166(6) |

Table 18: Standard deviation (and entropy) of average $k^*$ values on TriviaQA with LLaMA 3.1 8B for various temperatures, $\alpha$, and $\lambda$.

| Temp | $\lambda = 0.1$ | | $\lambda = 0.01$ | | $\lambda = 0.001$ | | $\lambda = 0.0001$ | |
| | $\alpha = 1.5$ | $\alpha = 2.0$ | $\alpha = 1.5$ | $\alpha = 2.0$ | $\alpha = 1.5$ | $\alpha = 2.0$ | $\alpha = 1.5$ | $\alpha = 2.0$ |
| --- | --- | --- | --- | --- | --- | --- | --- | --- |
| 0.3 | 0.41 (0.65) | 0.35 (0.60) | 1.37 (1.90) | 0.86 (1.42) | 3.65 (2.85) | 2.09 (2.42) | 10.36 (3.63) | 5.35 (3.28) |
| 0.7 | 0.48 (0.83) | 0.33 (0.55) | 1.44 (2.00) | 0.93 (1.56) | 4.24 (3.09) | 2.20 (2.53) | 12.18 (4.10) | 5.98 (3.56) |
| 1.0 | 0.47 (0.81) | 0.34 (0.56) | 1.42 (2.01) | 0.98 (1.63) | 4.42 (3.03) | 2.43 (2.68) | 12.07 (3.77) | 6.54 (3.68) |
| 1.5 | 0.46 (0.78) | 0.35 (0.58) | 1.42 (2.01) | 1.00 (1.66) | 5.07 (3.02) | 2.46 (2.62) | 12.90 (3.34) | 7.18 (3.35) |

Table 19-20 show analougous adaptivity results for Phi-3-medium-4k-instruct on TriviaQA.

Table 19: Mean (and rounded) average $k^*$ values on TriviaQA with Phi-3-medium-4k-instruct for various temperatures, $\alpha$, and $\lambda$.

| Temp | $\lambda = 0.1$ | | $\lambda = 0.01$ | | $\lambda = 0.001$ | | $\lambda = 0.0001$ | |
| | $\alpha = 1.5$ | $\alpha = 2.0$ | $\alpha = 1.5$ | $\alpha = 2.0$ | $\alpha = 1.5$ | $\alpha = 2.0$ | $\alpha = 1.5$ | $\alpha = 2.0$ |
| --- | --- | --- | --- | --- | --- | --- | --- | --- |
| 0.3 | 1.7393(2) | 1.4142(1) | 3.6184(4) | 2.8184(3) | 9.2976(9) | 5.2226(5) | 18.7026(19) | 10.4901(10) |
| 0.7 | 1.7148(2) | 1.4288(1) | 3.6134(4) | 2.6381(3) | 8.4512(8) | 4.8061(5) | 16.8627(17) | 9.3718(9) |
| 1.0 | 1.7348(2) | 1.4216(1) | 3.6840(4) | 2.6050(3) | 8.3500(8) | 4.8924(5) | 16.7567(17) | 9.6411(10) |
| 1.5 | 1.6687(2) | 1.4378(1) | 3.6081(4) | 2.6601(3) | 8.6007(9) | 5.1906(5) | 18.2735(18) | 9.7162(10) |

Table 20: Standard deviation (and entropy) of average $k^*$ values on TriviaQA with Phi-3-medium-4k-instruct for various temperatures, $\alpha$, and $\lambda$.

| Temp | $\lambda = 0.1$ | | $\lambda = 0.01$ | | $\lambda = 0.001$ | | $\lambda = 0.0001$ | |
| | $\alpha = 1.5$ | $\alpha = 2.0$ | $\alpha = 1.5$ | $\alpha = 2.0$ | $\alpha = 1.5$ | $\alpha = 2.0$ | $\alpha = 1.5$ | $\alpha = 2.0$ |
| --- | --- | --- | --- | --- | --- | --- | --- | --- |
| 0.3 | 0.87 (1.43) | 0.49 (0.98) | 2.84 (2.65) | 1.76 (2.26) | 8.19 (4.18) | 3.99 (3.32) | 16.54 (5.06) | 8.88 (4.33) |
| 0.7 | 0.87 (1.41) | 0.49 (0.99) | 2.69 (2.76) | 1.70 (2.22) | 7.50 (4.16) | 3.78 (3.28) | 15.26 (5.16) | 8.38 (4.31) |
| 1.0 | 0.87 (1.43) | 0.49 (0.98) | 2.68 (2.82) | 1.65 (2.24) | 7.04 (4.22) | 3.62 (3.37) | 13.81 (5.27) | 7.75 (4.46) |
| 1.5 | 0.84 (1.40) | 0.50 (0.99) | 2.58 (2.84) | 1.63 (2.27) | 6.60 (4.30) | 3.58 (3.45) | 13.94 (5.37) | 7.51 (4.51) |

