# OpenReview forum: "Foundations of Top-$k$ Decoding for Language Models"
_NeurIPS.cc/2025/Conference — NeurIPS 2025 poster_

### Official Review · Reviewer_rK4x · 2025-06-14

**Clarity:** 3
**Significance:** 3
**Originality:** 3
**Rating:** 5
**Confidence:** 3

**Summary:**

This work establishes a theoretical framework for (generalized) top-$k$ decoding using Bregman divergence. Specifically, it introduces a general optimization objective for large language model (LLM) decoding that promotes sparsity via the $\ell_0$ norm and investigates separable Bregman divergences. The resulting solution extends the top-$k$ decoding strategy, with a notable application being the adaptive selection of $k$ based on the target distribution $p$.

**Questions:**

Section 3.1 is overly dense and lacks intuitive explanations for the proposed setup. The authors should include a clear, high-level overview to provide context and intuition. Additionally, the role of Equation (5) in the dual renormalization mapping is unclear; a detailed explanation or example illustrating its implications would improve accessibility.

 In Section 3.2, Theorem 3.2 appears to say that top-$\hat{k}$ is an optimal solution for Equation (2). If so, the introduction of $\mathrm{cost}(k)$ in Equation (6) is confusing. Why is this cost term necessary, and why wouldn’t a naive search over all $\hat{k}$ suffice? The authors should clarify the purpose and necessity of $\mathrm{cost}(k)$.

The paper does not discuss scenarios where dual decoding might be preferred over primal decoding. The authors should highlight specific use cases or advantages of dual decoding, including any theoretical or practical benefits, to strengthen the contribution.

Are there any experimental or theoretical results on applying this new metric to beam search? Regarding the adaptivity property, I wonder if there's any connection between this and other methods such as min-p and top-p?

**Ethical Concerns:**

["NO or VERY MINOR ethics concerns only"]

**Final Justification:**

The paper provides good theoretical contribution to top-k decoding, which is interesting for the community and LLM development.

**Limitations:**

Figure 1 should include a contour plot alongside the 3D visualization to enhance clarity and interpretation.

The final paragraph of Section 2.2 notes that when $\lambda > 0$, a sparsity constraint is induced, which is a well-known property in the convex optimization literature. However, the unique aspect here appears to be the focus on the top-$k$ values within $p$. This distinction should be emphasized, with specific, non-trivial details (e.g., novel computational or theoretical implications) clearly articulated to highlight the contribution.

The adaptivity property should be highlighted in the introduction.

Also see Questions.

**Paper Formatting Concerns:**

No Formatting Concerns.

**Quality:**

3

**Strengths And Weaknesses:**

Strengths:
- The paper provides detailed studies of top-k decoding through the lens of Bergman divergence.

- The empirical results demonstrate good performance compared to the baselines.

- The adaptivity property of the provided method is interesting.

Weakness:
- Some discussion are not well-motivated until very later sections.

- Discussion on practical application is lacking. (see questions)

---

> ### Author Rebuttal · Authors · 2025-07-31
>
> Thank you so much for your careful reading and interesting questions. As a high-level comment, we'll make sure to incorporate, your suggestions for improving the readability of the manuscript (including: de-congesting parts that are dense (Section 3.1, 3.3); highlighting adaptivity in the intro; highlighting the specific optimization challenges we solve; discussing dual vs primal decoding; adding contour plot lines; etc). While we've put a lot of effort into making the presentation accessible and intuitive, the page limits have forced us to exclude some bits of intuition --- the future extra page will let us to bring them back into the manuscript.
>
> **Q:** Section 3.1 overly dense, lacks intuitive explanations. Include clear, high-level overview with context/intuition. Role of Equation (5) in dual renormalization mapping unclear; detailed explanation/example of its implications?
>
> **A:** We are committed to refining and making more accessible the presentation of our results in this subsection. As briefly described in lines 156-160 at the top of this subsection, the focus of this subsection is on projecting, i.e., turning the selected $k$ tokens' probabilities into a valid probability distribution. This is necessary as the sum of the probabilities of the selected tokens is generally less than $1$, and so the remaining probability mass (coming from the non-selected tokens) needs to be somehow "added back in". The rest of the subsection defines conditions/assumptions under which these projection maps $T$ and $T^*$ (for the primal and dual Bregman rules) are unique and well-defined, and gives formulas for computing these projections --- an explicit one in the primal case (Equation 4) and an implicit one in the dual case (Equation 5). Both these formulas are defined in terms of the monotonic function $f$ which specifies the Bregman divergence used to generate the decoder. We will make sure to include this additional description in the updated version.
>
> For Equation (5) specifically, its role is to define, for a *dual Bregman* decoder, the transformation that the probability $x_i$ of each individual selected token $i$ will undergo when renormalizing the top-$k$ tokens' probabilities to a valid distribution (in parallel to how Equation (4) defines the same transformation for a *primal* Bregman decoder). This equation for each $i$ is implicit, as both of its sides feature [T^\*_\phi(x)] _i.
>
> However, as we rigorously show under our dual validity assumptions, it still uniquely specifies [T^\*_\phi(x)] _i, whose value can be found via a simple root-finding procedure.
>
> For additional intuition about this set of equations, we can make sure to explicitly mention the following structural properties in the updated version:
> (1) This dual transformation redistributes the probability mass of the non-top-$k$ tokens among the top-$k$ tokens in a way that increases the probability of each top-$k$ token: that is, [T^\*_\phi(x)]_i $\geq x_i$ for all $i \in [k]$;
> (2) The parameter $\nu^\*$ governs how much is added to each $x_i$ by the transformation (the larger $\nu^\*$, the larger each transformed $x_i$);
> (3) Thus, between $\nu = 0$ (when no mass gets added to any $x_i$) and $\nu \to \infty$ (when very large mass is added to each selected token's probability), there exists a (unique) intermediate value $\nu^\*$ at which the transformed probabilities sum up to $1$, thus forming the desired valid distribution over top-$k$ tokens. This intermediate value $\nu^\*$ can itself be identified via simple root-finding.
>
> **Q:** Theorem 3.2 says top-$k$
>  is optimal solution for Equation (2). Why is Equation (6) cost term necessary, why wouldn’t naive search over all k
>  suffice? Purpose and necessity of cost($k$)?
>
> **A:** Indeed, as Theorem 3.2 establishes, the optimal solution to Equation (2) is to select *some number* $k$ of top tokens (and then renormalize), so the naive search over all $k$ would suffice to identify the optimal number of top tokens. However, the challenge that we address in Section 3.3 is that an exhaustive search over all possible $k \in [1, V]$ is computationally prohibitive, as $V$ is large and each tried value of $k$ requires projecting the top-$k$ tokens (and computing the objective value). Practically, one could settle for grid search over some subset of values of $k$, but this will not give provable theoretical guarantees.
>
> So instead we exhibit a surprising extra structure in the optimization problem: Namely, that *it is convex with respect to k*, and thus one only needs to try $O(\log V)$, rather than all $V$, candidate values of $k$ to find the optimal one! The notation $cost(k)$, thus, is to help cleanly formalize this convexity --- it makes explicit the dependence of the objective value on the choice of $k$ (while keeping implicit the token distribution $p$).
>
> **Q:** Scenarios where dual might be preferred over primal? Specific use cases/advantages of dual decoding, theoretical or practical benefits?
>
> **A:** The question of when dual decoding can be preferred over primal one is indeed interesting and subtle. While we currently already provide some visual comparisons between the renormalizations induced in the primal and dual case  --- such as in the dedicated subsections G in the Appendix (and e.g. on figures 4 and 6) --- we will be happy to discuss their implications more prominently in the main part. (That said, we believe the subtle nature of the comparison merits its own detailed investigation beyond the scope of this paper.)
> Below are some initial considerations for when dual decoding may be preferred, which we are happy to include in the updated version and hope will guide future work:
>
> (A) *Inducing larger range of optimal $k$:* Dual decoders can induce a wider range of k* than primal ones: e.g. on the plot of the objective as a function of k in Figure 6, one can notice that the minimum is located in a wider span of locations for the dual (k* $\in [1, 40]$) than for the primal (k* $\in [1, 20]$) methods. Thus, when a broader adaptivity range (i.e., how many tokens get selected) is desired, dual decoding may prove favorable.
>
> (B) *Less effort in tuning k:* Also in the context of choosing optimal $k^*$, choosing dual decoding may be preferable when one *doesn't* want to carefully tune the optimal $k$: As can be seen in Figure 6, the minima of the objective are flatter and misspecifying $k$ is of less impact than it would be for primal decoding.
>
> (C) *Tuning $\alpha$:* The $\alpha$-parameterization of the primal and dual families exhibits different degrees of sensitivity to changes in $\alpha \in (1, \infty)$ --- e.g. see Figure 4 for how the top-token probability changes with $\alpha$. Thus, an experimenter who knows an approximate range/location (but not the exact value) of $\alpha$, should prefer the dual decoding whenever it has lower sensitivity to $\alpha$ in that range, to cut down on hyperparameter tuning and to ensure robustness to misspecification of $\alpha$.
>
> We point out that the above considerations are not distribution-free (i.e. they depend on the underlying distribution $p$, unlike the main framework results that we develop in this paper in a distribution-free manner) --- and hence will require task-specific quantification and careful empirical studies beyond the scope of the present foundations paper.
>
> **Q:** Experimental or theoretical results on applying this metric to beam search? Regarding adaptivity, any connection to methods like min-p and top-p?
>
> **A:** We agree that improving the efficiency of beam search by performing decoding with our new methods is an interesting question for future work. For instance, empirically, the structural property of the Bregman parameter $\alpha$ --- that it controls the diversity of the decoding --- may result in *controllably diverse* sequences in the beam, helping better explore options within the beam. Mathematically, in this paper we focus on per-token decoding (which already presents interesting challenges to resolve), and building this up to a theory that would provide insights into beam search decoding will require one to develop models of how local, per-token decoding properties translate to global, sequence-level decoding properties.
>
> Regarding the adaptivity property, this is indeed an insightful question --- indeed there is a (non-trivial) connection/extension that one can build to make our framework cover methods such as top-$p$ and min-$p$. The structural reason for which this is possible is that just like top-$k$, these decoding methods also follow the template of selecting some number $k'$ of top tokens, *which depends on the underlying distribution over tokens and hence is adaptive*, and then renormalizing them back to a proper distribution. The difference is that the parameter $k$ in these other methods is reparameterized in terms of the total desired probability mass (for top-$p$) or in terms of the desired minimum included probability (for min-$p$).
>
> We have not explicitly detailed this extension in this paper (as we wanted to make sure the presentation stays focused for the reader's benefit), but our theory does address both fundamental questions: how to *adaptively* pick the number of tokens to keep, and how to renormalize them back to a proper distribution once they are picked --- which are the essential considerations for top-$k$ and top-$p$ and min-$p$ likewise.
> We will mention the development of the full theory as a direction for future work in our paper.

---

> > ### Comment · Reviewer_rK4x · 2025-08-03
> >
> > Thank you for your concern, I updated my score to 5

---

> > > ### Author Response · Authors · 2025-08-03
> > >
> > > Thank you so much for your valuable feedback and for supporting our work. Your expertise is highly appreciated. Sincerely, The authors.

---

### Official Review · Reviewer_4MPc · 2025-07-01

**Clarity:** 3
**Significance:** 2
**Originality:** 2
**Rating:** 3
**Confidence:** 4

**Summary:**

The authors considered the well-known top-k decoding rule, and provided an optimization interpretation of the rule. It is shown that it can be viewed as the solution of a l_0-regularized Bregman divergence minimization problem. Based on this interpretation, the authors specialized it to a set of potential functions in the Bregman divergence, which induce different decoding rules other than the top-k decoding.

**Questions:**

1. l-1 norm is often used as a surrogate for l_0 norm. In the setting considered here, will replacing l-0 with l_1 result in the same top-k decoding rule?

2. If a goal of adopting the Bregman divergence is for its generality, then the following question appears important: Under the same regularized minimization setup, if the decoding rule is greedy, does it imply that the divergence must be a Bregman divergence?

3. Assumption 3.1 seems redundant, since otherwise, Bregman divergence won't be well defined?

4. Why is the class of permutation-equivariant maps important? It is true that the solutions given her all belong to this class, but it is not clear why the restriction of "permutation-equivariant" plays a role anywhere?

**Ethical Concerns:**

["NO or VERY MINOR ethics concerns only"]

**Final Justification:**

In general, I feel the paper provides a reassuring result and has some value, but I didn't find it to be exciting. Moreover, I'm not convinced that the generality via Bregman divergence can bring any practical benefits. I have taken into account the rebuttal/discussion and decided to keep the score.

**Limitations:**

some discussion is found in the section 7.

**Quality:**

3

**Strengths And Weaknesses:**

Strengths:

1. It is reassuring to see the top-k decoding rule has an optimization interpretation.
2. The theory development appears rigorous.

Weakness: Though the results are reassuring, the theory development seems of limited use for decoding rules in the practice.

1. The interpretation is somewhat obvious. In practice, the top-k decoding rule is mostly motivated by the simplicity of the computation, particularly in its logit form. It is then not surprising that there is an optimization problem that can yield the same solution.

2. Using the Bregman divergence as the measure of divergence lends some generality to the result, but also makes it less interpretable. The main intuition seems to be already clear if we consider KL divergence or the squared l_2 norm. Though this generality leads to other decoding rules, these rules seem to be purely of mathematical interest, but of little practical advantage.

3. The issue of simple computation is not considered, which is in fact the main motivation for the top-k decoding rule. The new decoding rules appear to lose this obvious advantage, and there are no clear benefits in their eventual performance.

4. The numerical evaluation is relatively weak: What is the main point of having these evaluations? These decoding rules do not lead to better performance, nor are they computationally more efficient. It is not clear what the readers can learn from these numerical results.

---

> ### Author Rebuttal · Authors · 2025-07-31
>
> Thank you for your effort in evaluating this manuscript. Before answering your concrete questions, we’d like to comment on this paper's goals: It aims to provide a theoretical foundation for top-$k$ decoding, to achieve the following:
> (1) Through a rigorous optimization/loss-theoretic framework, explain the critical design choices inherent in top-$k$ decoding, including: How to adaptively choose the optimal $k$? How to re-normalize the chosen probabilities of the top-$k$ tokens? etc.;
> (2) Illustrate the framework on the rich class of Bregman losses (as elaborated below, this let us elicit many novel decoding intuitions/insights);
> (3) Identify, and solve, the mathematical challenges that arise when rigorously establishing basic properties of a decoding method: namely, what we call its ``greedy property'', and the convexity of the optimization problem for $k$;
> (4) Confirm, through a few realistic simulations, that decoding rules generated by this framework are sensible (and indeed, we would argue, promising) for further study.
>
> Now, we'll address your points in the weaknesses and questions sections:
>
> ---
>
> **Weaknesses:**
>
> **Q:** Interpretation is obvious. Top-k: easy computation. Optimization problem not surprising.
>
> **A:** We'd like to begin by remarking that we do not wish to claim that our general sparse optimization formulation is extremely "surprising", but that it's:
>
> (1) Very natural, as it formalizes and expands on the major motivation behind top-$k$: Sparsity.
>
> (2) Novel and fills an important knowledge gap: There are currently very few theoretical investigations into the foundations of LLM decoding in general, with the field being predominantly empirical and not resting on rigorous foundations.
>
> To the point of our framework being "somewhat obvious": While the general L0 formulation is natural as explained above, though we still wouldn't call it obvious, the tractable structures that we uncover when instantiating this optimization problem (with the expansive class of Bregman divergences) are not at all obvious.
>
> Indeed, we start with a regularized problem where the regularization is the nonsmooth + nonconvex L0 pseudonorm, and in which the divergence objective itself is not even necessarily convex (e.g. in the dual Bregman case). In these circumstances, our finding about the presence of a doubly favorable optimization landscape, where:
> (1) One only has to check linearly many sparsity patterns out of exponentially many (the "greedy" property), and
> (2) The optimization of sparsity norm $k$ is discretely convex in $k$ --- is arguably both surprising and deep. Proving greediness and, even more so, $k$-convexity, required us to develop nontrivial and multifaceted proof techniques (which we feel will be useful even beyond this setting).
>
> **Q:** KL and $l_2$ enough, Bregman less interpretable.
>
> **A:** From our perspective, the generalization of vanilla top-$k$ decoding (KL) to other Bregman divergences actually makes top-$k$ *more*, not less, interpretable.
>
> For instance, the *general interpretation of renormalization as a simplex projection* is an insightful way to think about this ingredient of generalized top-$k$ decoding, which we believe adds rather than subtracts interpretability.
>
> Revisiting your assessment that the main intuition is already clear from KL and L2 cases, we would like to argue the opposite. Looking at these two cases, it may not be obvious at all if these decoders are just two distinct heuristic choices --- where you either normalize top-$k$ by dividing by the sum, or you add back the remaining mass in equal chunks to all top-$k$ tokens --- or if there is a sense in which one can smoothly interpolate between them, creating "intermediate decoders".
>
> In fact, our primal and dual families provide two distinct ways to put KL and L2 decoding on a continuum --- with the parameter $\alpha$ (with $\alpha=1$ giving KL and $\alpha=2$ giving L2) being nicely interpretable: as $\alpha$ increases, our family of Bregman decodings ranges, for any fixed $k$, from "less diverse" to "more diverse" (see also more on this below in this answer). This e.g. lets us easily read off from this continuous family that L2-decoding ($\alpha=2$) is more diverse than vanilla top-$k$ ($\alpha=1$), and even that there is an "intermediately diverse" closed-form decoding for $\alpha=1.5$.
>
> The limiting cases are also insightful. When $\alpha \to \infty$, our insight (see L. 259) says that not only do decoders not diverge in this case as one might suppose, but they in fact converge to the beautiful *water-filling* limit in which the remaining mass is interpreted as "water" that gets "poured" on the lowest-probability tokens out of top-$k$. On the opposite end, if one takes $\alpha \to -\infty$, one gets (see L. 256) another clean, "least diverse", limit in which the remaining mass gets placed "on top of" the top token.
>
> In the context of decoding, the most diverse option is commonly thought of as a uniform distribution over the $k$ tokens, and the least diverse option as simply putting all mass on the top token.
> By contrast, the limits just described are substantially different.
>
> **Q:**  New rules only of theoretical, not practical interest.
>
> **A:**  We do believe that our empirical evaluations show some practical promise for future work within our framework --- please see the response to Question 5 for a discussion of our empirical evaluations.
>
> **Q:**  Simple computation isn't considered but motivates top-k; new rules not simple, no clear performance gains.
>
> **A:**  We would like to clarify that we did in fact carefully consider the issue of simple computation, from both theoretical and empirical angles, making sure that the new decoding rules do not lose the scalability that's inherent to vanilla top-$k$ decoding.
>
> We have established the discrete $k$-convexity of the optimization problem; please see Section 3.3.
> These theorems demonstrate that rather than having to naively grid search over all $V$ (vocab size) possible values of $k$, it provably suffices to check (via binary search) only $O(\log V)$ of them. This is a drastic improvement from the overhead on computing the best $k$ that one would expect a-priori.
>
> What's more, in fact even $O(\log V)$ tries are unnecessary in typical practice, since via exponential search, we would only need $O(\log k^*)$ tried values of $k$ (see responses to Reviewers 1 and 2), which is barely any overhead at all over the vanilla top-$k$ decoding, given that in practice, optimal $k$ is far smaller (on the order of at most 50 or so) than the vocab size.
>
> On the empirical side, for important special cases of Bregman decoders, such as the choices $\alpha \in \{1.5, 2\}$ that we tested in the empirical evaluations, we actually match the speed of vanilla top-$k$ decoding, benefiting from vectorized computations of projections. This is due to the closed forms for renormalization in those cases, as well due to improved search for the optimal $k^*$.
>
> Moreover, we reiterate that we do see some promise in the empirical results, as explained in the next answer.
>
> **Q:**  Numerical evaluation is weak?
>
> **A:** We position this paper as studying the theoretical foundations of top-k decoding (with the corresponding title, exposition, and NeurIPS category). Thus, we view
> the purpose of our simulations as simply to confirm that our theory leads to decoders that are both meaningful and reasonable, and that further hold promise for future study. We believe our evaluations clear that bar; and we believe our approach matches the standards of other theoretical papers published in NeurIPS.
>
> Regarding the performance of our methods, we would like to clarify that there are in fact a few promising aspects: (1) $\alpha=2$ broadly achieves better perplexity in our experiments, and repetitiveness reduction is observed.
> Moreover, (2) on GSM-8k the performance of our decoders appears more robust across temperatures (i.e., accuracy doesn't degrade as badly as the standard top-$k$).
>
> We have already mentioned these claims in our paper (lines 303 and 310), but didn't emphasize them too much, due to our focus on the theoretical foundations. However, we will be happy to add some additional emphasis to the claims.
>
> In the manuscript we made sure to appropriately caution the reader to not over-interpret these preliminary results but instead to view them as a starting point towards a fuller evaluation, on more models, tasks and metrics.
>
> ---
>
> **Questions:**
>
> **Q:** Replace $l_0$ with $l_1$?
>
> **A:** The $l_1$ norm of a probability vector (which $\hat{p}$ must be) is always 1 --- so this regularization would not actually lead to a top-$k$-sparse optimization problem (e.g. for Bregman losses it would just output the original probability vector back).
>
> **Q:** If decoding rule is greedy, must its divergence be Bregman?
>
> **A:** The greedy property is not limited to this class---e.g. from some additional investigation (performed after submitting the paper), we can also obtain greedy decoders by minimizing certain $f$-divergences.
>
> In general, the greedy property appears to be a natural one from intuitive considerations, and we expect it to hold for a variety of reasonable loss functions.
>
> **Q:** Assumption 3.1 redundant?
>
> **A:**  Assumption 3.1 adds the extra requirement of first-order smoothness (precisely, continuous differentiability) of the generator --- which is not necessary in the weakest definitions of Bregman divergences. One can operate them even in the nondifferentiable setting by using subgradients. We will clarify this in the revision.
>
> **Q:** Why permutation-equivariance?
>
> **A:** We talk about permutation-equivariance mainly for expository purposes, to underscore that in this paper we focus on decoders which view different tokens symmetrically. You are right that we don't have to enforce it separately since it automatically holds for our setup. If you suggest, we would be happy to remove it.

---

> > ### Comment · Reviewer_4MPc · 2025-08-03
> >
> > Thanks for the rebuttal. This may be personal preference, but to me, any theoretical generality needs to be just enough for interpretation or for practical benefits. Bregman divergence probably not the most general divergence that has this effect, but it is not clear to me why this generality is even needed. In other words, using Bregman’s divergence allows a general framework, but its’ generality does not bring any new practical benefit, but more like a pure theoretical curiority. The evaluation also does not bring any obvious benefit to practice.
> >
> > The author may have misunderstood my intention of the “simple computation” comment, by which I meant the computation of top-k generation itself. Simplicity is the overwhelming benefits for taking this approach in LLMs. It is clear that the justification of top-k using an optimization formulation is only reassuring in nature, but not the original main motivation to adopt this approach.
> >
> > In general, though there is nothing wrong with the work, I did not find anything exciting. I’ll like to keep the score.

---

> ### Author Response · Authors · 2025-08-03
>
> Thanks so much for your comments!
>
> We agree with your comments regarding simplicity and computation; and sorry for giving the impression of misunderstanding---we think we are on the same page here. Even though simplicity is the main motivation for top-k, there are also other simple and good methods such as top-p and min-p. So what is special about top-k? Our investigation reveals some answers, which were certainly non-obvious a priori. So in this sense, we think our work adds some insight. We will add additional discussion about this to the paper.
>
> Additionally, the Bregman framework is not only general, but we would also point out that it provides a new way for future researchers to *generate* novel task-appropriate top-k methods as they please; all that they will need is decide on the shape of the Bregman loss function that their application needs. Indeed, even within the alpha-family that we consider as an example, there is no a-priori reason that the hyper parameter setting alpha=1 (vanilla top-k), of all alphas, should uniformly dominate all other choices in practice. So we view our example Bregman families not just as a generalization for its own sake, but as an *actionable* generalization.
>
> Regarding the comment that our new method "does not bring any new practical benefit", we would like to reiterate our point that (1) alpha-decoding with alpha=2 is more robust than top-k at higher temperature, and (2) achieves better/lower perplexity than top-k decoding. The level of comprehensiveness of these experiments (especially with our new models---Qwen, Phi-3, and new dataset---TriviaQA) is close in comprehensiveness to that of even the most thorough recent empirical papers in the field, for instance the min-p paper (Turning Up The Heat, ICLR 2025 Oral). We will add more emphasis to these results in the revision.
>
> Regarding the Bregman framework, to us it is quite exciting that we can develop a unified theoretical framework at this level of generality. Everyhing goes through, and we get interesting examples that show some promise as well (alpha-Bregman). From a theoretical standpoint, we believe that this gives ample evidence that this is a good framework; though we understand that opinions can differ, especially from more applied perspectives. We will explain our motivation in more depth in the revision.
>
> Again, we are very grateful for engaging with our paper, and for giving us the opportunity to clarify our points. While we certainly understand where you stand, we do hope that, if indeed all your concerns are addressed and "there is nothing wrong with this paper", then you can reconsider the evaluation.
>
> Sincerely,
> The Authors

---

### Official Review · Reviewer_wNkD · 2025-07-02

**Clarity:** 3
**Significance:** 3
**Originality:** 4
**Rating:** 5
**Confidence:** 4

**Summary:**

This work discusses theoretical foundations of top-$k$ decoding through a generalized class of Bregman decoders. More concretely, they formulate “generalized top-$k$ decoding” as $l_0$ sparsity-regularized Bregman divergence minimization. The asymmetry of Bregman divergences then yields a primal and dual formulation of the problem, which are treated independently due to their differing properties. It is noted that Bregman divergences are interesting as without regularization the optimization problem yields exactly $p$. The authors proceed to derive the renormalization map of top$k$ decoding for a fixed sparsity pattern, which simplifies the problem to $\arg\min_{\hat p \in \Delta_k} \text{Div}(\hat p , x)$. Whereas the authors make use of existing results which derive the primal normalization, they further derive an optimal dual renormalization. They then show that both Bregman decoding strategies are greedy, i.e., that the recovered entries of p are indeed the most probable (under some conditions on $\phi$), and followingly show that in the greedy top-$k$ setting both Bregman decoding strategies are $k$-convex. As a practical example, the authors then discuss a class of Bregman divergence, namely, the Havrda-Charvat-Tsallis $\alpha$-entropies, for which a few special instances exist with easy-to-compute renormalization maps. Finally, the authors compare the $\alpha$-entropies with basic top-$k$ decoding across two models and two datasets in (1) a full decoding setting, where $k^*$ is adaptively chosen, and (2) a setting where $k$ is fixed, and the renormalization maps are evaluated.

**Questions:**

### Questions

- If I understand correctly, adaptive $k$-sampling finds a $k^*$ at each decoding step, which is described in L.151 as being $O(V \log V)$ operation for $V \sim 10^5$. This seems quite heavy. How does this affect decoding runtime? How would you envision speedups?

- Re. Fig.3: is there a theoretical/qualitative reason to expect other members of the $\alpha$-entropies to perform better than the $\alpha=1$ basic top-$k$ decoding? A qualitative discussion would help.

**Ethical Concerns:**

["NO or VERY MINOR ethics concerns only"]

**Final Justification:**

The paper is theoretically sound, and provides a novel, and exciting perspective on top-k decoding as $\ell_0$-regularized Bregman divergence minimization. As shown in the discussions, the experimental section is not fully satisfying in terms of the evaluated settings. However, after engaging with the authors, they provide some additional experiments and vastly extending the empirical section may be beyond the scope of the current study.

**Limitations:**

yes

**Quality:**

3

**Strengths And Weaknesses:**

### Strengths

- The theoretical framing around the top-$k$ heuristic as $\ell_0$-regularized Bregman divergence minimization is sound, novel, and exciting: it is intuitive to derive a family of optimal top-$k$ decoding strategies by minimizing divergence between $p$ and its truncated version. Further, it is fascinating to think about *families* of top-$k$ decoding strategies immediately derived from families of divergences.

- The proposed method addresses two problems within the same optimization objective; adaptive top-$k$ sampling during decoding, and finding optimal normalization maps.

- Even though the content is rather dense and technical (§3), the presentation is clear and well-structured. Concrete applicability is nicely shown using Havrda-Charvat-Tsallis $\alpha$-entropies in §4 and in initial experiments in §5.

---

### Weaknesses

Although the paper rightfully claims that the contributions are mainly theoretical, I think the experimental section could still benefit from some improvement to complete the paper.

- Evaluates limited settings (models, datasets, regularization strengths, …) yielding uncertainty about the generalization of findings. A straightforward expansion could just be to expand it to the same settings as the referenced paper by Chen et al. (L.287).
- Leaves out an interesting analysis about what values of $k$ are chosen by the algorithm, as a function of the regularization strength. Further, does $k^*$ significantly change during decoding? This may tell us something interesting about failure modes of fixed-$k$ decoding.
- Only covers primal decoding, leaving open an interesting comparison between the two asymmetric formulations, although I assume this is left out as nice normalization maps only arise in the primal case?
- I recommend a better visual grouping for both Figure 3 and Table 1 as they are a bit hard to initially navigate.

Further, it is unclear how well this approach generalizes to other generators, i.e, if the class of Havrda-Charvat-Tsallis $\alpha$-entropies is a special outlier class or other useful such families exist.

---

#### Minor

- L.26-29; The listing of “decoding methods” is a bit imprecise to me. As I know it, best-of-N is used for post-hoc sample selection (thus, not really a decoding method), whereas top-$k$ and top-$p$ are token level sampling strategies, “temperature scaling” just means the temperature recalibration of probabilities. A more nuanced distinction and situation of top-$k$ decoding would be useful.
- L. 44-54: it is initially unclear (but cleared up later on) what the correspondence of $k$ and $\alpha$ is this early-on. The passage also seems a bit out of place. I recommend reformulating this passage to avoid confusion.

---

> ### Author Rebuttal · Authors · 2025-07-31
>
> Thank you so much for your careful reading, favorable evaluation, and insightful question and comments! We now respond to your points in the Weaknesses and Questions sections.
>
> -----------------------
>
> **Weaknesses:**
>
> First, we agree with your presentational comments about Ll. 26-29 and 44-54, these will be improved!
>
> **Q:** Evaluates limited settings (models, datasets, regularization strengths, …). Expand it to settings in Chen et al.?
>
> **A:** We have added experiments with larger models and new datasets, see response to reviewer 1 Q7. Our open-ended generation setup directly follows Chen et al.: same dataset (WebText test split) and evaluation metrics. Since we evaluate under fixed truncation size k (partial evaluation), other decoding methods (e.g., top-p, min-p, typical decoding) effectively reduce to top-k. Thus, our comparisons isolate the effect of normalization under various pre-defined k values.
>
> **Q:** Leaves out interesting analysis about values of $k$ chosen by algorithm as a function of $\lambda$. Does $k$ significantly change during decoding?
>
> **A:** Thank you for these insightful questions! For the latter question we will make sure to include various plots of adaptivity in $k^\*$ to the manuscript: as one would anticipate from their adaptivity properties, the Bregman decoders do choose diverse values of $k^\*$ for different token distributions throughout our experiments. Plotting the histograms of the distributions of $k^\*$ per-task (we unfortunately are not allowed to include images anymore by the new conference rules), one can observe distributions of $k^\*$ over $k \geq 1$ that are peaked at $k = 1$ and then near-monotonically decay until modest values of $k \in [20, 50]$. In other words, the decoding methods choose a diverse set of small $k$s. (One can stretch these distributions further out by making the regularization parameter smaller, $\lambda \to 0$; and conversely, can concentrate $k^\*$ on $1$ by increasing $\lambda$.)
>
> To illustrate this numerically without being able to show histograms, below we provide the standard deviations and Shannon entropy for the distributions of $k^\*$ on a subset of the Llama and Phi evaluations, on GSM8K and TriviaQA respectively, for $\lambda = 0.0001$ and $\lambda = 0.001$:
>
> **Table: Std dev (and entropy) of average $k^\*$ on GSM8K using LLama3.1-8B-instruct with $\lambda = 0.0001$.**
>
> | Temp | $\alpha=1.5$ | $\alpha=2.0$ |
> |------|-----------------|-----------------|
> | 0.3 | 3.1 (2.6) | 2.0 (2.2) |
> | 0.7 | 4.0 (2.8) | 2.5 (2.4) |
> | 1.0 | 7.3 (3.2) | 3.7 (2.7) |
> | 1.5 | 18.0 (4.0) | 7.8 (3.5) |
>
> **Table: Std dev (and entropy) of average $k^\*$ on TriviaQA using Phi-3-medium-4k-instruct with $\lambda = 0.001$.**
>
> | Temp | $\alpha=1.5$ | $\alpha=2.0$ |
> |------|-----------------|-----------------|
> | 0.3 | 8.2 (4.2) | 4.0 (3.3) |
> | 0.7 | 7.5 (4.2) | 3.8 (3.3) |
> | 1.0 | 7.0 (4.2) | 3.6 (3.4) |
> | 1.5 | 6.6 (4.3) | 3.6 (3.5) |
>
> Now, while it is clear that as $\lambda$ increases, $k^\*$ decreases and vice versa, deriving the asymptotics of the optimal $k^\*$ as a function of $\lambda$ and $\alpha$ (with or without assumptions on $p$) turns out to be an intricate task that does not to our knowledge immediately fall out of known techniques.
>
> For a concrete observation on that front, one can derive via a brief calculation that $k^\*$ is the floor or the ceiling of $1/(\lambda \alpha)^{1/\alpha}$ when $p = (1/V, \ldots, 1/V)$.
> But in fact, our preliminary experiments suggest that the scaling $k^\* \sim 1/(\lambda \alpha)^{1/\alpha}$ holds much more broadly. It also faithfully approximates the optimal $k^\*$ in expectation over drawing uniformly random (or other well-behaved Dirichlet) random probability vectors $p$ from the $V$-simplex (unfortunately we cannot include the plot confirming so by the new conference rules). Thus, this relationship $k \sim 1/(\lambda \alpha)^{1/\alpha}$ should serve as a good guideline for tuning $\lambda$ to ensure the desired ballpark of $k^\*$. Once again, we had not included this information to keep this paper focused and fit in the page limit.
>
> **Q:** Only covers primal decoding, leaving open comparison between primal and dual?
>
> **A:** We agree that a more detailed empirical study of the dual family and comparisons to the primal family are very interesting directions! For this manuscript, we mainly aimed for a minimalistic evaluation with the goal to show that the new decoding rules are worth a more substantial empirical investigation in future work, and in this sense aimed to pick out just a couple natural primal decoding strategies, that we also gave as an example in Section 4 right before that, which would demonstrate some of the ``first order effects'' of using them instead of vanilla top-$k$.
>
> There indeed appear to be some subtle connections and correspondences between the primal and dual families, which we thought was best to leave to follow-up work to keep the presentation focused and to do proper justice to this question. For instance, primal and dual decodings not only coincide at $\alpha=2$ by symmetry, but also have similar but subtly different renormalizations for other values of $\alpha$. As another observation, the primal and dual families have different sensitivities to changing the parameter $\alpha$ for various values of $\alpha$; so hypothetically, a practitioner with prior knowledge about the approximate range of the optimal $\alpha$ may want to correspondingly choose primal or dual decoding.
>
> **Q:** Better visual grouping for Figure 3, Table 1.
>
> **A:** We would be very happy to improve the visual grouping of the empirical results' displays (Figure 3 and Table 1). As simple improvements, we could bring them closer together in the document (e.g., to appear on the same page one under the other), and we could arrange the sub-figures in Figure 3 in a $2 \times 2$ grid for better interpretability (given that extra space would be available later). As for Table 1, arranging it in a non-confusing way was admittedly tricky for us, given that more than $2$ dimensions had to be presented; if you have any desired presentational tweaks in mind on that front, please let us know.
>
> **Q:** Generalization to generators beyond Havrda-Charvat-Tsallis-entropies.
>
> **A:** To us, going beyond $\alpha$-decoders is an exciting direction for future work. Indeed, $\alpha$-decoding is a rich, concrete and interpretable class of decoders, but we don't necessarily believe it ``spans'' the whole space of decoding behaviors one can elicit from Bregman divergences --- and we expect and hope that other decoding-appropriate Bregman scores will be designed/discovered in follup-ups!
>
> For our part, to help kickstart such future work, we have put effort into refining assumptions under which our structural results provably hold --- and as a result, our theory (i.e., the greedy and k-convexity properties) does apply broadly to Bregman scores beyond Havrda-Charvat-Tsallis-entropies.
>
> -----------------------
>
> **Questions:**
>
> **Q:** Adaptive-sampling $O(V\log V)$ complexity for $V\sim 10^5$ seems heavy, how does this affect runtime? Envisioned speedups?
>
> **A:** There are indeed some natural speedup options. First and foremost, in practice one would usually restrict attention to $k \leq V'$ for some $V' < V$ (e.g. $V' = 200$ or $V' = 50$ or even smaller) --- thus the complexity becomes $O(V' \log V')$, which is very manageable. (In fact, in our evaluations we selected $V'=50$, hence getting barely any runtime overhead.)
>
> Secondly, if one doesn't want to commit to a particular $V'$, one can still take advantage of the prior knowledge that $k^\*$ is likely small, while keeping the rigorous guarantees: Instead of binary search, just perform *exponential search* (gallop by trying $V' = 1, 2, 4, \ldots$, and then perform binary search once a true upper bound $V'$ is found) --- this will give the time complexity $O(k^\* \log k^\*)$.
>
> **Q:** Re. Fig.3: qualitative discussion about why other Bregmans can outperform vanilla top-$k$?
>
> **A:** This is a great question, and we will be happy to add explicit discussion of this to the empirical section. Theoretically and qualitatively (and as schematically shown in the Figure in lines 44-54), if we keep $k$ fixed and gradually increase $\alpha$, the $\alpha$-Bregman renormalization/projection will go from boosting higher-probability tokens (for small $\alpha$), to boosting lowest-probability tokens among the top-$k$ (for large $\alpha$ and in the limit as $\alpha \to \infty$). (By ``boosting'' we mean how the renormalization redistributes the residual mass of the non-top-$k$ tokens between the top-$k$ ones). For this reason, we believe that e.g.\ the observed marked decrease in repetitiveness that the $\alpha \in \{1.5, 2\}$ decoding exhibits over standard top-$k$ (i.e., $\alpha=1$) is likely due to higher $\alpha$ leading to higher diversity of decoding.

---

> > ### Comment · Reviewer_wNkD · 2025-08-07
> >
> > I highly appreciate the thorough rebuttal and additional experiments. They help clear up some open questions. The additional experiments now align with prior evaluation setups from related work help address earlier concerns regarding the scope and generalization of the empirical results. To me, although not fully satisfying, it is acceptable to cover dual decoding as well as the generalization beyond the $\alpha$-decoders in further work, given the strong theoretical foundation the paper lays. For the final manuscript I highly encourage the authors to mention these insightful qualitative arguments (primal vs dual, other possible function families, performance gap to vanilla top-$k$), as it will help a reader's intuition.
> > However, I am still confused about your response about runtime complexity, as it seems from your response that you are limiting vocabulary size for open-ended generation? Is this a fixed truncation or is this done dynamically to the $V'$ most likely? I may be simply confused by the notation.

---

> > > ### Author Response · Authors · 2025-08-07
> > >
> > > Thank you so much for your followup. We sincerely appreciate your expertise, your effort in evaluating and providing feedback on our manuscript, and your support.
> > >
> > > For the updated version, we are fully on board with your suggestions on the extra qualitative and quantitative/empirical details to provide --- for the reader's understanding of the developed framework and techniques, as well as to better point to the ensuing future work directions: constructing novel Bregman-based decoders, further analyzing the primal vs. dual connection, tuning k* and lambda with respect to each other, and other topics. We'll make sure to implement these additions by making the best use of the extra space.
> > >
> > > And thank you for your clarification question, our apologies for this confusion!  Looking back at our response above we realize that the notation could suggest as if the vocabulary may be getting uniformly truncated (which is indeed not what we do). Indeed as you said, the vocab truncation is done dynamically/adaptively, per each token rather than for all tokens; and the optimal k* (for each respective token) is observed to stay within small limits (i.a. $\leq 50$) in our experiments, which by our k-convexity result certifies that small effective per-token vocab size suffices. We'll make sure to use less confusing notation for this explanation when including it in the revision. Please let us know if this makes better sense or if you have further follow-up questions on this or any other aspects --- which we'll be very happy to address!
> > >
> > > Thanks so much again,
> > >
> > > The authors.

---

### Official Review · Reviewer_HJHQ · 2025-07-05

**Clarity:** 3
**Significance:** 3
**Originality:** 3
**Rating:** 5
**Confidence:** 3

**Summary:**

The paper provides a theoretical fundation of top-k decoding as solving an ell_0-regularized separable Bregman divergence minimization problem.

It proves that (i) greedy selection of the top-k entries is always optimal and (ii) the resulting cost is discretely convex in k, enabling an exact O(V log V) binary-search algorithm to pick the optimal k.

This framework subsumes standard top-k (the KL case) and introduces a family of alpha-Bregman decoders that re-weight remaining probabilities, matching or surpassing top-k performance in open-ended text and GSM8K math reasoning tasks.

**Questions:**

Q1. Your guarantees rely on strong smoothness/convexity assumptions on phi (Assumptions 3.1 & 3.2) plus either (A1) phi^{prime} convex or (A2) monotone elasticity. Which common divergences or practical temperature/logit-transformations used in LLM decoding fail these conditions, and how would the framework extend to them?

Q2. lambda critically governs the optimal k. Could you provide tuning rules for lambda?

Q3. Equation (6) proves discrete convexity for fixed lambda. If lambda is adapted per step, does the convexity guarantee still hold, and how would this affect the binary-search algorithm?

Q4. Dual Bregman decoding requires root-finding in a potentially non-convex landscape. What empirical evidence can you provide that the proposed solver always converges to the global optimum on large-scale logit vectors and mixed-precision hardware?

Q5. Runtime analysis counts O(V log V) per token, dominated by a full sort. Have you benchmarked end-to-end latency on real GPUs versus standard softmax + top-k? Are approximate partial-sort methods viable without breaking optimality?

Q6. For alpha-Bregman decoders, what intuition or guidance exists for choosing alpha versus lambda? Is there any interaction that can lead to degenerate cases (e.g., alpha to infty with small lambda)?

Q7. Experiments report results on WebText and GSM8K, but only with LLaMA 8B.
Have you tested on larger models or different domains (dialog, code) to confirm that the gains are not model-specific?

**Ethical Concerns:**

["NO or VERY MINOR ethics concerns only"]

**Final Justification:**

The authors addressed my concerns.
I would like to keep my high score.

**Limitations:**

Even if the primary goal is to establish a theoretical foundation, I believe the paper would be more complete if it included experimental comparisons with other top-k decoding methods.

**Paper Formatting Concerns:**

I didn't notice any format issues.

**Quality:**

3

**Strengths And Weaknesses:**

Strengths:
The paper gives the first exact, discrete-convex formulation of top-k decoding and an O(V log V) algorithm, putting the method on solid theoretical ground.
It unifies top-k, sparsemax, and entmax within a single Bregman + ell_0 framework and introduces tunable alpha-Bregman decoders that broaden design options.
Greedy optimality and discrete convexity proofs cover both primal and dual cases and are supported by indicative language-generation and math-reasoning experiments.

Weaknesses:
The optimal k is highly sensitive to λ, yet the paper offers no principled way to set λ across temperatures, tasks, or sequence positions.
Each time step needs full-vocabulary sorting and root-finding; despite the O(V log V) bound, latency and numerical stability may hinder use on large LLMs.

---

> ### Author Rebuttal · Authors · 2025-07-31
>
> Thanks so much for your careful reading and interesting questions! We now address your Weaknesses comments and then your questions.
>
> **Q:** Optimal k is sensitive to $\lambda$, how to set?
>
> **A:** In our empirical evaluations, the optimal $k$ is not highly sensitive but is rather reasonably robust to the choice of $\lambda$. As can be seen e.g. from Table 2 (in the Appendix), even for values of $\lambda$ on different orders of magnitude, the average optimal $k$ does not fluctuate too much (in particular, it's never larger than $\approx 50$ in our experiments). That being said, there is a reasonable ballpark way to set $\lambda$ to get reasonable optimal $k$: please see our answers below to your Q2, and to Reviewer 2, about the relationship between the optimal $k$ and the hyperparameter $\lambda$.
>
> **Q:** New decoders need full sorting and root-finding, which can hinder use on large LLMs.
>
> **A:** Actually, we would argue that the computational and numerical aspects of our decoders are quite a bit more optimistic than this assessment, in a few ways, and that they shouldn't cause big issues when used on large LLMs (certainly they don't cause issues in our empirics).
>
> As for root finding/numerics: First, some important special cases like primal $\alpha \in \{1.5, 2\}$ decoding (and the original top-$k$ decoding) have closed-form renormalization maps hence don't require root finding. In the other cases when root finding is required, the functions whose root we need to find are scalar, increasing, and well-behaved, precluding numerical issues that could occur for nonmonotonic or nonsmooth functions.
>
> As for full-vocabulary sorting: This is also not quite required in practical settings. To see that, note that the optimal $k^\*$ for practical purposes is usually not greater than (e.g.) $V' \in [50, 200]$, which is $\ll V$. We can certainly enforce that we limit to the top $200$ tokens if need be. Even without enforcing this, our methods can in fact run faster in such settings.
> Specifically, they can run in $O(V + V' \log V')$ time where $V'$ is a true upper bound on the optimal $k^\*$.
> This can be significantly more efficient than the worst-case bound $O(V \log V)$.
>
> To see this, we note:
> (1) identifying the top $V'$ probabilities doesn't require full vocab sorting and can be done in $O(V)$ time (e.g. using a heap);
>
> (2) instead of binary searching for $k \in [1, V]$, one can search for $k \in [1, V']$. If $V'$ is unknown, then one can use *exponential search* (gallop by trying $V' = 1, 2, 4, \ldots$, and then perform binary search once a true upper bound $V'$ is found---i.e., when the discrete difference is positive according to the convexity property) to find the correct $V'$ and binary search within $[1, V']$.
>
> This will take $O(V' \log V')$ time (see also our answer to the similar question from Reviewer 2).  While this paper was written mainly from the theoretical standpoint, which was the reason we gave the worst-case bound $O(V \log V)$, we will be happy to clarify this in a remark in the revision.
>
> ---
>
> **Q1:** On our assumptions on Bregmans.
>
> **A:** Assumption 3.1 suffices for all our primal decoding results, and is very mild (it only requires a well-defined Bregman divergence with first-order smoothness) hence is satisfied by KL (the main Bregman score used in practice) and in fact all our primal rules (Havrda-Charvat-Tsallis entropies) for $\alpha \neq 0$ (Section 4). The dual conditions look complex, bur are also relatively mild and are satisfied for all Havrda-Charvat-Tsallis entropies with $\alpha > 1$. Intuitively, all our assumptions will hold for Bregmans whose generators are "not too singular" at the boundary ($x \to \{0, 1\}$).
>
> **Q2:** On tuning $\lambda$ to control $k$.
>
> **A:** While clearly $\lambda$ increases, $k^\*$ decreases and vice versa, deriving the asymptotics of the optimal $k^\*$ as a function of $\lambda$ and $\alpha$ (with or without assumptions on $p$) turns out to be an intricate task that does not to our knowledge immediately fall out of known techniques.
> For a concrete observation, one can derive via a brief calculation that $k^\*$ is the floor or the ceiling of $1/(\lambda \alpha)^{1/\alpha}$ when $p = (1/V, \ldots, 1/V)$.
> But in fact, our preliminary experiments suggest that the scaling $k^\* \sim 1/(\lambda \alpha)^{1/\alpha}$ holds much more broadly. It also faithfully approximates the optimal $k^\*$ in expectation over drawing uniformly random (or other well-behaved Dirichlet) random probability vectors $p$ from the $V$-simplex (we can't include the plot by new NeurIPS rules). Thus, $k \sim 1/(\lambda \alpha)^{1/\alpha}$ should serve as a good guideline for tuning $\lambda$ to ensure the desired ballpark of $k^\*$. (We had not included this information to keep this paper focused and fit in the page limit.)
>
> **Q3:** On discrete convexity for adaptive $\lambda$.
>
> **A:** The convexity of the cost function in $k$ indeed holds regardless of $\lambda \geq 0$ (as $\lambda k$ is linear in $k$). That being said, the shape and the location of the argmin $k^\*$ of the cost function will change, and binary search may need to be redone. Thus, we recommend selecting a fixed $\lambda$ for each invocation of the decoder.
>
> **Q4:** Non-convexity in dual Bregman decoding?
>
> **A:** One of the novel structural insights into the dual case the optimization of $\hat{p}$ reduces to the case where $\hat{p}_i \geq p_i$ for $i \in [k]$, and in this regime *the problem becomes convex*! So in fact, we show that the optimum must lie within a convex subdomain of the generally nonconvex problem, and we are able to solve the problem within that region.
> Thus, both theoretically and practically the problem becomes convex.
>
> **Q5:** Runtime of decoding methods, improvements to sorting algorithm?
>
> **A:** For new Bregman decoders ($\alpha \in \{1.5, 2\}$) that we evaluated, the runtime is actually (nearly) the same as for standard top-k ($\alpha=1$). It is important to note that the algorithmic primitive here is binary search (for $k \in [1, V]$) rather than sorting. Practical speedups thus can be obtained by searching for best $k$ in a smaller range (e.g. $k \in [1, V']$ for $V' < V$, e.g. $V' = 50$); and retrieving the top $V'$ tokens for some constant $V' \ll V$ takes only $O(V)$, not $O(V \log V)$, time (e.g. via heaps).
> Further, in practice, approximation algorithms could be tried, and should perform well if relatively accurate.
>
> **Q6:** Interactions between $\alpha$ and $\lambda$ in the limiting cases? Choosing them jointly?
>
> **A:** Any pair of $\alpha$, $\lambda > 0$ can be used; e.g. if some $k^*$ is targeted, could use 3-way formula btw $k, \alpha, \lambda$ in Q2.
>
> The limiting behavior for both $\alpha$ and $\lambda$ is well-defined and well-interpreted, and their limiting interaction is nicely disentangled:
> Intuitively, $\lambda$ controls how many tokens $k$ end up selected while $\alpha$ controls the shape of renormalization on these $k$ tokens. When $\alpha \to \infty$: regardless of $\lambda$, renormalization converges to ``water-filling'' (i.e., it adds mass to the smallest selected token probabilities). When $\lambda \to 0$: regardless of $\alpha$, *all tokens* ($k=V$) will be selected in the limit; conversely, when $\lambda \to \infty$, only one token ($k=1$) will be selected.
>
> **Q7:** Experiments: larger models, different domains (dialog, code).
>
> **A:** In addition to the experiments presented in the main paper, we have conducted further evaluations to strengthen our findings:
>
> 1. Evaluated larger models
> — **Phi-3-medium-4k-instruct (14B)** and **Qwen2.5-14B-Instruct**
> — on both open-ended text generation and GSM8K.
>
> 2.Added **TriviaQA** evaluations, using LLaMA-3.1-8B-Instruct,
> **Phi-3-medium-4k-instruct (14B)** and
> **Qwen2.5-14B-Instruct**.
>
> Due to the character limit in this response, we can only include a subset of our results.
> We would be happy to provide the full results upon request.
>
> **Selected Evaluation Results:**
> > Open-ended text generation:
>
> Primal decoding matches top-k in PPL and repetition diff.
>
> Primal-2.0 consistently yields lowest PPL gap.
>
> ### Phi-3-medium-4k-instruct
>
> | k| PPL Diff|||Rep Diff %|| |
> | --|--------|-----|-------|----------|-----|------- |
> || Top-k| P-1.5|P-2.0|Top-k| P-1.5|P-2.0 |
> | 10|4.8| 4.2|**4.2**|1.1| 0.8|**0.6** |
> | 20|4.1| 3.7|**3.4**|0.5| 0.5|**0.2** |
> | 30|3.7| 3.3|**2.9**|0.3| 0.6|**0.3** |
> | 40|3.4| 3.0|**2.6**|0.3| 0.3|0.4 |
> | 50|3.2| 2.8|**2.4**|**0.1**| 0.4|**0.1** |
>
> ---
>
> ### Qwen2.5-14B-Instruct
>
> | k| PPL Diff|||Rep Diff %|| |
> | --|--------|-----|-------|----------|-----|------- |
> || Top-k| P-1.5|P-2.0|Top-k| P-1.5|P-2.0 |
> | 10|6.3| 5.9|**5.5**|1.2| 0.8|1.2 |
> | 20|5.3| 4.8|**4.3**|1.4| 1.3|**1.2** |
> | 30|4.8| 4.1|**3.6**|1.0| 1.6|**0.8** |
> | 40|4.3| 3.7|**3.1**|1.0| 1.2|1.1 |
> | 50|4.0| 3.3|**2.7**|1.2| 1.2|1.4 |
>
> ---
>
> > GSM8K: We only report \lambda=0.0001.
> Primal decoding matches top-k.
> Top-k uses average k\* from primal.
> At high temp (e.g. 1.5), top-k degrades faster.
>
> ### Phi-3-medium-4k-instruct
>
> | T|P-1.5|P-2.0|Top-k-1.5|Top-k-2.0 |
> | ---|-----|-----|---------|--------- |
> | 0.3|87.5| 87.5| 86.2| 86.5|
> | 0.7|85.0| 84.9| 85.6| 85.3|
> | 1.0|82.7| 82.1| 81.4| 82.7|
> | 1.5|72.2| 76.0| 62.6| 65.8|
>
> ---
>
> ### Qwen2.5-14B-Instruct
>
> | T|P-1.5|P-2.0|Top-k-1.5|Top-k-2.0 |
> | ---|-----|-----|---------|--------- |
> | 0.3|81.9| 82.3| 82.0| 82.4|
> | 0.7|79.6| 79.8| 80.0| 80.1|
> | 1.0|78.5| 79.3| 78.8| 79.4|
> | 1.5|74.9| 74.9| 71.2| 75.9|
>
> ---
>
> > TriviaQA: We only report \lambda=0.0001
>
> ### Llama3.1-8B-Instruct
>
> | T|P-1.5|P-2.0|Top-k-1.5|Top-k-2.0 |
> | ---|-----|-----|---------|--------- |
> | 0.3|66.7| 67.1| 67.2| 67.6|
> | 0.7|62.7| 62.7| 63.8| 63.7|
> | 1.0|58.0| 59.1| 58.6| 60.1|
> | 1.5|49.2| 52.6| 50.2| 51.3|
>
> ---
>
> ### Phi-3-medium-4k-instruct
>
> | T|P-1.5|P-2.0|Top-k-1.5|Top-k-2.0 |
> | ---|-----|-----|---------|--------- |
> | 0.3|59.8| 59.4| 59.4| 59.4|
> | 0.7|54.5| 54.4| 56.2| 54.7|
> | 1.0|48.7| 48.6| 48.6| 48.6|
> | 1.5|38.2| 39.9| 36.0| 37.7|

---

> > ### Comment · Reviewer_HJHQ · 2025-08-08
> >
> > Thank you for your detailed response to my questions.
> > They clarified my concerns.
> > I have already achieved a high score from the outset, so I would like to maintain it at this level.

---

> > > ### Author Response · Authors · 2025-08-08
> > >
> > > We truly appreciate your thorough review and your valuable and thoughtful feedback! We are very glad that our response has clarified your concerns --- and thank you again for supporting our paper.

---

### Comment · Area_Chair_xDQX · 2025-08-05
**[From AC] Reviewer Discussion Reminder**

Dear Reviewers,

Thank you for your time and effort in reviewing the paper.

As the reviewer-author discussion period ends on August 6 at 11:59 PM AoE, please take a moment to acknowledge the rebuttal and engage in the discussion if you haven’t already.

Thank you again for your contributions to the review process.

Best,\
Area Chair

---

### Note · Authors · 2025-08-16

We would like to take this opportunity to thank the reviewers again for their active and productive engagement with our paper, and for their excellent and very helpful feedback and clarification questions. Having carefully incorporated information from our discussions with the reviewers into the exposition (and having performed further empirical evaluations as requested by reviewers; we'd like to highlight, at the same time, that we still squarely view this paper as a theoretical contribution), we are confident that readers will appreciate the improvements to the manuscript’s presentation and the additional qualitative and quantitative insights. We are optimistic and excited for future work on the theory and practice of LLM decoding based on our theoretical framework — which we already caught a first glimpse of in the amount of excellent future work directions pointed out by the reviewers! Thank you so much again.

---

### Decision · Program_Chairs · 2025-09-17

**Decision:**

Accept (poster)

**Comment:**

This paper develops a theoretical framework for top-$k$ decoding by formulating it as an $\ell_{0}$-regularized Bregman divergence minimization problem, proving that greedy selection is always optimal and that the cost function is discretely convex, enabling efficient binary search for the optimal $k$. This unifies existing decoding rules and introduces new Bregman-based decoders, some of which show improved robustness and perplexity in preliminary experiments.

The proposed theoretical foundations of top-$k$ decoding are valuable in placing a widely used heuristic on solid theoretical ground and offering a principled generalization, though concerns remain regarding limited empirical validation, sensitivity to hyperparameters, and unclear practical benefits relative to the simplicity of standard top-$k$ (which should be included in the updated version).

During rebuttal, the authors addressed these concerns with additional experiments on larger models and datasets, clarified runtime considerations, and expanded the discussion of primal vs.\ dual decoding and adaptivity. Overall, the strong theoretical contributions and clarified empirical evidence make this a solid and timely paper.

Thus, I recommend acceptance.